# Homeostatic control of energy metabolism by monocyte-derived macrophages

Rui Martins[1,8], Birte Blankehaus[1,8], Faouzi Braza[1], Miguel Mesquita [1], Pedro Ventura[1], Sumnima Singh[1], Sebastian Weis [2,3,4], Maria Pires [1], Sara Pagnotta[1], Qian Wu [1,5], Sílvia Cardoso[1], Elisa Jentho [1], Ana Figueiredo[1], Pedro Faísca[1], Ana Nóvoa [1], Vanessa Alexandra Morais [1], Stefanie K Wculek [6], David Sancho[7], Moises Mallo[1] & Miguel P Soares [1]✉

## Abstract

**Multicellular organisms rely on inter-organ communication networks to maintain vital parameters within a dynamic physiological range. Macrophages are central to this homeostatic control system, sensing and responding to deviations of those parameters to sustain organismal homeostasis. Here, we demonstrate that dysregulation of iron (Fe) metabolism, imposed by the deletion of ferritin H chain (FTH) in mouse parenchymal cells, is sensed by monocyte-derived macrophages. In response, monocyte-derived macrophages support tissue function, energy metabolism, and thermoregulation via a mechanism that sustains the mitochondria of parenchymal cells. Mechanistically, FTH supports a transcriptional program promoting mitochondrial biogenesis in macrophages, involving mitochondrial transcription factor A (TFAM). Moreover, FTH sustains macrophage viability and supports intercellular mitochondrial transfer from donor parenchymal cells. In conclusion, monocyte-derived macrophages cross-regulate iron and energy metabolism to support tissue function and organismal homeostasis.**

**Keywords** Ferritin; Macrophages; Mitochondria; Homeostasis; Metabolism
**Subject Categories** Immunology; Metabolism

## Introduction

Iron (Fe) was co-opted through evolution to exchange electrons with acceptor (i.e., electrophile) and donor (i.e., nucleophile) molecules, partaking in vital biochemical processes, including some of those supporting mitochondrial function and energy metabolism (Muchowska et al, 2019; Teh et al, 2024). Presumably as an evolutionary trade-off (Stearns and Medzhitov, 2015), Fe can catalyze in a unfettered manner the production of hydroxyl radicals (HO⁻) and other reactive oxygen species (ROS), when exchanging electrons with mitochondrial superoxide ($O_2^{\bullet-}$) or with hydrogen peroxide ($H_2O_2$) (Winterbourn, 1995).

Cellular ROS accumulation can catalyze lipid peroxidation and in doing so, trigger programmed cell death via ferroptosis (Dixon et al, 2012). The fitness cost of uncontrolled Fe redox activity is limited by a number of evolutionarily conserved Fe-regulatory genes (Galy et al, 2024). These include ferritin h chain (FTH), the master regulator of Fe redox activity and bioavailability (Blankenhaus et al, 2019; Galy et al, 2024; Gozzelino and Soares, 2014; Harrison and Arosio, 1996).

FTH is highly expressed by Fe-recycling macrophages, a cell compartment that plays a central role in the control of organismal Fe homeostasis (Galy et al, 2024). Fe-recycling macrophages develop from yolk sac progenitors (Gomez Perdiguero et al, 2015), similar to other lineages of tissue-resident macrophages (Ginhoux et al, 2010). Throughout adult life however, these can be replaced by monocyte-derived macrophages (van Furth and Diesselhoff-den Dulk, 1984) that develop from bone marrow (BM) hematopoietic progenitors (van Furth and Cohn, 1968). This process of differentiation, from monocytes into tissue resident macrophages (Guilliams et al, 2018), occurs via the activation of transcriptional and epigenetic programs (Gosselin et al, 2014; Lavin et al, 2014), in response to tissue-specific cues (Amit et al, 2016; Okabe and Medzhitov, 2015). Heme, an Fe-containing protoporphyrin used as prosthetic group of hemoglobin, induces the genetic program supporting the development of Fe-recycling macrophages (Haldar et al, 2014).

Macrophages establish functional interactions with parenchyma cells in all tissues, early through embryonic development (Lazarov et al, 2023) and throughout post-natal life (Nobs and Kopf, 2021; Okabe and Medzhitov, 2015; Zhou et al, 2018). Under this conceptual framework, macrophages are essential to "support" the core effector functions of "primary" parenchyma (Adler et al, 2023; Meizlish et al,

[1]Gulbenkian Institute for Molecular Medicine, Lisboa, Portugal. [2]Department for Anesthesiology and Intensive Care Medicine, Jena University Hospital, Friedrich-Schiller University, Jena, Germany. [3]Institute for Infectious Disease and Infection Control, Jena University Hospital, Friedrich-Schiller University, Jena, Germany. [4]Leibniz Institute on Aging-Fritz Lipmann Institute, Jena, Germany. [5]Department of Radiology, Center of Regenerative and Aging Medicine, the Fourth Affiliated Hospital of School of Medicine, and International School of Medicine, International Institutes of Medicine, Zhejiang University, Yiwu, China. [6]Institute for Research in Biomedicine (IRB Barcelona), The Barcelona Institute of Science and Technology (BIST), Barcelona, Spain. [7]Centro Nacional de Investigaciones Cardiovasculares Carlos III (CNIC), Madrid, Spain. [8]These authors contributed equally: Rui Martins, Birte Blankehaus. ✉E-mail: miguel.soares@gimm.pt

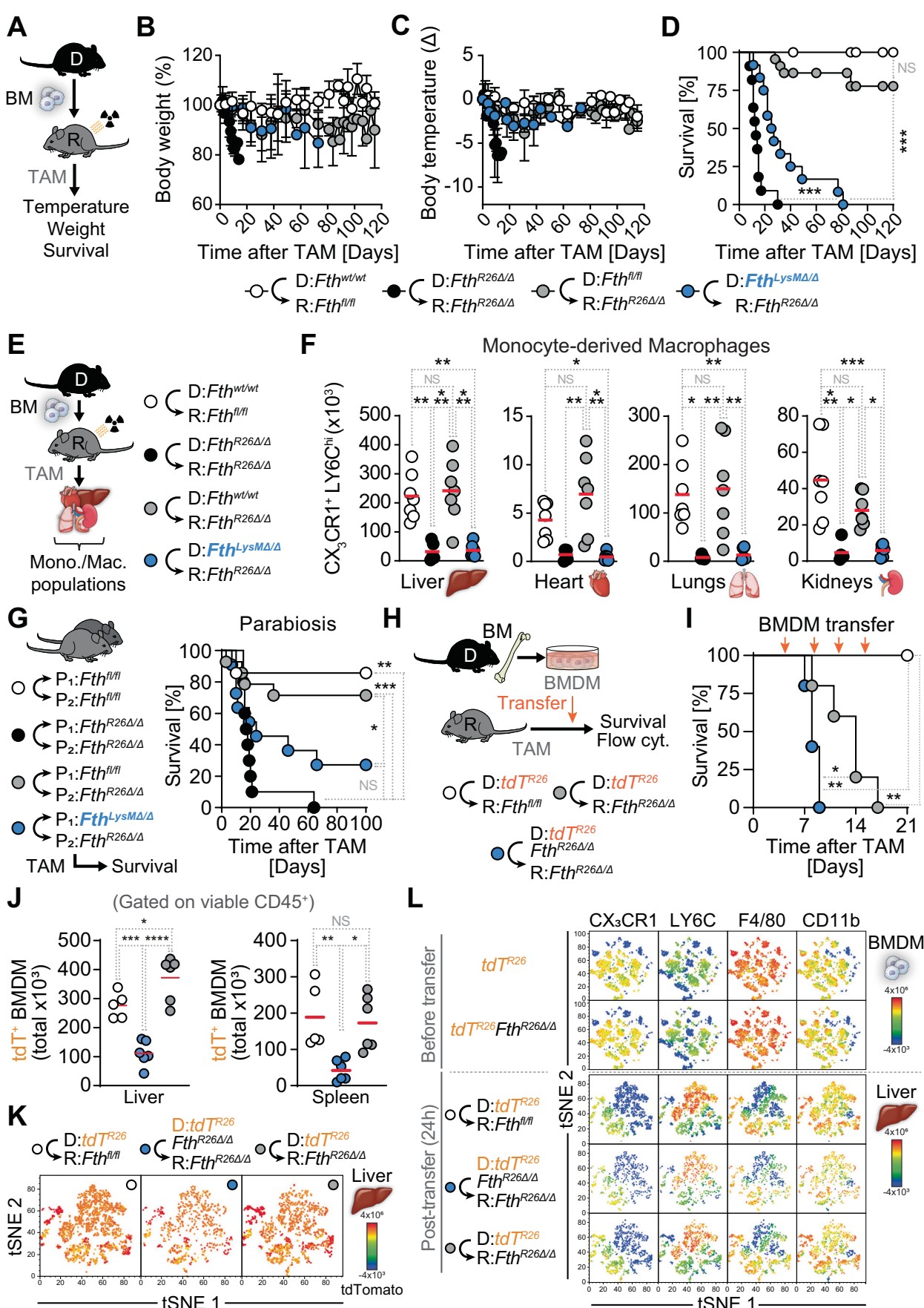

**Figure 1.** *Fth*-competent monocyte-derived macrophages rescue *Fth*-deleted mice.

(A) Schematic representation of TAM-induced *Fth* deletion in chimeric mice (day 0) and vital parameters monitored. Relative (B) Body weight, (C) Temperature and (D) survival of $Fth^{wt/wt} \Rightarrow Fth^{fl/fl}$ ($n = 7$–8), $Fth^{R26\Delta/\Delta} \Rightarrow Fth^{R26\Delta/\Delta}$ ($n = 6$), $Fth^{wt/wt} \Rightarrow Fth^{R26\Delta/\Delta}$ ($n = 10$–21) and $Fth^{LysM\Delta/\Delta} \Rightarrow Fth^{R26\Delta/\Delta}$ ($n = 9$–12) chimeric mice. Data in (B–D) is pooled from 6 independent experiments with similar trends. Data in (B, C) represented as mean ± SD. (E) Schematic representation of TAM-induced *Fth* deletion in chimeric mice (day 0) and flow cytometry analysis of monocyte/macrophage populations. (F) Absolute number of $CX_3CR1^+LY6C^{high}$ monocyte-derived macrophages (backgated as Ly6G$^-$CD11b$^+$F4/80$^{low}$) in the liver, heart, lungs and kidneys of $Fth^{fl/fl} \Rightarrow Fth^{fl/fl}$ ($n = 7$), $Fth^{R26\Delta/\Delta} \Rightarrow Fth^{R26\Delta/\Delta}$ ($n = 5$), $Fth^{fl/fl} \Rightarrow Fth^{R26\Delta/\Delta}$ ($n = 7$) and $Fth^{LysM\Delta/\Delta} \Rightarrow Fth^{R26\Delta/\Delta}$ ($n = 6$) chimeric mice, 7 to 19 days post-TAM administration. Data in (F) presented as individual values (circles) and mean (red bars), pooled from three independent experiments with similar trends. (G) Schematic representation of parabiosis, TAM administration (day 0) and survival of parabiotic $Fth^{fl/fl} \Leftrightarrow Fth^{fl/fl}$ ($n = 7$ pairs), $Fth^{R26\Delta/\Delta} \Leftrightarrow Fth^{R26\Delta/\Delta}$ ($n = 10$ pairs), $Fth^{fl/fl} \Leftrightarrow Fth^{R26\Delta/\Delta}$ ($n = 14$ pairs) and $Fth^{LysM\Delta/\Delta} \Leftrightarrow Fth^{R26\Delta/\Delta}$ ($n = 11$ pairs). Data in (G) pooled from four independent experiments with similar trends. (H) Schematic representation of adoptive transfer of in vitro-differentiated BMDM into recipient mice, following TAM administration (day 0). (I) Survival of $Fth^{fl/fl}$ or $Fth^{R26\Delta/\Delta}$ mice adoptively transferred with *Fth*-competent $tdT^{R26}$ or *Fth*-deleted $tdT^{R26}Fth^{R26\Delta/\Delta}$ BMDM. Times of adoptive transfer indicated by orange arrows. (J) Number of adoptively-transferred tdT$^+$ cells per liver and spleen from recipient mice, 24 h post-adoptive transfer, on day 7 post-tamoxifen treatment. Data in (J) represented as individual values (circles, $n = 5$–6 per group) and mean (red bars) and is pooled from two independent experiments. (K) tSNE analysis of tdTomato$^+$ cells recovered from the liver of control ($Fth^{fl/fl}$) or FTH-deficient ($Fth^{R26\Delta/\Delta}$) mice, following adoptive transfer with control, or FTH-deficient tdTomato reporter BMDM ($tdT^{R26}$, $tdT^{R26}Fth^{R26\Delta/\Delta}$, respectively), with tdTomato expression levels mapped onto the projections (L) tSNE plots showing marker expression profiles in control $tdT^{R26}$ or $tdT^{R26}Fth^{R26\Delta/\Delta}$ BMDMs, prior to adoptive transfer (top 2 rows), as well as marker expression profiles in tdTomato$^+$ cells recovered from the liver of control $Fth^{fl/fl}$ or $Fth^{R26\Delta/\Delta}$ mice, following adoptive transfer with control, or FTH-deficient tdTomato reporter BMDM ($tdT^{R26}$, $tdT^{R26}Fth^{R26\Delta/\Delta}$, respectively). Each plot displays the expression of the designated marker across the tSNE projection for each genotype. Data in (K, L) represented as a dot plot where dots indicate individual cells. tSNE was used to generate projections from 2500 cells (downsampled) for BMDM prior to adoptive transfer, and from 701 to 1932 cells for BMDM recovered from the liver following adoptive transfer (pooled from three mice per group). Survival analysis was performed using Log-rank (Mantel-Cox) test. One-way ANOVA with Tukey's range test for multiple comparison correction was used for comparison between multiple groups. NS: non-significant, *$P < 0.05$, **$P < 0.01$, ***$P < 0.001$. Source data are available online for this figure.

2021; Zhou et al, 2018). Iron metabolism exerts a major impact on macrophage function (Soares and Hamza, 2016), suggesting that macrophages sense and respond to iron in a manner that contributes to "support" cellular and tissue function (Winn et al, 2020).

Having established that regulation of Fe metabolism by FTH exerts a major impact on organismal energy metabolism in adult mice (Blankenhaus et al, 2019) and considering the impact of Fe metabolism energy homeostasis (Joffin et al, 2022), we asked whether FTH expression in macrophages impacts on organismal energy homeostasis. We found that monocyte-derived macrophages respond to dysregulation of Fe metabolism in parenchymal cells via an FTH-dependent mechanism that controls the activation of a transcriptional program associated with mitochondrial biogenesis and regulated in macrophages by the mitochondrial transcription factor A (TFAM). This is essential to support the mitochondria of parenchymal cells and restore organismal energy balance in mice lacking FTH in parenchymal cells. Moreover, FTH expression in macrophages is essential to support macrophage viability as they act as acceptor cells of intercellular mitochondrial transfer from donor parenchymal cells. We infer that monocyte-derived macrophages operate as a central component of an inter-organ surveillance system that cross-regulates Fe and energy metabolism to support organismal homeostasis.

## Results

### *Fth*-competent hematopoietic cells rescue chimeric *Fth*-deleted mice

Having established that regulation of Fe metabolism by FTH exerts a critical role in the control energy metabolism in adult mice (Blankenhaus et al, 2019), we sought to determine the relative contribution of FTH expression in hematopoietic *vs.* parenchyma (i.e., non-hematopoietic) to energy homeostasis. To this end we used $Fth^{R26fl/fl}$ mice allowing for global *Fth*-deletion (i.e. $Fth^{R26\Delta/\Delta}$) in response to tamoxifen (TAM) administration and $Fth^{wt/wt}$ or $Fth^{fl/fl}$ mice as controls (Blankenhaus et al, 2019). $Fth^{R26fl/fl}$ and control

$Fth^{wt/wt}$ or $Fth^{fl/fl}$ mice were lethally irradiated and transplanted with bone marrow (BM) cells from $Fth^{R26fl/fl}$ *vs.* control $Fth^{wt/wt}$ or $Fth^{fl/fl}$ mice, to generate chimeric mice carrying an *Fth* deletion in hematopoietic *vs.* parenchyma cells, in response to TAM administration. We confirmed that BM engraftment was >95%, as assessed by the relative proportion of donor *vs.* recipient CD45.1 *vs.* CD45.2 cells, respectively, 4 weeks after BM transplantation (Appendix Fig. S1A–C).

Systemic *Fth* deletion in hematopoietic and parenchyma cells from $Fth^{R26\Delta/\Delta} \Rightarrow Fth^{R26\Delta/\Delta}$ chimeras (i.e., $Fth^{R26fl/fl}$ mice reconstituted with $Fth^{R26fl/fl}$ BM) led to wasting (Fig. 1A,B; Appendix Fig. S1D,E), hypothermia (Fig. 1A,C; Appendix Fig. S1D,F) and death (Fig. 1A,D; Appendix Fig. S1D,G), within 10–40 days after TAM administration, consistent with global *Fth* deletion in adult $Fth^{R26\Delta/\Delta}$ mice (Blankenhaus et al, 2019). Control *Fth*-competent $Fth^{wt/wt} \Rightarrow Fth^{fl/fl}$ chimeric mice (i.e., $Fth^{fl/fl}$ mice reconstituted with $Fth^{wt/wt}$ BM) did not develop this lethal wasting syndrome in response to TAM administration at the same dosage and schedule (Fig. 1B–D; Appendix Fig. S1D–G).

*Fth* deletion specifically in parenchyma cells (i.e., $Fth^{wt/wt} \Rightarrow Fth^{R26\Delta/\Delta}$ chimeras; $Fth^{R26fl/fl}$ mice reconstituted $Fth^{wt/wt}$ BM) did not cause the development of wasting (Fig. 1B), hypothermia (Fig. 1C) or lethality (Fig. 1D). Moreover, *Fth* deletion specifically in hematopoietic cells (i.e., $Fth^{R26\Delta/\Delta} \Rightarrow Fth^{fl/fl}$ chimeras; $Fth^{fl/fl}$ mice reconstituted with $Fth^{R26fl/fl}$ BM) also did not cause wasting (Appendix Fig. S1D,E), hypothermia (Appendix Fig. S1D,F) nor mortality (Appendix Fig. S1D,G). These observations suggest that *Fth*-competent hematopoietic cells can rescue the lethal outcome of systemic *Fth* deletion in chimeric mice.

### *Fth*-competent myeloid cells rescue chimeric *Fth*-deleted mice

To establish whether myeloid cells are necessary to rescue the lethal outcome of *Fth* deletion in adult mice, we generated chimeric mice with BM cells from $Fth^{LysM\Delta/\Delta}$ mice, carrying an *Fth* deletion specifically in myeloid cells (Ramos et al, 2019; Wu et al, 2023). *Fth*-deleted myeloid cells failed to prevent the development of wasting (Fig. 1B), hypothermia (Fig. 1C) and death in $Fth^{LysM\Delta/}$

$^{\Delta}{\Rightarrow}Fth^{R26\Delta/\Delta}$ chimeras (i.e., $Fth^{R26fl/fl}$ mice reconstituted with $Fth^{LysM\Delta/\Delta}$ BM), with all chimeric mice succumbing within 80 days of TAM administration (Fig. 1D). This suggests that FTH expression in myeloid cells is essential to rescue the lethal outcome of global Fth deletion in chimeric mice.

Although $Fth^{LysM\Delta/\Delta}{\Rightarrow}Fth^{R26\Delta/\Delta}$ chimeras succumbed to Fth deletion, the temporal dynamics were delayed, as compared to $Fth^{R26\Delta/\Delta}{\Rightarrow}Fth^{R26\Delta/\Delta}$ chimeras. This likely reflects (i) mosaic/incomplete Fth deletion in $LysM^{Cre}$ cells, (ii) in vivo selection for cells that express some FTH or (iii) both. As such, subsequent analyses were conducted as the body weight of $Fth^{R26\Delta/\Delta}{\Rightarrow}Fth^{R26\Delta/\Delta}$ or $Fth^{LysM\Delta/\Delta}{\Rightarrow}Fth^{R26\Delta/\Delta}$ was reduced by more than 10% defining "early onset" and "late onset", respectively.

To establish whether Fth-competent lymphocytes are necessary to rescue the lethal metabolic collapse caused by systemic Fth deletion, we generated $Fth^{Cd2\Delta/\Delta}$ mice, carrying an Fth deletion specifically in lymphocytes. Fth deletion in lymphocytes did not compromise the capacity of hematopoietic cells to prevent the development of wasting (Appendix Fig. S1H,I), hypothermia (Appendix Fig. S1H,J) nor lethality (Appendix Fig. S1H,K) in $Fth^{Cd2\Delta/\Delta}{\Rightarrow}Fth^{R26\Delta/\Delta}$ chimeras (i.e., $Fth^{R26fl/fl}$ mice reconstituted with $Fth^{Cd2\Delta/\Delta}$ BM). This suggests that FTH expression in lymphocytes, the most abundant population of circulating leukocytes in adult mice, is not required to rescue the lethal outcome of Fth deletion in parenchyma cells.

## Monocyte-derived macrophage depletion in chimeric Fth-deleted mice

Hematopoietic Fth deletion was associated with the depletion of monocyte-derived macrophages (Ly6G⁻CD11b⁺F4/80low) expressing C-X-3-C Motif Chemokine Receptor 1 (CX₃CR1) and high levels of Lymphocyte Antigen 6 Complex, Locus C (LY6C) (CX₃CR1⁺LY6Chigh) in $Fth^{R26\Delta/\Delta}{\Rightarrow}Fth^{R26\Delta/\Delta}$ chimeras, as illustrated in the liver, heart, lungs and kidneys (Figs. 1E,F and EV1A). In contrast, tissue-resident CD11b⁻/lowF4/80high macrophages were not affected, with the exception of the lungs (Fig. EV1B,C). Moreover, monocyte-derived (Figs. 1E,F and EV1A) and tissue-resident macrophages (Fig. EV1B,C) were not depleted when Fth was deleted specifically in parenchyma cells from $Fth^{wt/wt}{\Rightarrow}Fth^{R26\Delta/\Delta}$ chimeras.

There was a marked reduction of the number of monocyte-derived macrophages when Fth was deleted in myeloid cells from $Fth^{LysM\Delta/\Delta}{\Rightarrow}Fth^{R26\Delta/\Delta}$ chimeras (Figs. 1E,F and EV1A). This was not associated however with a reduction of tissue-resident macrophages, with the exception of the kidneys (Fig. EV1B,C).

This suggests that when Fth is deleted in parenchyma cells, FTH expression becomes essential to support tissue-dependent differentiation, and/or viability, of CX₃CR1⁺LY6Chigh monocyte-derived macrophages (Guilliams et al, 2018; Trzebanski et al, 2024). Of note this is not observed when FTH is expressed in parenchyma cells from mice in which Fth is deleted specifically in myeloid cells (Bolisetty et al, 2015; Ikeda et al, 2020),

## Fth-competent monocyte-derived macrophages rescue chimeric Fth-deleted mice

Monocyte-derived macrophages support tissue function (Amit et al, 2016; Guilliams et al, 2018), suggesting that classical monocytes (CX₃CR1⁺LY6ChighF4/80⁻) prevent the lethal outcome of global Fth deletion in chimeric mice. To test this hypothesis, we generated $Fth^{Cx3cr1\Delta/\Delta}$ mice, carrying an Fth deletion under the control of the Cx3cr1 promoter (Jung et al, 2000). Fth-deleted classical monocytes failed to rescue the lethal outcome of Fth deletion in 50% of $Fth^{Cx3cr1\Delta/\Delta}{\Rightarrow}Fth^{R26\Delta/\Delta}$ chimeras (i.e., $Fth^{R26fl/fl}$ mice reconstituted with $Fth^{Cx3cr1\Delta/\Delta}$ BM) (Fig. EV1D). This partial lethality likely reflects some level of mosaicism in CX3CR1 expression, and therefore in CX3CR1-driven Cre expression in monocyte/macrophages (Yona et al, 2013). As such, mosaic FTH deletion may justify why not all in $Fth^{Cx3cr1\Delta/\Delta}{\Rightarrow}Fth^{R26\Delta/\Delta}$ chimeras succumb to Fth deletion. These data suggest, nevertheless, that Fth-competent classical monocytes contribute to support organismal homeostasis, when Fth is deleted in parenchyma cells.

Monocyte-derived macrophages migrate to non-lymphoid tissues via a mechanism that relies on the chemokine/chemokine receptor monocyte chemotactic protein 1 (MCP-1; CCL2) and C-C chemokine receptor type 2 (CCR2; CD192), respectively (Boring et al, 1997). To test whether the CCL2-CCR2 chemotactic axis is required to support the salutary effects of monocyte-derived macrophages, we reconstituted $Fth^{R26fl/fl}$ mice with BM from Ccr2 deficient $Ccr2^{-/-}$ mice (Boring et al, 1997). Fth deletion in $Ccr2^{-/-}{\Rightarrow}Fth^{R26\Delta/\Delta}$ chimeras was only partially lethal, compared to control $Ccr2^{+/+}{\Rightarrow}Fth^{fl/fl}$ chimeras (Fig. EV1E). This is likely explained by Ccr2 deletion not fully preventing monocyte BM egression (Tsou et al, 2007), with low numbers of monocyte-derived macrophages being sufficient to support organismal homeostasis, when Fth is deleted in parenchyma cells. As a non-mutually exclusive hypothesis, monocyte-derived macrophages might support organismal homeostasis via a mechanism that is only partially CCL2/CCR2-dependent.

## Fth-competent circulating monocytes rescue chimeric Fth-deleted mice

To test whether circulating monocytes carry the capacity to rescue the lethal outcome of systemic Fth deletion we used parabiosis, an experimental approach whereby a common circulatory system is established surgically to enable the transit of circulating cells (and soluble factors) between two mice (Harris, 2013). Global Fth deletion caused severe wasting (Fig. EV1F) and was lethal (Fig. 1G) in parabiotic $Fth^{R26\Delta/\Delta}{\leftrightarrow}Fth^{R26\Delta/\Delta}$ pairs. Expression of FTH in one parabiotic mouse restored the survival of the Fth-deleted parabiotic mouse ($Fth^{fl/fl}{\leftrightarrow}Fth^{R26\Delta/\Delta;}$ Fig. 1G), similar to control Fth-competent parabiotic pairs ($Fth^{fl/fl}{\leftrightarrow}Fth^{fl/fl}$), receiving TAM at the same dosage and schedule (Figs. 1G and EV1F). This suggests that Fth-competent hematopoietic-derived circulating cells and/or soluble factors generated in a mouse that expresses FTH are sufficient to rescue the lethal outcome of systemic Fth deletion.

The protective effect of Fth-competent circulating cells was contingent on the expression of FTH in myeloid cells, as revealed by the lethal outcome of Fth deletion in $Fth^{LysM\Delta/\Delta}{\leftrightarrow}Fth^{R26\Delta/\Delta}$ parabiotic pairs (Fig. 1G and EV1F). This suggests that circulating Fth-competent myeloid cells, which include circulating monocytes, are required to prevent the lethal outcome of systemic Fth deletion, in the absence of lethal irradiation and/or hematopoietic cell reconstitution.

## Fth-competent BM-derived monocytes (BMDM) rescue chimeric Fth-deleted mice

To prove unequivocally that Fth-competent monocytes carry the capacity to rescue the lethal outcome of global Fth deletion we

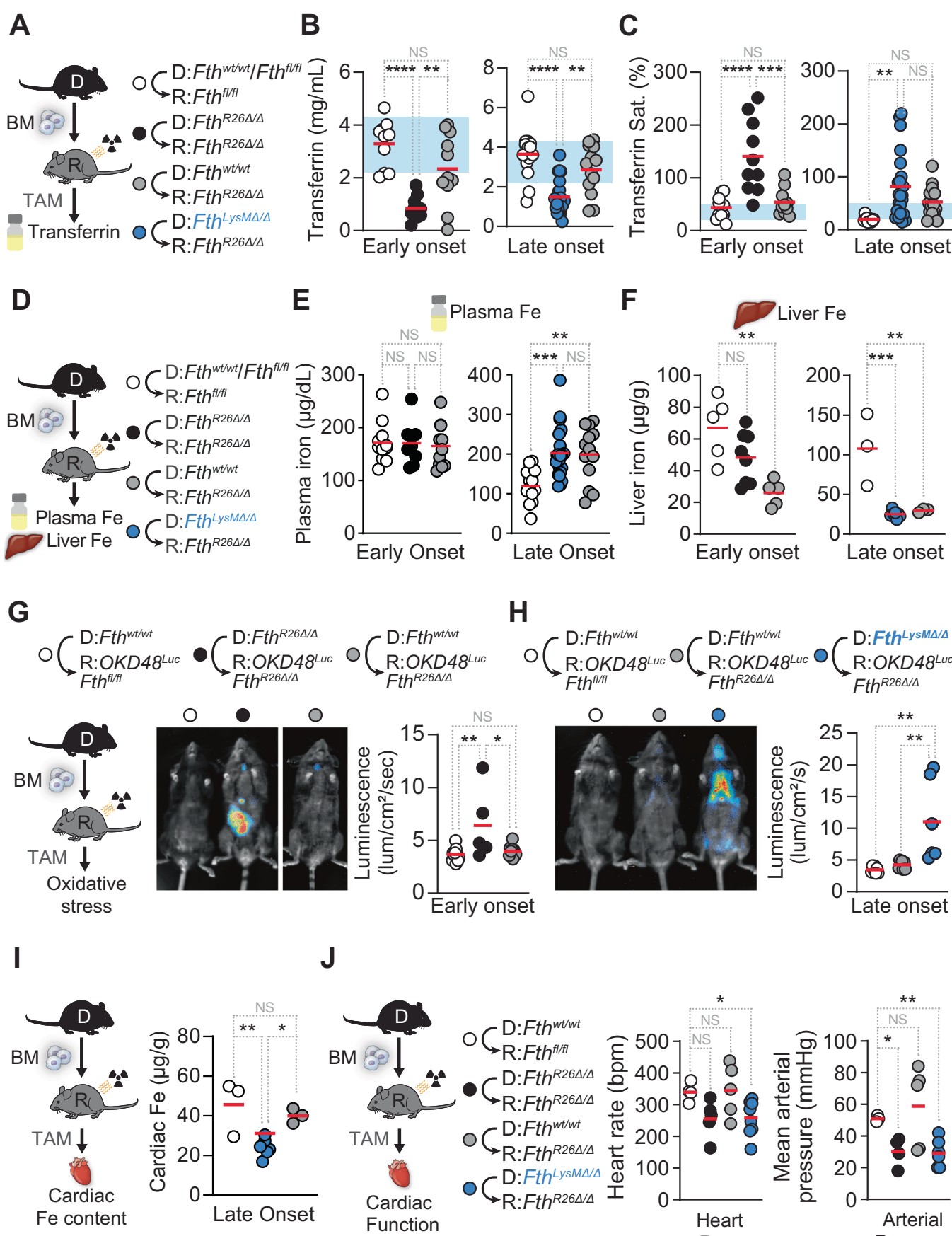

**Figure 2.** *Fth*-competent myeloid cells restore Fe homeostasis, prevent oxidative stress and restore cardiac function in chimeric *Fth*-deleted mice.

(A) Schematic representation of chimeric mice and TAM-induced *Fth* deletion (day 0). (B) Plasma transferrin concentration and (C) transferrin saturation in $Fth^{fl/fl} \Rightarrow Fth^{fl/fl}$ ($n = 10$–14), $Fth^{R26\Delta/\Delta} \Rightarrow Fth^{R26\Delta/\Delta}$ ($n = 10$–11), $Fth^{fl/fl} \Rightarrow Fth^{R26\Delta/\Delta}$ ($n = 11$–13) and $Fth^{LysM\Delta/\Delta} \Rightarrow Fth^{R26\Delta/\Delta}$ ($n = 23$) chimeric mice on days 7–15 (early onset), or 19–35 (late onset) following TAM administration. Data in (B, C) pooled from 6 independent experiments with similar trends. (D) Schematic representation of chimeric mice and TAM-induced *Fth* deletion. Iron levels in (E) plasma measured in $Fth^{fl/fl} \Rightarrow Fth^{fl/fl}$ ($n = 11$–13), $Fth^{R26\Delta/\Delta} \Rightarrow Fth^{R26\Delta/\Delta}$ ($n = 9$), $Fth^{fl/fl} \Rightarrow Fth^{R26\Delta/\Delta}$ ($n = 11$–13) and $Fth^{LysM\Delta/\Delta} \Rightarrow Fth^{R26\Delta/\Delta}$ ($n = 23$) chimeric mice on days 7–15 (early onset), or 19–35 (late onset) following TAM administration and (F) liver measured in $Fth^{fl/fl} \Rightarrow Fth^{fl/fl}$ ($n = 3$–5), $Fth^{R26\Delta/\Delta} \Rightarrow Fth^{R26\Delta/\Delta}$ ($n = 8$), $Fth^{fl/fl} \Rightarrow Fth^{R26\Delta/\Delta}$ ($n = 3$–5) and $Fth^{LysM\Delta/\Delta} \Rightarrow Fth^{R26\Delta/\Delta}$ ($n = 7$) chimeric mice on day 7 (early onset), or 22 (late onset) following TAM administration. Data in (E, F) pooled from three independent experiments with similar trends. (G) Schematic representation of TAM-induced *Fth* deletion in chimeric mice (day 0). Representative luminescence images depicting oxidative stress, as monitored by OKD48$^{Luc}$ reporter luciferase luminescence, in $Fth^{fl/fl} \Rightarrow OKD48^{Luc}Fth^{fl/fl}$ ($n = 7$–11), $Fth^{R26\Delta/\Delta} \Rightarrow OKD48^{Luc}Fth^{R26\Delta/\Delta}$ ($n = 5$), $Fth^{fl/fl} \Rightarrow OKD48^{Luc}Fth^{R26\Delta/\Delta}$ ($n = 7$–9) and $Fth^{LysM\Delta/\Delta} \Rightarrow OKD48^{Luc}Fth^{R26\Delta/\Delta}$ ($n = 6$) chimeric mice on days 7 (G; early onset) or 19 (H; late onset) following TAM administration. Data in (G, H) presented as individual values (circles) and mean (red bars), pooled from six independent experiments with similar trends. (I) Schematic representation of chimeric mice and TAM-induced *Fth* deletion (day 0) and cardiac iron content (right) in $Fth^{fl/fl} \Rightarrow Fth^{fl/fl}$ ($n = 3$–5), $Fth^{fl/fl} \Rightarrow Fth^{R26\Delta/\Delta}$ ($n = 3$–5) and $Fth^{LysM\Delta/\Delta} \Rightarrow Fth^{R26\Delta/\Delta}$ ($n = 6$–7). (J) Schematic representation of chimeric mice and TAM-induced *Fth* deletion (day 0) and cardiac function parameters (heart rate; mean arterial pressure) in $Fth^{fl/fl} \Rightarrow Fth^{fl/fl}$ ($n = 4$), $Fth^{R26\Delta/\Delta} \Rightarrow Fth^{R26\Delta/\Delta}$ ($n = 5$), $Fth^{fl/fl} \Rightarrow Fth^{R26\Delta/\Delta}$ ($n = 5$) and $Fth^{LysM\Delta/\Delta} \Rightarrow Fth^{R26\Delta/\Delta}$ ($n = 8$) chimeric mice between days 10 and 51 following TAM administration. Data in (J) presented as individual values (circles) and mean (red bars), pooled from two independent experiments with similar trends. Data in (B, C, E, F–I) represented as individual values (circles) and mean (red bars), one-way ANOVA with Tukey's range test for multiple comparison correction was used for comparison between multiple groups. Data in (J) represented as individual values (circles) and mean (red bars), Welch ANOVA with Dunnett's T3 test for multiple comparison correction was used for comparison between control and test groups NS: non-significant, $*P < 0.05$, $**P < 0.01$, $***P < 0.001$, $****P < 0.0001$. Source data are available online for this figure.

generated BM-derived monocytes (BMDM) in vitro and tested their protective effect upon adoptive transfer into adult *Fth*-deleted $Fth^{R26\Delta/\Delta}$ mice. To trace BMDM, we added a *Rosa26*-tandem dimer (td) Tomato-Flox-stop-Flox allele driving the expression of td Tomato (tdT) by Cre-driven excision, under the control of *Rosa26* promoter ($tdT^{R26}$ mice). Adoptive transfer of *Fth*-competent tdT$^+$ BMDM (generated from $tdT^{R26}$ mice) was sufficient to extend the lifespan of *Fth*-deleted $Fth^{R26\Delta/\Delta}$ mice (Fig. 1H,I). In contrast, Adoptive transfer of *Fth*-deleted tdT$^+$ BMDM (generated from $tdT^{R26}Fth^{R26\Delta/\Delta}$ mice) failed to extend the lifespan of *Fth*-deleted $Fth^{R26\Delta/\Delta}$ mice (Fig. 1H,I). All recipient *Fth*-deleted $Fth^{R26\Delta/\Delta}$ mice succumbed within 1–2 days after the last BMDM adoptive transfer, suggesting that long-term survival of $Fth^{R26\Delta/\Delta}$ mice is dependent on continuous supply of circulating monocytes, presumably via myelopoiesis.

## FTH supports the capacity of BMDM to give rise to tissue macrophages

The number of *Fth*-deleted tdT$^+$ cells recovered from the liver and spleen of recipient *Fth*-deleted ($Fth^{R26\Delta/\Delta}$) mice, 24 h after adoptive transfer of tdT$^+$ BMDM, was markedly lower, as compared to *Fth*-competent tdT$^+$ cells (Fig. 1H,J). This indicates that *Fth*-deleted BMDM have an impaired capacity to migrate and/or differentiate into tissue-resident macrophages. Moreover, recovery of FTH-competent tdT$^+$ BMDM from the liver, but not the spleen, of recipient FTH-deleted ($Fth^{R26\Delta/\Delta}$) was higher than in control ($Fth^{fl/fl}$) recipient mice (Fig. 1H,J). This suggests that FTH-deficient organs (e.g., liver) have an increased capacity to recruit, and/or retain, monocyte-derived macrophages upon the stress imposed by FTH deletion.

To gain further insight into the differentiation of tdT$^+$ BMDM after adoptive transfer, we performed a t-SNE (t-distributed Stochastic Neighbor Embedding) analysis based on CX3CR1, LY6C, F4/80 and CD11B expression, prior to, and 24 h after BMDM transfer (Fig. 1H,K). The t-SNE profile of tdT$^+$ BMDM before and after adoptive transfer was distinct, as assessed in liver of recipient mice (Fig. 1H,K,L). There was heterogeneity in CX3CR1, LY6C, F4/80 and CD11B expression with a sub-population of cells increasing Ly6C and decreasing CX3CR1 and

F4/80 expression, while maintaining CD11b and tdT expression (Fig. 1H,K,L). This suggests that upon adoptive transfer, tdT$^+$ BMDM differentiate into distinct tissue-specific monocyte-derived macrophage subpopulations.

*Fth*-deleted and *Fth*-competent BMDM tdT$^+$ cells from the liver of recipient *Fth*-deleted ($Fth^{R26\Delta/\Delta}$) mice presented similar t-SNE profiles (Fig. 1H,K,L). There was also no discernable difference in the t-SNE profiles of *Fth*-competent tdT$^+$ cells from the liver of *Fth*-deleted ($Fth^{R26\Delta/\Delta}$) vs. control ($Fth^{fl/fl}$) mice (Fig. 1H,K,L). This suggests that FTH does not regulate the initial capacity of BMDM to differentiate into monocyte-derived tissue macrophages.

## *Fth*-competent myeloid cells restore Fe homeostasis in chimeric *Fth*-deleted mice

The lethal outcome of global *Fth* deletion was associated with systemic dysregulation of Fe metabolism, as illustrated by reduced circulating transferrin and higher transferrin saturation in *Fth*-deleted $Fth^{R26\Delta/\Delta} \Rightarrow Fth^{R26\Delta/\Delta}$ vs. *Fth*-competent $Fth^{wt/wt} \Rightarrow Fth^{fl/fl}$ chimeras (Fig. 2A–C). *Fth*-competent hematopoietic cells restored the levels of circulating transferrin and normalized transferrin saturation (Fig. 2A–C) in plasma from $Fth^{fl/fl} \Rightarrow Fth^{R26\Delta/\Delta}$ chimeras to levels in the range of those in control *Fth* competent $Fth^{fl/fl} \Rightarrow Fth^{fl/fl}$ chimeras (Fig. 2A–C). This was not the case however, when *Fth* was deleted in myeloid cells from $Fth^{LysM\Delta/\Delta} \Rightarrow Fth^{R26\Delta/\Delta}$ chimeras, which presented a decrease in the levels of circulating transferrin and an increase in transferrin saturation (Fig. 2A–C), as compared to *Fth* competent $Fth^{fl/fl} \Rightarrow Fth^{fl/fl}$ chimeras (Fig. 2C). This was associated, over time, with an increase in the concentration of circulating Fe in both $Fth^{LysM\Delta/\Delta} \Rightarrow Fth^{R26\Delta/\Delta}$ and $Fth^{fl/fl} \Rightarrow Fth^{R26\Delta/\Delta}$ chimeras (Fig. 2D,E), suggesting that, over time, *Fth*-competent monocyte-derived macrophages do not prevent the accumulation of circulating redox-active Fe imposed by global *Fth* deletion, and that this increase in circulatory Fe is independent from *Fth* deletion-induced lethality.

Systemic *Fth* deletion was associated with a decrease of hepatic Fe content, not restored by *Fth*-competent hematopoietic cells in $Fth^{fl/fl} \Rightarrow Fth^{R26\Delta/\Delta}$ nor $Fth^{LysM\Delta/\Delta} \Rightarrow Fth^{R26\Delta/\Delta}$ chimeras, compared to control *Fth*-competent $Fth^{wt/wt} \Rightarrow Fth^{fl/fl}$ chimeras (Fig. 2F). Moreover, systemic *Fth* deletion was associated with the appearance of electron-dense crystalline inclusions in both the liver (Appendix

Fig. S2A) and WAT (Appendix Fig. S2B), from *Fth*-deleted $Fth^{R26\Delta/\Delta}$$\Rightarrow$$Fth^{R26\Delta/\Delta}$ and $Fth^{LysM\Delta/\Delta}$$\Rightarrow$$Fth^{R26\Delta/\Delta}$ chimeras. This is consistent with the formation of hemosiderin Fe deposits formed by a single membrane-bound lysosomal body, likely due to the loss of iron storage capacity imposed by *Fth* deletion (Fig. 2F). These siderosome-like structures were not observed in the liver (Appendix Fig. S2A) nor the WAT (Appendix Fig. S2B) from $Fth^{fl/fl}$$\Rightarrow$$Fth^{R26\Delta/\Delta}$ chimeras, similar to control $Fth^{fl/fl}$$\Rightarrow$$Fth^{fl/fl}$ chimeras. These observations suggest that *Fth*-competent monocyte-derived macrophages prevent the formation of hemosiderin Fe deposits associated with systemic loss of Fe storage capacity due to *Fth* deletion.

## *Fth*-competent myeloid cells restore redox homeostasis in chimeric *Fth*-deleted mice

Using $OKD48^{Luc}$ reporter mice, expressing a luciferase reporter ubiquitously to monitor oxidative stress in vivo (Oikawa et al, 2012), we confirmed that global *Fth* deletion led to systemic oxidative stress in $Fth^{R26\Delta/\Delta}$$\Rightarrow$$OKD48^{Luc}Fth^{R26\Delta/\Delta}$ chimeras (i.e., $OKD48^{Luc}Fth^{R26fl/fl}$ mice reconstituted with BM from $Fth^{R26fl/fl}$ mice) (Fig. 2G), consistent with described in adult $Fth^{R26\Delta/\Delta}$ mice (Blankenhaus et al, 2019). *Fth*-competent hematopoietic cells restored systemic redox balance in $Fth^{fl/fl}$$\Rightarrow$$OKD48^{Luc}Fth^{R26\Delta/\Delta}$ chimeras (Fig. 2G,H). This was no longer the case however, when *Fth* was deleted in myeloid cells (Fig. 2H), as assessed at a later point (i.e., late onset) in $Fth^{LysM\Delta/\Delta}$$\Rightarrow$$OKD48^{Luc}Fth^{R26\Delta/\Delta}$ chimeras (Fig. 2H). This suggests that *Fth*-competent monocyte-derived macrophages are essential to prevent the development of tissue oxidative stress imposed by systemic *Fth* deletion.

## *Fth*-competent myeloid cells support tissue function in chimeric *Fth*-deleted mice

Global *Fth* deletion was associated with the development of multiorgan damage, as illustrated serologically by the accumulation of alanine aminotransferase (ALT; liver damage) (Fig. EV2A,B), aspartate aminotransferase (AST; liver damage) (Fig. EV2A,C), lactate dehydrogenase (LDH) (Fig. EV2A,D), creatinine phosphokinase (CPK; muscle damage) (Fig. EV2A,E) and urea (kidney damage) (Fig. EV2A,F) in the plasma of $Fth^{R26\Delta/\Delta}$$\Rightarrow$$Fth^{R26\Delta/\Delta}$ chimeras *vs.* control *Fth*-competent $Fth^{fl/f}/Fth^{wt/wt}$$\Rightarrow$$Fth^{fl/fl}$ chimeras. Multiorgan damage was confirmed histologically, as illustrated in the liver (i.e., hepatocellular vacuolar degeneration) and in the kidneys (i.e., acute tubular cell necrosis) (Fig. EV2A,G,H).

*Fth*-competent hematopoietic cells prevented multiorgan damage in $Fth^{fl/fl}$$\Rightarrow$$Fth^{R26\Delta}$ chimeras (Fig. EV2A,G). This was no longer observed however, when *Fth* was deleted in myeloid cells from $Fth^{LysM\Delta/\Delta}$$\Rightarrow$$Fth^{R26\Delta/\Delta}$ chimeras, which showed hepatocellular hypertrophic lesions with granular acidophilic cytoplasm, hinting at possible liver mitochondrial dysfunction (Fig. EV2A,H). This suggests that *Fth*-competent monocyte-derived macrophages are essential to prevent multiorgan damage imposed by systemic *Fth* deletion.

## *Fth*-deleted mice succumb to life-threatening cardiac dysfunction

FTH is highly expressed in the heart (Munro and Linder, 1978) and *Fth* deletion in cardiac myocytes leads to cardiac dysfunction (Fang et al, 2020), suggesting that $Fth^{R26\Delta/\Delta}$ mice develop cardiac dysfunction. However, global *Fth* deletion was not associated with the development of acute cardiac damage, as illustrated by troponin I accumulation in plasma from $Fth^{R26\Delta/\Delta}$$\Rightarrow$$Fth^{R26\Delta/\Delta}$ *vs.* control $Fth^{fl/fl}$$\Rightarrow$$Fth^{fl/fl}$ chimeras (Fig. EV2I) and confirmed histologically (Fig. EV2J). Nevertheless, global *Fth* deletion in $Fth^{R26\Delta/\Delta}$ mice led to a profound reduction of heart rate, end-systolic pressure (ESP) and preload recruitable stroke work (PRSW), compared to control $R26^{Cre}$ or $Fth^{fl/fl}$ mice (Appendix Fig. S3A–C). A number of other cardiac function parameters, including cardiac mechanical power (MaxPwr) with ensuing reduction of mean arterial pressure, cardiac output, stroke work, pressure-volume area and peak systolic pressure (Pmax) were also severely impaired in $Fth^{R26\Delta/\Delta}$ *vs.* control $R26^{Cre}$ or $Fth^{fl/fl}$ mice (Appendix Fig. S3A–C). This suggests that FTH is essential to sustain heart function and blood circulation.

## *Fth*-competent myeloid cells support cardiac function in chimeric *Fth*-deleted mice

Systemic *Fth* deletion did not impact on cardiac Fe content in $Fth^{R26\Delta/\Delta}$$\Rightarrow$$Fth^{R26\Delta/\Delta}$ *vs.* control $Fth^{fl/fl}$$\Rightarrow$$Fth^{fl/fl}$ chimeras (Appendix Fig. S3D). In contrast however, myeloid *Fth* deletion led to a marked decrease in cardiac Fe content in $Fth^{LysM\Delta/\Delta}$$\Rightarrow$$Fth^{R26\Delta/\Delta}$ chimeras, at a later time point, compared to control $Fth^{fl/fl}$$\Rightarrow$$Fth^{fl/fl}$ chimeras (Fig. 2I). This was prevented by *Fth* competent hematopoietic cells in $Fth^{fl/fl}$$\Rightarrow$$Fth^{R26\Delta/\Delta}$ chimeras (Fig. 2I), suggesting that over time, *Fth*-competent myeloid cells become essential to control the cardiac Fe content of chimeric mice in which *Fth* is deleted globally.

We then asked whether macrophages prevent the development of life-threatening cardiac dysfunction imposed by systemic *Fth* deletion. Similar to observed upon *Fth* deletion in adult $Fth^{R26\Delta/\Delta}$ mice (Appendix Fig. S3A–C), global *Fth* deletion in $Fth^{R26\Delta/\Delta}$$\Rightarrow$$Fth^{R26\Delta/\Delta}$ chimeric mice led to the development of life-threatening cardiac dysfunction, revealed by a profound reduction of heart rate and mean arterial pressure (Fig. 2J) as well as ESP, PRSW, and MaxPwr (Appendix Fig. S3E), compared to control $Fth^{fl/f}/Fth^{wt/wt}$$\Rightarrow$$Fth^{fl/fl}$ chimeras (Fig. 2J; Appendix Fig. S3E). *Fth*-competent hematopoietic cells prevented the development of life-threatening cardiac dysfunction in *Fth*-deleted $Fth^{fl/fl}$$\Rightarrow$$Fth^{R26\Delta}$ chimeras (Fig. 2J; Appendix Fig. S3E). This suggests that *Fth*-competent hematopoietic cells carry the capacity to prevent the development of life-threatening cardiac dysfunction imposed by systemic *Fth* deletion.

*Fth*-deletion in the myeloid compartment of $Fth^{LysM\Delta/\Delta}$$\Rightarrow$$Fth^{R26\Delta/\Delta}$ chimeras was associated with the development of life-threatening cardiac dysfunction (Fig. 2J; Appendix Fig. S3E). This suggests that *Fth*-competent myeloid cells, including monocyte-derived macrophages, are essential to prevent the development of life-threatening cardiac dysfunction imposed by systemic *Fth* deletion. This is consistent with monocyte-derived macrophages promoting cardiac function under steady state conditions (Hulsmans et al, 2017) or in response to ischemic damage (Swirski et al, 2009).

## *Fth*-competent myeloid cells support energy metabolism in chimeric *Fth*-deleted mice

Cardiac function is a highly energy demanding process, entertaining the hypothesis that *Fth*-competent monocyte-derived macrophages

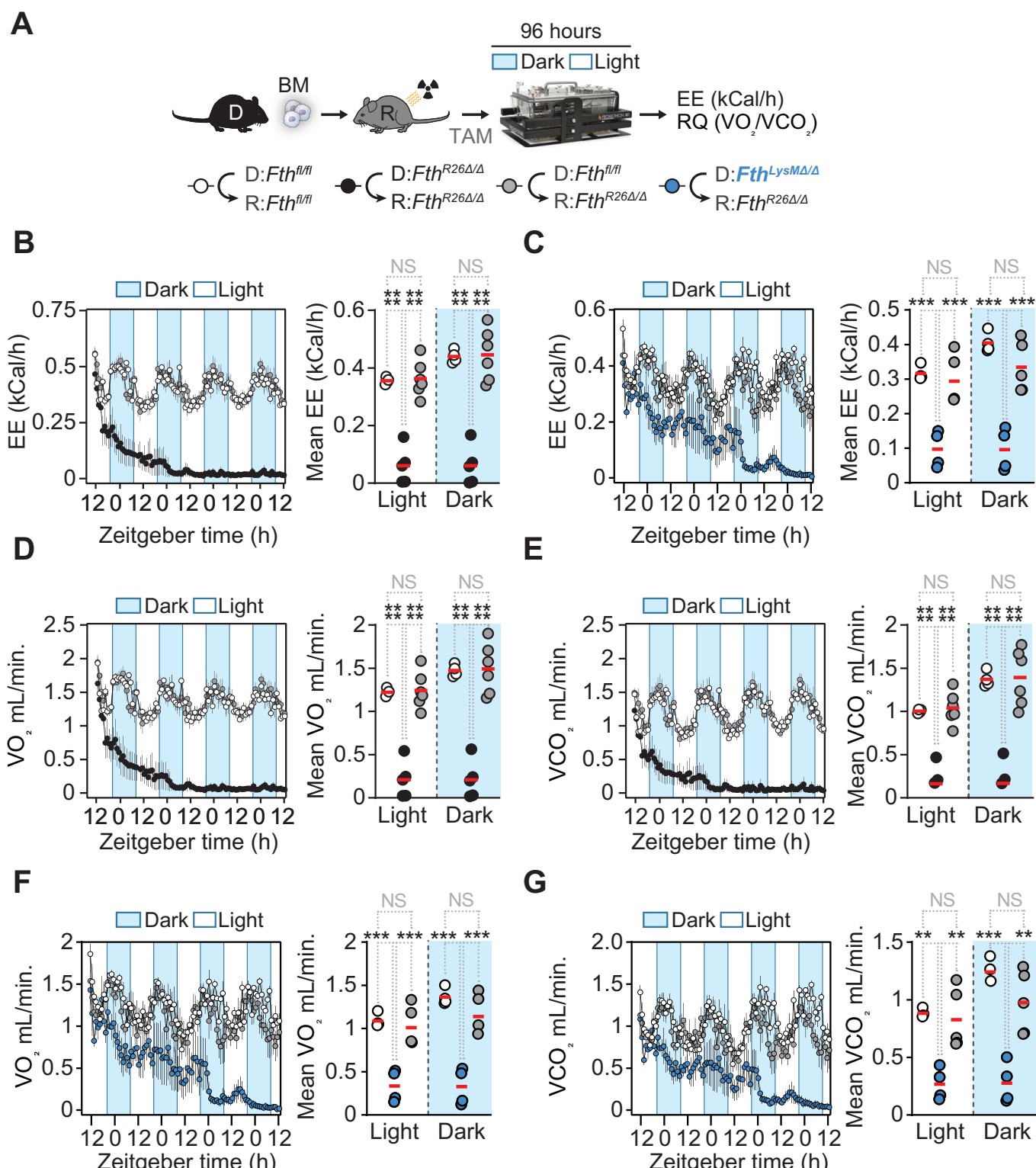

support cardiac function, indirectly, via a mechanism that sustains organismal energy expenditure. In support of this hypothesis, global *Fth* deletion led to a collapse of energy expenditure (EE) in $Fth^{R26\Delta/\Delta}{\Rightarrow}Fth^{R26\Delta/\Delta}$ chimeras (Fig. 3A,B), similar to observed $Fth^{R26\Delta/\Delta}$ mice (Blankenhaus et al, 2019). *Fth*-competent hematopoietic cells rescued

EE in $Fth^{fl/fl}{\Rightarrow}Fth^{R26\Delta/\Delta}$ chimeras (Fig. 3A,B) while *Fth*-deletion in myeloid cells compromised the capacity of hematopoietic cells to rescue EE in $Fth^{LysM\Delta/\Delta}{\Rightarrow}Fth^{R26\Delta/\Delta}$ chimeras (Fig. 3A,C).

Systemic *Fth* deletion reduced consumed O$_2$ (VO$_2$) (Fig. 3D) and exhaled CO$_2$ (VCO$_2$) to the same extent (Fig. 3E) and therefore did not

**Figure 3. *Fth*-competent myeloid cells support energy metabolism in chimeric *Fth*-deleted mice.**

(A) Schematic representation of TAM-induced *Fth* deletion in chimeric mice (day 0). (B, C) Time course and mean of energy expenditure (EE) during day/night time in *Fth*$^{fl/fl}$⇒*Fth*$^{fl/fl}$ (n = 4), *Fth*$^{R26Δ/Δ}$⇒*Fth*$^{R26Δ/Δ}$ (n = 5), *Fth*$^{fl/fl}$⇒*Fth*$^{R26Δ/Δ}$ (n = 5–6) and *Fth*$^{LysMΔ/Δ}$⇒*Fth*$^{R26Δ/Δ}$ (n = 4) chimeric mice, assessed from day 7 (B; early onset), or day 20 (C; late onset) post-TAM administration. (D, E) Time course and mean of (D) O$_2$ consumption rate (VO$_2$) and (E) CO$_2$ production rate (VCO$_2$) during day/nighttime in *Fth*$^{fl/fl}$⇒*Fth*$^{fl/fl}$ (n = 4), *Fth*$^{R26Δ/Δ}$⇒*Fth*$^{R26Δ/Δ}$ (n = 5) and *Fth*$^{fl/fl}$⇒*Fth*$^{R26Δ/Δ}$ (n = 5–6) chimeric mice, assessed from day 7 (early onset). Time course and mean of (F) O$_2$ consumption rate (VO$_2$) and (G) CO$_2$ production rate (VCO$_2$) during day/nighttime in *Fth*$^{fl/fl}$⇒*Fth*$^{fl/fl}$ (n = 4), *Fth*$^{LysMΔ/Δ}$⇒*Fth*$^{R26Δ/Δ}$ (n = 4) and *Fth*$^{fl/fl}$⇒*Fth*$^{R26Δ/Δ}$ (n = 5–6) chimeric mice, assessed from day 20 (late onset). Data in (B–G) is displayed as mean ± SD (time course), or as individual values (circles) and mean (red bars) (dot plots). Data in (B–G) is pooled from two independent experiments with similar trends. One-way ANOVA with Tukey's range test for multiple comparison correction was used for comparison between multiple groups. NS: non-significant, **$P < 0.01$, ***$P < 0.001$, ****$P < 0.0001$. Source data are available online for this figure.

reflect on respiratory quotient (RQ), (Appendix Fig. S2A–C). *Fth*-competent hematopoietic cells restored VO$_2$ and VCO$_2$ in *Fth*$^{fl/fl}$⇒*Fth*$^{R26Δ/Δ}$ chimeras (Fig. 3D,E) while *Fth*-deleted myeloid cells failed to do so in *Fth*$^{LysMΔ/Δ}$⇒*Fth*$^{R26Δ/Δ}$ chimeras (Fig. 3F,G).

Global *Fth* deletion in chimeric mice was associated with the suppression of locomotor activity (Fig. EV3D) and food intake (Fig. EV3E). *Fth*-competent hematopoietic cells restored locomotor activity (Fig. EV3D) and food intake (Fig. EV3E) in *Fth*$^{fl/fl}$⇒*Fth*$^{R26Δ/Δ}$ chimeras (Fig. EV3D,E). This was no longer observed upon *Fth*-deletion in myeloid cells (Fig. EV3F,G).

These observations suggest that *Fth*-competent monocyte-derived macrophages are essential to restore energy metabolism under *Fth* deletion in parenchyma cells. This salutary effect is associated with a regain of locomotor activity and food intake (Fig. EV3F,G).

### *Fth*-competent myeloid cells support BAT thermogenesis in chimeric *Fth*-deleted mice

Mice allocate up to one-third of their EE to support core body temperature, under standard husbandry conditions (~ 22 °C) (Ganeshan and Chawla, 2017; Ganeshan et al, 2019; Reitman, 2018). This is achieved, to a large extent, via brown adipose tissue (BAT) thermogenesis (Cannon and Nedergaard, 2004; Lowell et al, 1993), entertaining the hypothesis that *Fth*-competent monocyte-derived macrophages sustain BAT thermogenesis in chimeric mice. In support of this hypothesis, global *Fth* deletion compromised BAT thermogenesis (Fig. 4A,B) and caused BAT wasting (Fig. EV4A,B) in *Fth*$^{R26Δ/Δ}$⇒*Fth*$^{R26Δ/Δ}$ chimeras. *Fth*-competent hematopoietic cells restored BAT thermogenesis in *Fth*$^{fl/fl}$⇒*Fth*$^{R26Δ/Δ}$ chimeras (Fig. 4A,B): This was no longer the case when *Fth* was deleted in myeloid cells (Fig. 4C,D), with a tendency for BAT wasting and reduction in average lipid droplet at a later time point (Fig. EV4C,D), albeit not significant. This suggests that *Fth*-competent myeloid cells, including monocyte-derived macrophages, prevent the collapse of BAT thermogenesis and thermoregulation imposed by systemic *Fth* deletion.

### *Fth*-competent myeloid cells support WAT function in chimeric *Fth*-deleted mice

BAT thermogenesis is fueled via white adipose tissue (WAT) lipolysis (Heine et al, 2018), entertaining the hypothesis that *Fth*-competent monocyte-derived macrophages regulate WAT lipolysis. In keeping with this hypothesis, global *Fth* deletion led to visceral (i.e., gonadal) (Fig. 4E,F) and subcutaneous (i.e., inguinal) WAT wasting (Fig. EV4E,F), with reduction of lipid droplet (i.e.,

adipocyte) size in *Fth*$^{R26Δ/Δ}$⇒*Fth*$^{R26Δ/Δ}$ chimeras (Figs. 4E,F and EV4E,F). While *Fth*-competent hematopoietic cells prevented WAT wasting in *Fth*$^{fl/fl}$⇒*Fth*$^{R26Δ/Δ}$ chimeras (Figs. 4E,F and EV4E,F), WAT wasting was prominent when *Fth* was deleted in myeloid cells from *Fth*$^{LysMΔ/Δ}$⇒*Fth*$^{R26Δ/Δ}$ chimeras (Figs. 4G,H and EV4G,H). This suggests that *Fth*-competent myeloid cells that give rise to monocyte-derived macrophages, carry the capacity to normalize WAT function and BAT thermogenesis in chimeric mice in which *Fth* is deleted globally.

### *Fth*-competent myeloid cells partially restore ferritin expression in tissues from chimeric *Fth*-deleted mice

Macrophages secrete and transfer ferritin (Cohen et al, 2010; Meyron-Holtz et al, 2011), suggesting that FTH might be transferred from *Fth*-competent monocyte-derived macrophages to *Fth*-deficient bystander parenchyma cells. In keeping with this hypothesis, *Fth*-competent hematopoietic cells contribute with 30-40% of the relative level of FTH protein expression in heart, liver, lung and kidneys of *Fth*$^{fl/fl}$⇒*Fth*$^{R26Δ/Δ}$ chimeras, compared to control *Fth*$^{fl/fl}$⇒*Fth*$^{fl/fl}$ chimeras (Fig. 5A,B). In contrast, FTH protein expression was barely detectable when *Fth* was deleted in myeloid cells from *Fth*$^{LysMΔ/Δ}$⇒*Fth*$^{R26Δ/Δ}$ chimeras (Fig. 5A,B). This suggests that under global *Fth* deletion, *Fth*-competent monocyte-derived macrophages are essential to partially restore the level of FTH protein expression in different organs.

### *Fth*-competent myeloid cells do not deliver ferritin to parenchyma cells from chimeric *Fth*-deleted mice

To determine whether or not monocyte-derived macrophages transfer FTH to parenchyma cells we used a CRISPR/Cas 9 based approach (Jinek et al, 2012) to tag the N-terminus of FTH with a 14 amino-acid V5 peptide (*Fth*$^{V5/WT}$ mouse) (Fig. 5C). Expression of V5-tagged FTH protein was validated in whole heart and kidney extracts from hemizygous *Fth*$^{V5/WT}$ mice, by western blot (Fig. 5D).

Hematopoietic cells expressing V5-tagged FTH retained the capacity to rescue the lethal outcome of systemic *Fth* deletion in *Fth*$^{V5/V5}$⇒*Fth*$^{R26Δ/Δ}$ chimeras (i.e., *Fth*$^{R26fl/fl}$ mice reconstituted with *Fth*$^{V5/V5}$ BM) (Fig. 5E). However, the expression of the V5-tagged FTH protein was restricted to CD68$^+$ macrophages, without detectable transfer to parenchyma cells, as assessed by confocal microscopy in the liver of *Fth*$^{V5/V5}$⇒*Fth*$^{R26Δ/Δ}$ chimeras (Fig. 5F,G). This suggests that *Fth*-competent monocyte-derived macrophages rescue the lethal outcome of systemic *Fth* deletion, irrespective of ferritin transfer to *Fth*-deleted parenchyma cells.

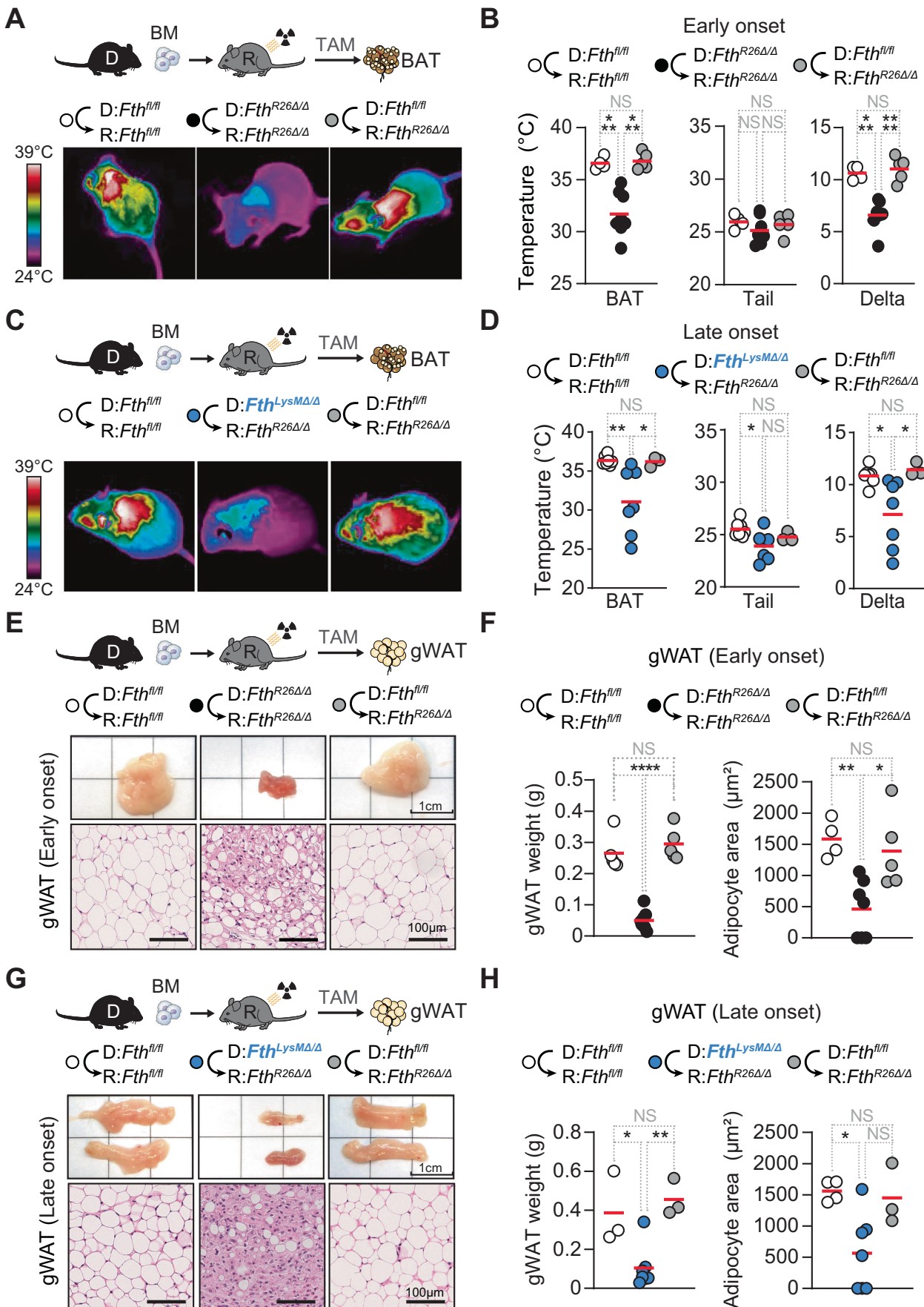

**Figure 4. *Fth*-competent myeloid cells support adipose tissue function in chimeric *Fth*-deleted mice.**

(A) Schematic representation of chimeric mice and TAM-induced *Fth* deletion (day 0) with corresponding representative infrared thermal images (FLIR) of $Fth^{fl/fl} \Rightarrow Fth^{fl/fl}$, $Fth^{R26\Delta/\Delta} \Rightarrow Fth^{R26\Delta/\Delta}$ and $Fth^{fl/fl} \Rightarrow Fth^{R26\Delta/\Delta}$ chimeric mice, on day 7 (early onset) post-TAM administration. (B) BAT (left) and tail (center) temperatures extracted from thermal images, and temperature delta (right; core body temperature – tail temperature) of $Fth^{fl/fl} \Rightarrow Fth^{fl/fl}$ ($n = 4$), $Fth^{R26\Delta/\Delta} \Rightarrow Fth^{R26\Delta/\Delta}$ ($n = 8$) and $Fth^{fl/fl} \Rightarrow Fth^{R26\Delta/\Delta}$ ($n = 5$) chimeric mice, collected on day 7 (early onset) post-TAM administration. (C) Schematic representation of chimeric mice and TAM-induced *Fth* deletion (day 0) with corresponding representative (FLIR) of $Fth^{fl/fl} \Rightarrow Fth^{fl/fl}$, $Fth^{LysM\Delta/\Delta} \Rightarrow Fth^{R26\Delta/\Delta}$ and $Fth^{fl/fl} \Rightarrow Fth^{R26\Delta/\Delta}$ chimeric mice, collected between days 16-39 (late onset) post TAM administration using an infrared thermal imaging camera (FLIR). (D) BAT (left) and tail (center) temperatures extracted from thermal images, and temperature delta (right; core body temperature – tail temperature) of $Fth^{fl/fl} \Rightarrow Fth^{fl/fl}$ ($n = 7$), $Fth^{LysM\Delta/\Delta} \Rightarrow Fth^{R26\Delta/\Delta}$ ($n = 7$) and $Fth^{fl/fl} \Rightarrow Fth^{R26\Delta/\Delta}$ ($n = 3$) chimeric mice, collected between days 16-39 (late onset) post TAM administration. Data in (B, D) represented as individual values (circles) and mean (red bars). Data in (A–D) is pooled from three independent experiments with similar trends. (E) Schematic representation of chimeric mice and TAM-induced *Fth* deletion, with corresponding representative macroscopic and histological images of gonadal white adipose tissue pads (gWAT) in $Fth^{fl/fl} \Rightarrow Fth^{fl/fl}$, $Fth^{R26\Delta/\Delta} \Rightarrow Fth^{R26\Delta/\Delta}$ and $Fth^{fl/fl} \Rightarrow Fth^{R26\Delta/\Delta}$ chimeric mice, collected on day 7 (early onset) post TAM administration. (F) gWAT weight (left) and mean (red bars) adipocyte area (right) in $Fth^{fl/fl} \Rightarrow Fth^{fl/fl}$ ($n = 4$–5), $Fth^{R26\Delta/\Delta} \Rightarrow Fth^{R26\Delta/\Delta}$ ($n = 7$–8) and $Fth^{fl/fl} \Rightarrow Fth^{R26\Delta/\Delta}$ ($n = 5$) chimeric mice, collected on day 7 (early onset) post TAM administration. (G) Schematic representation of chimeric mice and TAM-induced *Fth* deletion, with corresponding representative macroscopic and histological images of gonadal white adipose tissue pads (gWAT) in $Fth^{fl/fl} \Rightarrow Fth^{fl/fl}$, $Fth^{LysM\Delta/\Delta} \Rightarrow Fth^{R26\Delta/\Delta}$ and $Fth^{fl/fl} \Rightarrow Fth^{R26\Delta/\Delta}$ chimeric mice, collected between days 16-39 (late onset) post-TAM administration. (H) gWAT pad weight (left) and mean (red bars) adipocyte area (right) of $Fth^{fl/fl} \Rightarrow Fth^{fl/fl}$ ($n = 3$–4), $Fth^{LysM\Delta/\Delta} \Rightarrow Fth^{R26\Delta/\Delta}$ ($n = 7$) and $Fth^{fl/fl} \Rightarrow Fth^{R26\Delta/\Delta}$ ($n = 3$) chimeric mice, collected between days 16 and 39 (late onset) post-TAM administration. Data in (F, H) represented as individual values (circles) and mean (red bars). Data in (E–H) is pooled from three independent experiments with similar trends. One-way ANOVA with Tukey's range test for multiple comparison correction was used for comparison between multiple groups. NS: non-significant, *$P < 0.05$, **$P < 0.01$, ***$P < 0.001$, ****$P < 0.0001$. Source data are available online for this figure.

## *Fth*-competent myeloid cells rescue chimeric *Fth*-deleted mice irrespective of cellular Fe import/export

We hypothesized that *Fth*-competent macrophages rescue the lethal outcome of global *Fth* deletion via a mechanism that involves cellular Fe import and/or Fe export to or from *Fth*-deleted parenchyma cells, respectively. To test these hypothesis, we reconstituted $Fth^{R26fl/fl}$ mice with BM cells from mice harboring a myeloid-specific deletion of the main cellular Fe importer transferrin receptor 1 (TfR1; encoded by *Tfrc*) (Fig. EV5A) or the cellular Fe exporter Ferroportin (FPN1; encoded by *Slc40a1*) (Fig. EV5B) (Wu et al, 2023). *Slc40a1*-deleted myeloid cells (Fig. EV5A) or *Tfrc*-deleted myeloid cells (Fig. EV5B) retained the capacity to rescue the lethal outcome of systemic *Fth* deletion in $Slc40a1^{LysM\Delta/\Delta} \Rightarrow Fth^{R26\Delta/\Delta}$ or $Tfrc^{LysM\Delta/\Delta} \Rightarrow Fth^{R26\Delta/\Delta}$ chimeras, respectively (Fig. EV5). This shows that monocyte-derived macrophages rescue the lethal outcome of global *Fth* deletion, via a mechanism that does not rely on Fe transit between *Fth*-competent macrophages and *Fth*-deleted parenchyma cells.

## *Fth*-competent myeloid cells support the mitochondria of *Fth*-deleted parenchyma cells from chimeric *Fth*-deleted mice

Whole-body *Fth* deletion causes a severe disruption of mitochondria structure and function in parenchyma cells (Blankenhaus et al, 2019). This suggests that *Fth*-competent macrophages rescue the lethal outcome of parenchyma *Fth* deletion, via a mechanism that supports the mitochondria of *Fth*-deleted parenchyma cells. In contrast to *Fth*-deletion in adult $Fth^{R26\Delta/\Delta}$ mice (Blankenhaus et al, 2019), *Fth* deletion in $Fth^{R26\Delta/\Delta} \Rightarrow Fth^{R26\Delta/\Delta}$ or $Fth^{LysM\Delta/\Delta} \Rightarrow Fth^{R26\Delta/\Delta}$ chimeras was not associated with a reduction of the number of mitochondria *per* cell, compared to control $Fth^{fl/fl} \Rightarrow Fth^{fl/fl}$ chimeras, as assessed in the heart, WAT and liver (Fig. 6A–C). However, the mitochondria morphology of parenchyma cells from $Fth^{R26\Delta/\Delta} \Rightarrow Fth^{R26\Delta/\Delta}$ chimeras was clearly disrupted, as reveled by swelling and irregular shape of mitochondrial *cristae* and reduced electron density of the mitochondrial matrix (Fig. 6D). Normal mitochondrial morphological features were restored in $Fth^{fl/fl} \Rightarrow Fth^{R26\Delta/\Delta}$ but

not in $Fth^{LysM\Delta/\Delta} \Rightarrow Fth^{R26\Delta/\Delta}$ chimeras (Fig. 6D), suggesting that *Fth*-competent macrophages are essential to support the mitochondrial structure under systemic *Fth* deletion.

## *Fth*-competent myeloid cells restore mitochondria gene expression in chimeric *Fth*-deleted mice

Global *Fth* deletion in $Fth^{R26\Delta/\Delta} \Rightarrow Fth^{R26\Delta/\Delta}$ chimeras was associated with a reduction of the relative level of expression of hepatic mitochondrial mRNA, encoded by mitochondrial (i.e., *MT-Co1*: ETC complex IV Cytochrome C oxidase I, *MT-Cyb*: ETC complex III cytochrome B; *MT-PolG*: Mitochondrial DNA polymerase) or nuclear (*Cs*: Citrate synthase; *Nrf1*: Nuclear respiratory factor 1) genes, as compared to control $Fth^{fl/fl} \Rightarrow Fth^{fl/fl}$ chimeras (Fig. 6E,F). *Fth*-competent hematopoietic cells restored the expression of these hepatic mitochondrial mRNAs in $Fth^{fl/fl} \Rightarrow Fth^{R26\Delta}$ chimeras (Fig. 6E,F). This was no longer the case when *Fth* was deleted in myeloid cells from $Fth^{LysM\Delta/\Delta} \Rightarrow Fth^{R26\Delta/\Delta}$ chimeras (Fig. 6E,G).

Systemic *Fth* deletion was also associated with reduction of the relative level of a subset of mitochondrial ETC proteins, as assessed in the liver of $Fth^{R26\Delta/\Delta} \Rightarrow Fth^{R26\Delta/\Delta}$ *vs.* control $Fth^{fl/fl} \Rightarrow Fth^{fl/fl}$ chimeras (Figs. 6H–J and EV5E). *Fth*-competent hematopoietic cells restored the expression of these mitochondrial ETC proteins (Figs. 6H–J and EV5E), while deletion of *Fth* in myeloid cells compromised this rescuing effect (Figs. 6H–J and EV5E). These observations suggest that *Fth*-competent monocyte-derived macrophages engage in an intercellular crosstalk that supports the mitochondria of *Fth*-deficient parenchyma cells.

## Mitochondrial biogenesis is a hallmark of rescuing macrophages

To explore the mechanism via which *Fth*-competent monocyte-derived macrophages restore the mitochondria of *Fth*-deficient chimeric mice, we generated $tdT^{LysM}$ mice expressing tdT under the control of the LysM promoter. These were crossed with $Fth^{fl/fl}$ mice, to generate $tdT^{LysM}Fth^{LysM\Delta/\Delta}$ mice, deleting *Fth* specifically in tdT$^+$ myeloid cells (Fig. EV6A–C).

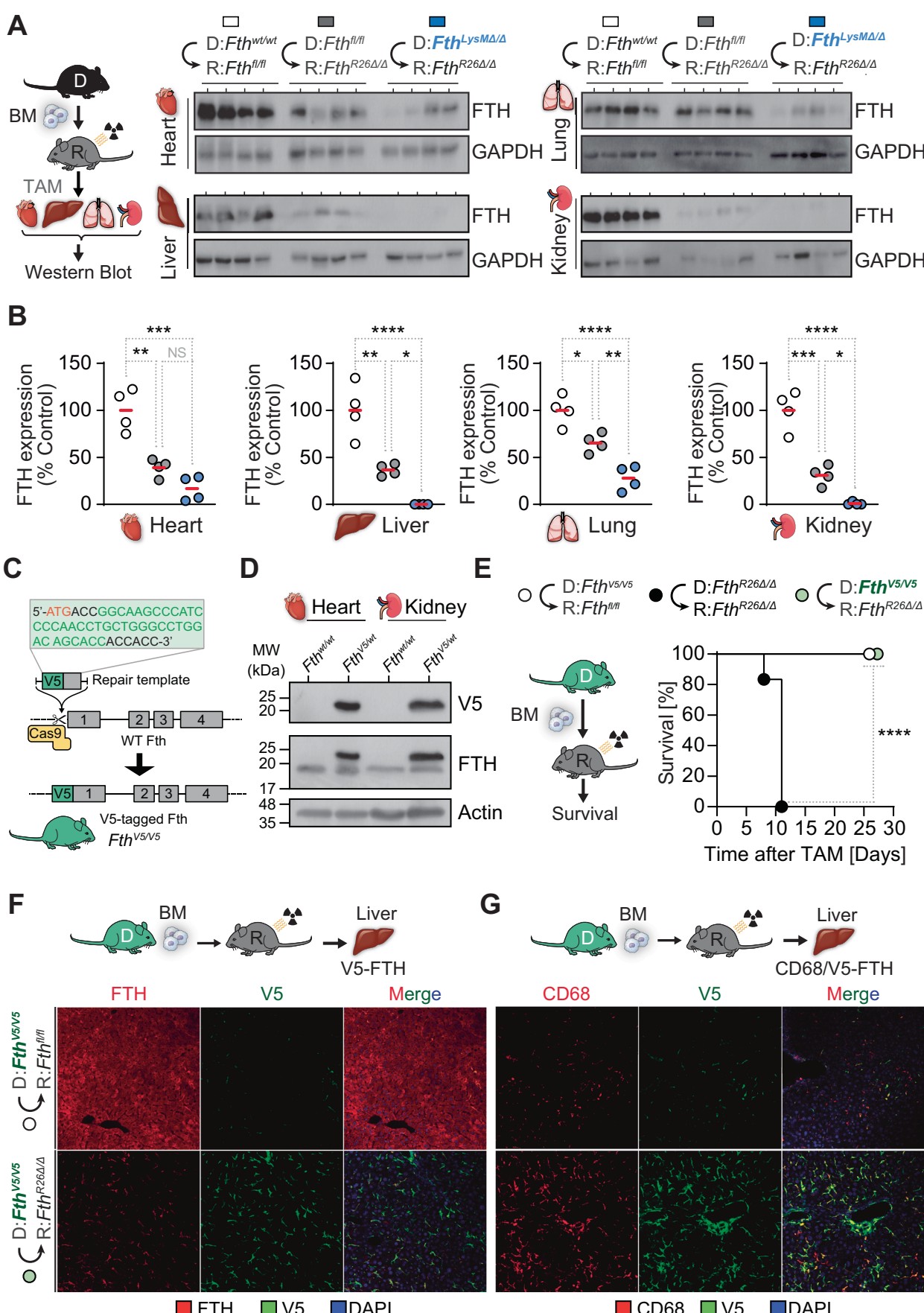

**Figure 5.** *Fth*-competent myeloid cells partially restore tissue ferritin expression but do not deliver ferritin to parenchyma cells in chimeric *Fth*-deleted mice.

(A) Schematic representation of TAM-induced *Fth* deletion in chimeric mice (day 0) and analysis of FTH protein expression (Western blot). (B) Relative quantification of FTH protein expression in heart, liver, lungs and kidneys from $Fth^{fl/fl} \Rightarrow Fth^{fl/fl}$ ($n = 4$), $Fth^{LysM\Delta/\Delta} \Rightarrow Fth^{R26\Delta/\Delta}$ ($n = 4$) and $Fth^{fl/fl} \Rightarrow Fth^{R26\Delta/\Delta}$ ($n = 4$) chimeric mice, between days 7 and 23 post-TAM administration. Data in (B) represented as the % of FTH expression relative to $Fth^{fl/fl} \Rightarrow Fth^{fl/fl}$ control chimeras and is pooled from 3 independent experiments. (C) Schematic representation of the CRISPR/Cas 9 editing strategy for V5-tagged FTH. (D) Detection of V5-tagged and endogenous FTH protein expression (Western blot) in control $Fth^{wt/wt}$ and edited $Fth^{V5/V5}$ mice. (E) Schematic representation of TAM-induced *Fth* deletion (day 0) and survival of $Fth^{V5/V5} \Rightarrow Fth^{fl/fl}$ ($n = 13$), $Fth^{R26\Delta/\Delta} \Rightarrow Fth^{R26\Delta/\Delta}$ ($n = 6$) and $Fth^{V5/V5} \Rightarrow Fth^{R26\Delta/\Delta}$ ($n = 11$) chimeric mice. Data in (E) is pooled from 4 independent experiments with similar trends. (F) Schematic representation of TAM-induced *Fth* deletion in chimeric mice, and representative images of FTH and V5 immunofluorescence detection in livers from $Fth^{V5/V5} \Rightarrow Fth^{fl/fl}$ and $Fth^{V5/V5} \Rightarrow Fth^{R26\Delta/\Delta}$ chimeric mice. (G) Schematic representation of TAM-induced *Fth* deletion and representative images of CD68 (macrophages) and V5 immunofluorescence detection in livers from $Fth^{V5/V5} \Rightarrow Fth^{fl/fl}$ and $Fth^{V5/V5} \Rightarrow Fth^{R26\Delta/\Delta}$ chimeric mice. One-way ANOVA with Tukey's range test for multiple comparison correction was used for comparison between multiple groups. Survival analysis was performed using Log-rank (Mantel–Cox) test. NS: non-significant, *$P < 0.05$, **$P < 0.01$, ***$P < 0.001$, ****$P < 0.0001$. Source data are available online for this figure.

There was a marked reduction in the relative proportion and number of viable monocytes/macrophages in the liver WAT and heart from chimeric mice reconstituted with tdT$^+$ *Fth*-deleted myeloid cells (i.e., $Fth^{R26fl/fl}$ mice reconstituted with $tdT^{LysM}Fth^{LysM\Delta/\Delta}$ BM; $tdT^{LysM}Fth^{LysM\Delta/\Delta} \Rightarrow Fth^{R26\Delta/\Delta}$ chimeras), compared to control chimeras (i.e., $Fth^{fl/fl}$ mice reconstituted with $tdT^{LysM}$ BM; $tdT^{LysM} \Rightarrow Fth^{fl/fl}$) (Fig. EV6A,C,D). This was not observed in chimeric mice reconstituted with *Fth*-competent hematopoietic cells (i.e., $Fth^{R26fl/fl}$ mice reconstituted with $tdT^{LysM}$ BM; $tdT^{LysM} \Rightarrow Fth^{R26\Delta/\Delta}$) (Fig. EV6A,C,D).

To characterize the transcriptional response of rescuing monocyte-derived macrophages, tdT$^+$ (CD45$^+$CD11b$^+$Ly6G$^-$CD19$^-$TCRβ$^-$) monocytes/macrophages were FACS-sorted from the liver and WAT of chimeric mice (Fig. EV6A,C). Analyses of RNA sequencing (RNAseq) revealed a marked upregulation of a large swath of mitochondrial genes in *Fth*-competent *vs.* *Fth*-deleted macrophages from $tdT^{LysM} \Rightarrow Fth^{R26\Delta/\Delta}$ *vs.* $tdT^{LysM}Fth^{LysM\Delta/\Delta} \Rightarrow Fth^{R26\Delta/\Delta}$ chimeras, respectively (Figs. 7A,B and EV6E,F). Gene ontology enrichment showed a wide range of terms directly related to mitochondria (Fig. 7C), suggesting that *Fth* is strictly required to support this transcriptional response.

Analysis of protein-protein interactions for induced genes from enriched mitochondria related ontologies (Fig. 7C), revealed tight interaction networks involving structural mitochondrial, mitoribosome and respirasome/ETC (oxidative phosphorylation) proteins, in both liver (Appendix Fig. S4A) and WAT (Appendix Fig. S4B) macrophages. In addition, the intersection of upregulated genes revealed a network subset shared in both organs, suggesting that these genes are part of a core effector response (Appendix Fig. S4C). Specifically, the transcriptional response of *Fth*-competent macrophages included the induction of genes supporting the ETC, such as *Uqcrq* (complex III subunit), as well as genes involved in mitochondria biogenesis (e.g., *Nrf1* and *PolG2*), maintenance of mitochondrial cristae structure (e.g., *ChChd2*) and mitochondrial protein translation (Fig. 7D). Moreover, we also observed a downregulation of genes in the mitochondria tricarboxylic acid (TCA) cycle (Fig. 7D).

We noted that mitochondrial fission regulator 1 like (*Mtfr1l*), showed decreased expression in in *Fth*-competent *vs.* *Fth*-deleted macrophages (Fig. 7D). This suggests that suppression of mitochondrial regulation through fission contributes to the rescuing capacity of *Fth*-competent macrophages (Fig. 7D).

RNAseq analysis of cardiac *Fth*-competent *vs.* *Fth*-deleted macrophages from chimeric mice, revealed a distinct transcriptional response that did not include the regulation of mitochondrial

genes (Fig. EV7A–D). This suggests that *Fth*-competent monocyte-derived macrophages activate transcriptional programs that are tissue-specific and likely prevent multiorgan dysfunction imposed by global *Fth* deletion.

To monitor the mitochondria of monocyte-derived macrophages, we introduced an additional *Rosa26-mito-Dendra2-Flox-stop-Flox* allele in *LysM^Cre^*, driving the expression of a Dendra2 fluorescent protein fused to mitochondria targeting sequence of the cytochrome c oxidase subunit VIII (mito-Dendra2/PhAM) (Pham et al, 2012) mice. *PhAM^LysM^* mice were crossed with *Fth^fl/fl^* mice to generate *PhAM^LysM^Fth^fl/fl^* mice, deleting *Fth* specifically in PhAM$^+$ myeloid cells.

The relative level of PhAM expression was higher in *Fth*-competent *vs.* *Fth*-deleted monocyte-derived macrophages, from the liver (Fig. 7E,F) and WAT (Fig. 7E,G) of $PhAM^{LysM} \Rightarrow Fth^{R26\Delta/\Delta}$ *vs.* $PhAM^{LysM}Fth^{R26\Delta/\Delta} \Rightarrow Fth^{R26\Delta/\Delta}$ chimeras, respectively. This is consistent with the transcriptional program supporting mitochondrial biogenesis increasing the number of mitochondria in *Fth*-competent *vs.* *Fth*-deleted monocyte-derived macrophages.

Of note, the percentage of PhAM$^+$ monocyte-derived macrophages was lower in WAT from $PhAM^{LysM} \Rightarrow Fth^{R26\Delta/\Delta}$ *vs.* $PhAM^{LysM}Fth^{R26\Delta/\Delta} \Rightarrow Fth^{R26\Delta/\Delta}$ chimeras (Fig. 7G). This suggests that there is an alteration on the relative percentage of one or several other CD45+ populations in $PhAM^{LysM}Fth^{R26\Delta/\Delta} \Rightarrow Fth^{R26\Delta/\Delta}$ or $PhAM^{LysM} \Rightarrow Fth^{R26\Delta/\Delta}$ chimeras.

## Myeloid cells rescue chimeric *Fth*-deleted mice via a mechanism that relies on a transcriptional program supporting mitochondrial biogenesis

To test the involvement of mitochondrial biogenesis on the rescuing capacity of *Fth*-competent macrophages, we deleted TFAM, the master regulator of mitochondrial DNA (mtDNA) transcription and replication (Larsson et al, 1998), specifically in myeloid cells (Wculek et al, 2023). *Tfam*-deleted myeloid cells failed to prevent the lethal outcome of *Fth*-deletion in $Tfam^{LysM\Delta/\Delta} \Rightarrow Fth^{R26\Delta/\Delta}$ chimeras, as compared to $Tfam^{fl/fl} \Rightarrow Fth^{R26\Delta/\Delta}$ chimeras reconstituted with *Tfam*-competent myeloid cells (Fig. 8A). This suggests that mtDNA replication and/or transcription regulation by TFAM is essential to support the rescuing capacity of monocyte-derived macrophage.

TFAM regulates the transcription of several components of the mitochondrial ETC (i.e., complex I, III, IV, and V), supporting oxidative phosphorylation (OXPHOS) (Wculek et al, 2023). To disentangle OXPHOS from other mitochondrial functions

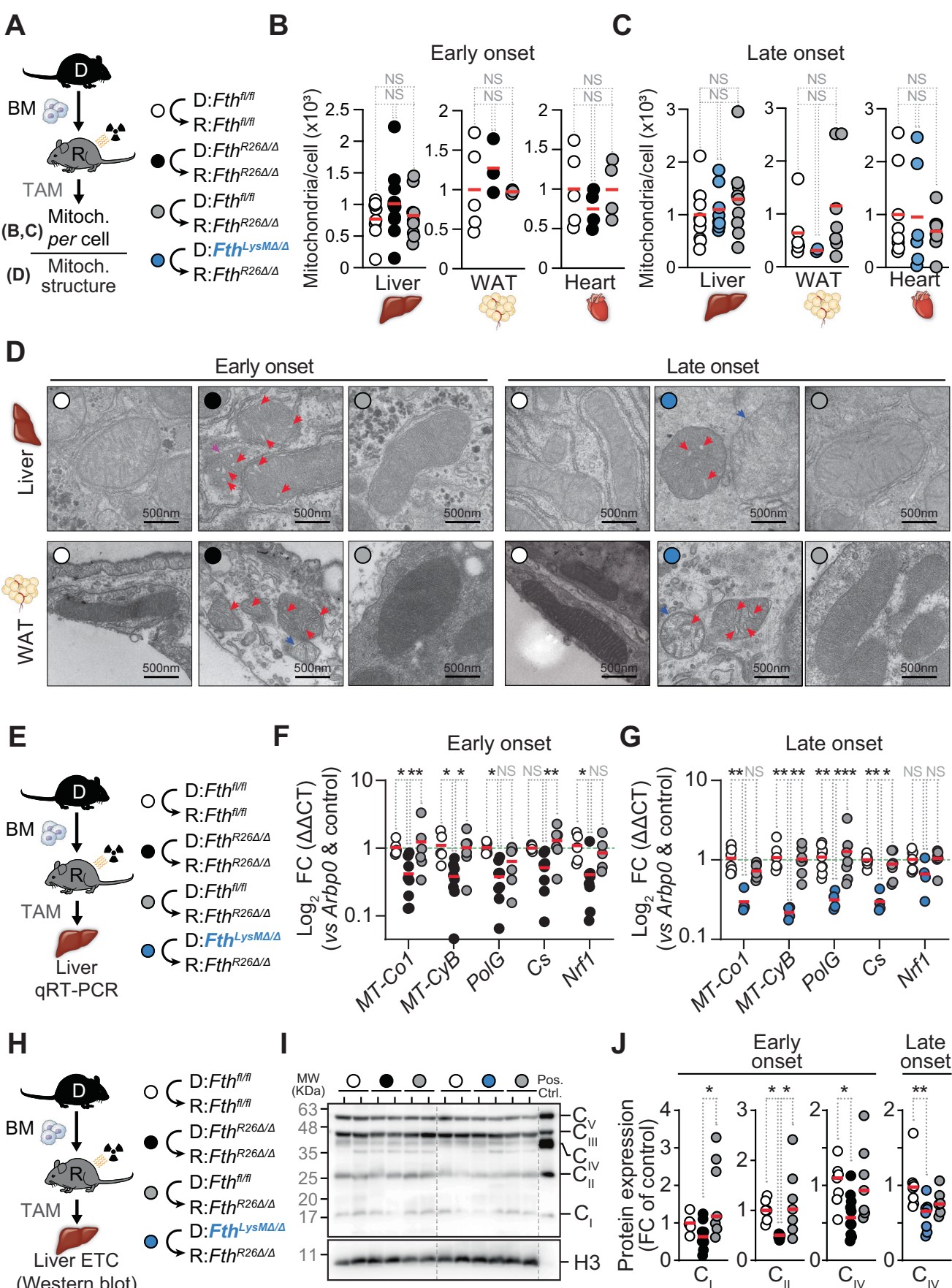

**Figure 6. *Fth*-competent monocyte-derived macrophages support the mitochondria of parenchyma cells from *Fth*-deleted chimeric mice.**

(A) Schematic representation of chimeric mice, TAM administration (day 0) and mitochondrial analysis. (B, C) Number of mitochondria *per* cell in liver, WAT and heart of (B) $Fth^{fl/fl} \Rightarrow Fth^{fl/fl}$ ($n = 5$–8), $Fth^{R26\Delta/\Delta} \Rightarrow Fth^{R26\Delta/\Delta}$ ($n = 3$–12) and $Fth^{fl/fl} \Rightarrow Fth^{R26\Delta/\Delta}$ ($n = 3$–9) chimeric mice, between days 7 and 9 (early onset) or (C) $Fth^{fl/fl} \Rightarrow Fth^{fl/fl}$ ($n = 6$–10), $Fth^{LysM\Delta/\Delta} \Rightarrow Fth^{R26\Delta/\Delta}$ ($n = 2$–8) and $Fth^{fl/fl} \Rightarrow Fth^{R26\Delta/\Delta}$ ($n = 7$–8) chimeric mice, on day 22–39 (late onset). Data in (B, C) represented as individual values (circles) and mean (red bars), assessed by quantitative PCR according to the ratio of nuclear (i.e., hexokinase 2; *Hk2*) and mitochondrial (i.e., NADH-ubiquinone oxidoreductase chain 1; *mt-Nd1*) DNA. (D) Representative transmission electron microscopy images of mitochondria structure in gWAT adipocytes and liver hepatocytes from $Fth^{fl/fl} \Rightarrow Fth^{fl/fl}$, $Fth^{R26\Delta/\Delta} \Rightarrow Fth^{R26\Delta/\Delta}$ and $Fth^{fl/fl} \Rightarrow Fth^{R26\Delta/\Delta}$ chimeric mice, on day 8 (early onset), or from $Fth^{fl/fl} \Rightarrow Fth^{fl/fl}$, $Fth^{LysM\Delta/\Delta} \Rightarrow Fth^{R26\Delta/\Delta}$ and $Fth^{fl/fl} \Rightarrow Fth^{R26\Delta/\Delta}$ chimeric mice, on day 30 (late onset). Red arrows indicate swollen cristae, purple arrows indicate disrupted mitochondrial membranes, blue arrows indicate loss of mitochondrial membrane potential, as suggested by decreased electron density. (E) Schematic representation of chimeric mice, TAM administration (day 0) and qRT-PCR assessment. (F, G) Log$_2$ fold change (FC) in expression of mitochondrial genes or genes involved in regulation of mitochondria in livers from (F) $Fth^{fl/fl} \Rightarrow Fth^{fl/fl}$ ($n = 5$), $Fth^{R26\Delta/\Delta} \Rightarrow Fth^{R26\Delta/\Delta}$ ($n = 9$) and $Fth^{fl/fl} \Rightarrow Fth^{R26\Delta/\Delta}$ ($n = 6$) chimeric mice, between days 7 and 8 (early onset), or from (G) $Fth^{fl/fl} \Rightarrow Fth^{fl/fl}$ ($n = 9$), $Fth^{LysM\Delta/\Delta} \Rightarrow Fth^{R26\Delta/\Delta}$ ($n = 4$) and $Fth^{fl/fl} \Rightarrow Fth^{R26\Delta/\Delta}$ ($n = 7$) chimeric mice, between days 19 and 22 (late onset). Data in (F, G) is pooled from three independent experiments with similar trend and represented as individual values (circles) and mean (red bars), normalized to housekeeping gene expression (acidic ribosomal phosphoprotein P0, *Arbp0*) and to gene expression values of control $Fth^{fl/fl} \Rightarrow Fth^{fl/fl}$ chimeric mice. (H) Schematic representation of chimeric mice, TAM administration (day 0) and western blot assessment. (I) Representative Western blot and (J) quantification of ETC subunits for complexes I-V in the livers of $Fth^{fl/fl} \Rightarrow Fth^{fl/fl}$ ($n = 7$), $Fth^{R26\Delta/\Delta} \Rightarrow Fth^{R26\Delta/\Delta}$ ($n = 10$) and $Fth^{fl/fl} \Rightarrow Fth^{R26\Delta/\Delta}$ ($n = 7$) chimeric mice, between days 7 and 8 (early onset), or from $Fth^{fl/fl} \Rightarrow Fth^{fl/fl}$ ($n = 8$), $Fth^{LysM\Delta/\Delta} \Rightarrow Fth^{R26\Delta/\Delta}$ ($n = 8$) and $Fth^{fl/fl} \Rightarrow Fth^{R26\Delta/\Delta}$ ($n = 6$) chimeric mice, between days 19 and 22 (late onset). Data in (J) is pooled from 4 independent experiments with similar trends. One-way ANOVA with Tukey's range test for multiple comparison correction was used for comparison between multiple groups. Two-way ANOVA with Tukey's range test for multiple comparison correction was used for comparison between multiple groups within multiple genes. NS: non-significant, *$P < 0.05$, **$P < 0.01$, ***$P < 0.001$. Source data are available online for this figure.

supported by TFAM, we deleted *Uqcrq*, the gene encoding the Ubiquinol-cytochrome c reductase, complex III subunit VII, specifically in myeloid cells from $Uqcrq^{LysM\Delta/\Delta}$ mice (Wculek et al, 2023; Weinberg et al, 2019). *Uqcrq*-deleted myeloid cells retained the capacity to prevent the lethal outcome of *Fth*-deletion in $Uqcrq^{LysM\Delta/\Delta} \Rightarrow Fth^{R26\Delta/\Delta}$ chimeras, similar to *Uqcrq*-competent myeloid cells in $Uqcrq^{fl/fl} \Rightarrow Fth^{R26\Delta/\Delta}$ chimeras (Fig. 8B). This suggests that OXPHOS is not essential to support the rescuing capacity of monocyte-derived macrophages.

### *Fth*-competent myeloid cells engage in intercellular mitochondria transfer with parenchyma cells from chimeric *Fth*-deleted mice

Intercellular mitochondria transfer is an evolutionarily conserved process whereby mitochondria are delivered from donor to acceptor cells, as a mechanism to maintain mitochondria quality in donor cells and/or to provide metabolic support to acceptor cells (Borcherding and Brestoff, 2023; Brestoff et al, 2025; Nakai et al, 2024). We reasoned that *Fth*-competent monocyte-derived macrophages support the mitochondria of *Fth*-deleted parenchyma cells through a mechanism associated with intercellular mitochondrial transfer. To test this hypothesis, we generated chimeric mice expressing mito-Dendra2/PhAM specifically in mitochondria from myeloid *vs*. parenchyma cells. Intercellular mitochondria transfer, from donor to acceptor cells, was quantified by the PhAM index of acceptor cells (median mito-Dendra2/PhAM fluorescence intensity x % mito-Dendra2/PhAM$^+$ cells).

Intercellular mitochondria transfer from donor PhAM$^+$ monocyte-derived macrophages to acceptor parenchyma (tdT$^+$) cells was not detectable, as assessed in the liver of control $PhAM^{LysM} \Rightarrow tdT^{R26}$ chimeras (i.e., $tdT^{R26}$ mice reconstituted with $PhAM^{LysM}$ BM) (Figs. 8C–E and EV7E–G). There was also no detectable intercellular mitochondria transfer from donor PhAM$^+$ *Fth*-competent or *Fth*-deleted monocyte-derived macrophages to acceptor *Fth*-deleted parenchyma cells, as assessed in the liver of $PhAM^{LysM} \Rightarrow tdT^{R26}Fth^{R26\Delta/\Delta}$ and $PhAM^{LysM}Fth^{LysM\Delta/\Delta} \Rightarrow tdT^{R26}Fth^{R26\Delta/\Delta}$ chimeras, respectively (Figs. 8C–E and EV7E–G). This suggests that the mechanism via which *Fth*-competent monocyte-derived

macrophages support the mitochondria of *Fth*-deleted parenchyma cells is not associated with intercellular mitochondrial transfer from macrophages to parenchyma cells.

In contrast, there was intercellular mitochondria transfer from donor PhAM$^+$ parenchyma cells to acceptor monocyte-derived macrophages, as assessed in the liver of control $Fth^{fl/fl} \Rightarrow PhAM^{R26}$ chimeras (i.e., $PhAM^{R26}$ reconstituted with $Fth^{fl/fl}$ BM) (Fig. 8F–H and EV7H–J). Intercellular mitochondria transfer from parenchyma cells to macrophages was also observed in the liver from $Fth^{LysM\Delta/\Delta} \Rightarrow PhAM^{R26}Fth^{R26\Delta/\Delta}$ (i.e., $PhAM^{R26}Fth^{R26fl/fl}$ mice reconstituted with $Fth^{LysM\Delta/\Delta}$ BM) and $Fth^{fl/fl} \Rightarrow PhAM^{R26}Fth^{R26\Delta/\Delta}$ (i.e., $PhAM^{R26}Fth^{R26fl/fl}$ mice reconstituted with $Fth^{fl/fl}$ BM) chimeras (Figs. 8F–H and EV7H–J). Importantly, there was tenfold reduction in the PhAM index of hepatic monocyte-derived macrophages from $Fth^{LysM\Delta/\Delta} \Rightarrow PhAM^{R26}Fth^{R26\Delta/\Delta}$ *vs*. $Fth^{fl/fl} \Rightarrow PhAM^{R26}Fth^{R26\Delta/\Delta}$ chimeras or control $Fth^{fl/fl} \Rightarrow PhAM^{R26}$ chimeras (Figs. 8F–H and EV7H–J). This suggests that *Fth*-competent monocyte-derived macrophages support the mitochondria of *Fth*-deleted parenchyma cells via a mechanism associated with intercellular mitochondrial transfer, from donor parenchyma cells to acceptor. Moreover, this also suggest that FTH is required to support the capacity of macrophages to accept mitochondria from parenchyma cells.

Mitochondria are iron-rich organelles (Ben Zichri-David et al, 2025), suggesting that their transfer and uptake by macrophages is cytotoxic in the absence of FTH. Mitochondria purified from the liver of *Fth*-deficient mice were cytotoxic to *Fth*-deficient but not *Fth*-competent BMDM (Fig. 8I–K). This was mimicked by exposure of *Fth*-deficient BMDM to exogenous Fe (ferric ammonium citrate), at a concentration not lethal to *Fth*-competent BMDM (Fig. 8I–L). This suggests that FTH is strictly required to maintain macrophage viability following the uptake of exogenous mitochondria.

## Discussion

Our findings suggest that monocyte-derived macrophages carry an intrinsic capacity to sense the iron status of parenchyma cells and to respond in a manner that supports tissue function and

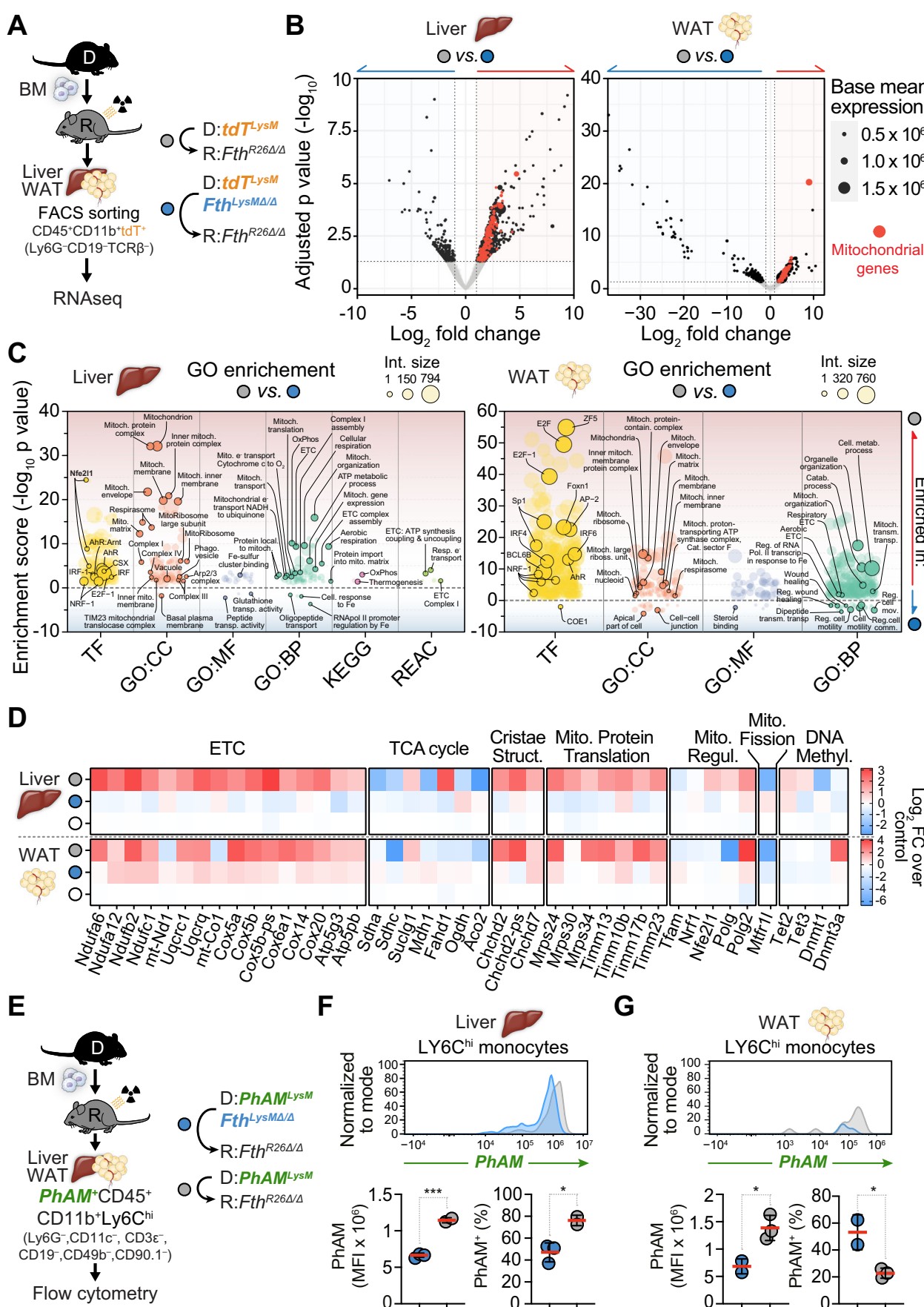

**Figure 7. *Fth*-competent monocyte-derived macrophages deploy a mitochondrial gene transcriptional program.**

(A) Schematic representation of chimeric mice, TAM administration and fluorescence-activated cell sorting (FACS) of *LysM*⁺ monocyte/macrophages (CD45⁺,CD11b⁺,Ly6G⁻,CD19⁻,TCRβ⁻) in liver and WAT. (B) Volcano plots of differentially regulated genes between *LysM*⁺ monocyte/macrophages sorted from liver (left) or WAT (right) of *tdT^{LysM}⇒Fth^{R26Δ/Δ}* (n = 3), *tdT^{LysM}Fth^{LysMΔ/Δ}⇒Fth^{R26Δ/Δ}* (n = 5) chimeric mice, on day 19 post-TAM administration. Red dots depict mitochondrial genes that are significantly differentially regulated. (C) Gene ontology analysis depicting ontologies that are significantly enriched comparing *LysM*⁺ monocyte/macrophages sorted from liver (left) and WAT (right) from *tdT^{LysM}⇒ Fth^{R26Δ/Δ}* (n = 3) and *tdT^{LysM}Fth^{LysMΔ/Δ}⇒Fth^{R26Δ/Δ}* (n = 5) chimeric mice on day 19 post-TAM administration. Ontologies significantly enriched in *LysM*⁺ monocyte/macrophages from *tdT^{LysM}⇒ Fth^{R26Δ/Δ}* are depicted as: enrichment score −log₁₀ P value > 1.301. Ontologies significantly enriched in *LysM*⁺ monocyte/macrophages from *tdT^{LysM}Fth^{LysMΔ/Δ}⇒Fth^{R26Δ/Δ}* are depicted as: enrichment score −log₁₀ P value < −1.301. Ontology classes: TF = transcription factors; GO:CC = gene ontology : cellular component; GO:MF = GO : molecular function; GO:BP = GO : biological process; KEGG = Kyoto Encyclopedia of Genes and Genomes pathway; REAC = reactome. (D) Heatmap of mean Log₂ FC (over control; *tdT^{LysM}⇒ Fth^{fl/fl}*) in gene expression of genes involved in mitochondrial function and regulation, in *LysM*⁺ monocyte/macrophages sorted from the liver (top) and WAT (bottom) of *tdT^{LysM}⇒Fth^{fl/fl}* (n = 3), *tdT^{LysM}⇒Fth^{R26Δ/Δ}* (n = 3) and *tdT^{LysM}Fth^{LysMΔ/Δ}⇒Fth^{R26Δ/Δ}* (n = 5) chimeric mice on day 19 post-TAM administration. (E) Schematic representation of chimeric mice, TAM administration and flow cytometry analysis of PhAM reporter-expressing mitochondria in *LysM*⁺ cells. Representative histograms of PhAM fluorescence intensity, mean fluorescence intensity (MFI) of PhAM, and percentage of PhAM⁺ cells in LY6C^{high} monocytes from the liver (F) and WAT (G) of *PhAM^{LysM}⇒Fth^{fl/fl}* (n = 2–3) and *PhAM^{LysM}Fth^{LysMΔ/Δ}⇒Fth^{R26Δ/Δ}* (n = 2–3) chimeric mice, collected on day 100 post-TAM administration. Data in (F, G) dot plots represented as individual values (circles) and mean (red bars). Student's t test was used for comparison between two groups. NS non-significant, *P < 0.05, ***P < 0.001. Source data are available online for this figure.

organismal homeostasis. This is consistent with the notion of tissues representing an emergent property of the interactions between "primary" and "supportive" cells (Adler et al, 2023; Meizlish et al, 2021; Zhou et al, 2018), whereby macrophages act as ferrostats to facilitate tissue-specific parenchyma cell functions (Winn et al, 2020).

The interaction of macrophages with parenchyma cells is thought to give rise to integrated cellular modules (Bonnardel et al, 2019). Additional interactions with peripheral neurons provide these with capacity to surveil and respond to systemic variations of vital parameters (Godinho-Silva et al, 2019; Veiga-Fernandes and Mucida, 2016). In adipose tissue, these integrated cellular modules regulate WAT lipolysis (Ko et al, 2020; Pirzgalska et al, 2017) and BAT thermoregulation (Wolf et al, 2017; Zeng et al, 2015). This occurs via stable physical interaction of macrophages with adipocytes, vascular endothelial cells (Moura Silva et al, 2021; Silva et al, 2019), mesenchymal cells (Ko et al, 2020) and peripheral neurons (Pirzgalska et al, 2017). Our finding that *Fth*-competent monocyte-derived macrophages control BAT thermogenesis as well as BAT and WAT lipolysis in *Fth*-deleted chimeras (Figs. 4 and EV4) is consistent with this tissue organizing principle (Adler et al, 2023; Meizlish et al, 2021; Zhou et al, 2018).

The notion that *Fth*-competent macrophages act as ferrostats, sensing and responding to tissue iron status, to control WAT and BAT function provides further understanding of how regulation of Fe metabolism modulates WAT function (Blankenhaus et al, 2019; Romero et al, 2022; Wang et al, 2024; Yook et al, 2021), BAT thermogenesis (Blankenhaus et al, 2019; Wang et al, 2024; Yook et al, 2021) and energy balance (Blankenhaus et al, 2019; Lu et al, 2024; Wang et al, 2024). This is also consistent with dietary Fe-deficiency compromising thermoregulation in rodents (Dillmann et al, 1979) and humans (Beard et al, 1990; Brigham and Beard, 1996; Lukaski et al, 1990) as well as with dietary Fe overload causing a negative energy balance in rodents (Romero et al, 2022).

That circulating *Fth*-competent monocyte-derived, rather than tissue-resident, macrophages take control of energy homeostasis is demonstrated using parabiosis (Fig. 1I) as well as by the adoptive transfer of BMDM (Fig. 1J). While illustrating the extraordinary capacity of circulating monocytes to partake in the inter-organ crosstalk that regulates systemic Fe and energy metabolism, this does not exclude other circulating myeloid-derived, such as

myeloid-derived suppressor cells (Veglia et al, 2021), from contributing to this process.

Fe exerts a major impact on macrophage function (Soares and Hamza, 2016) suggesting that macrophages act as ferrostats via a mechanism that senses and responds directly to the Fe status of parenchyma cells. This hypothesis, however, is not supported by macrophages retaining the capacity to rescue the lethal outcome of global *Fth* deletion irrespectively of cellular Fe import via TFR1 (Fig. EV5A,B) or Fe export via SLC40a1 (Fig. EV5C,D). Moreover, macrophages rescue the lethal outcome of global *Fth* deletion irrespectively of ferritin secretion and transfer to parenchyma cells (Fig. 5C–G). Overall, this suggests that *Fth*-competent macrophages do not rely on intercellular Fe sensing and/or transfer to support *Fth*-deleted parenchyma cells.

As an alternative hypothesis, macrophages act as ferrostats via a mechanism that senses and responds to the consequences of dysregulated tissue Fe metabolism, namely, mitochondrial dysfunction (Fig. 6D-G) (Blankenhaus et al, 2019). In support of this hypothesis, macrophages respond to a number of cues released from dysfunctional mitochondria, including mtDNA (Murphy and O'Neill, 2024). In support of with this hypothesis *Fth*-competent macrophages respond to *Fth* deletion in parenchyma cells via the induction of a gene expression profile consistent with a type I interferon response (Fig. 7A–C), a hallmark of the macrophage response to mtDNA (Al Amir Dache and Thierry, 2023; He et al, 2022). Moreover, *Fth* deletion in parenchyma cells is associated with remodeling of the mitochondria *cristae* structure (Fig. 6D) (Blankenhaus et al, 2019), which regulate mtDNA release from mitochondria to induce a type I interferon response in macrophages (He et al, 2022).

*Fth*-competent macrophages respond to *Fth* deletion in parenchyma cells via induction of a singular gene expression profile associated with maintenance of mitochondrial function (i.e., ETC, TCA) and structure (i.e., *cristae*), as well as with mitochondrial biogenesis (Fig. 7A–G). This genetic program is controlled by the mitochondrial transcriptional regulator TFAM (Larsson et al, 1998), which is required for macrophages to rescue from the lethal outcome of global *Fth* deletion (Fig. 8A).

While TFAM supports the expression of mitochondrial ETC genes (Wculek et al, 2023), this is probably not essential to rescue the lethal outcome of global *Fth* deletion (Fig. 8B). This suggests that mitochondria biogenesis is a functional hallmark of

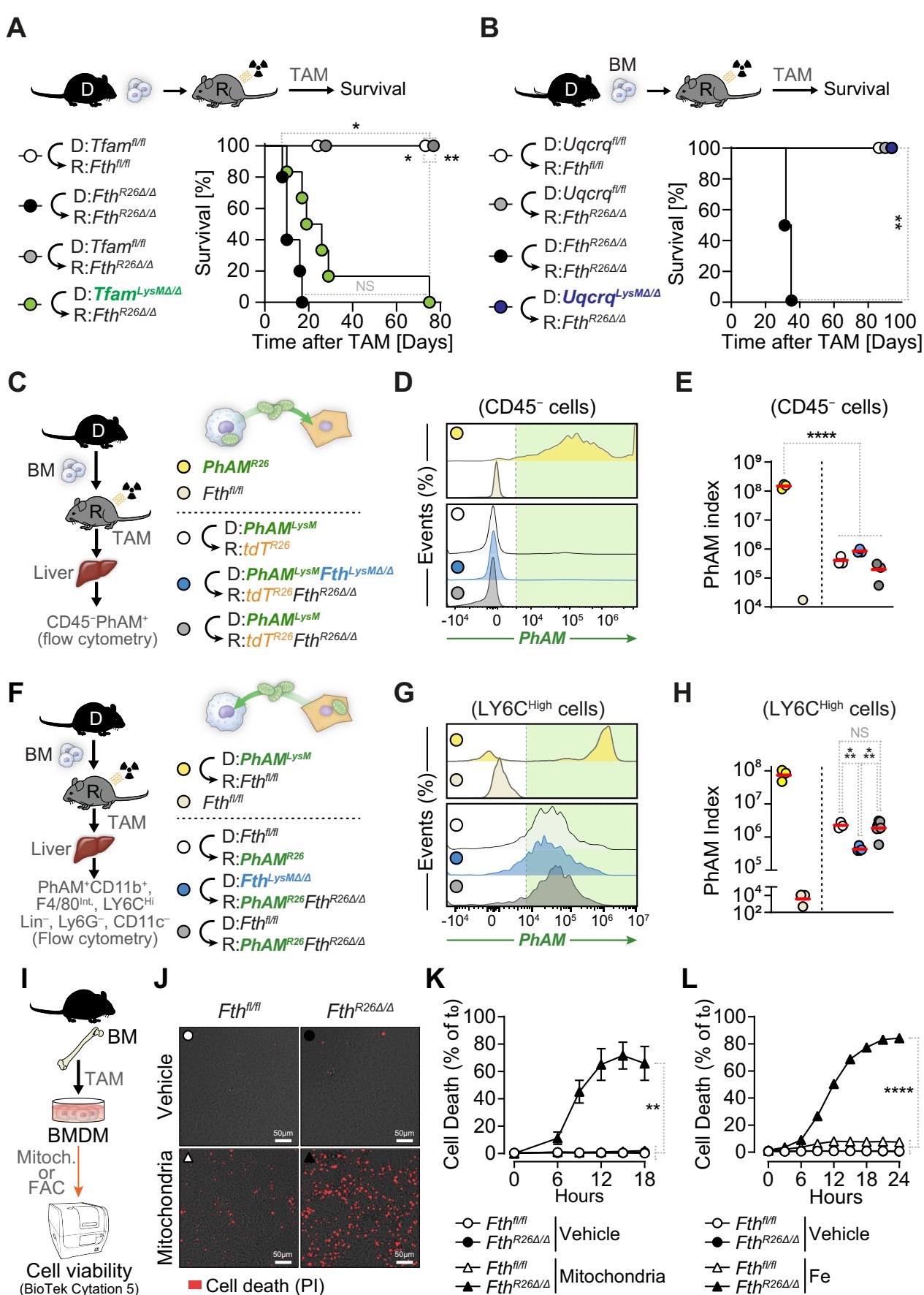

**Figure 8. Mitochondrial biogenesis supports the rescuing capacity of *Fth*-competent monocyte-derived macrophages in chimeric *Fth*-deleted mice.**

(A) Schematic representation of chimeric mice, TAM administration (day 0), and survival of $Tfam^{fl/fl} \Rightarrow Fth^{fl/fl}$ ($n = 4$), $Fth^{R26\Delta/\Delta} \Rightarrow Fth^{R26\Delta/\Delta}$ ($n = 5$), $Tfam^{fl/fl} \Rightarrow Fth^{R26\Delta/\Delta}$ ($n = 6$) and $Tfam^{LysM\Delta/\Delta} \Rightarrow Fth^{R26\Delta/\Delta}$ ($n = 6$) chimeric mice on day 0. Data in (B) is pooled from 2 independent experiments with similar trends. (B) Schematic representation of chimeric mice, TAM administration (day 0), and survival of $Uqcrq^{fl/fl} \Rightarrow Fth^{fl/fl}$ ($n = 6$), $Uqcrq^{fl/fl} \Rightarrow Fth^{R26\Delta/\Delta}$ ($n = 6$), $Fth^{R26\Delta/\Delta} \Rightarrow Fth^{R26\Delta/\Delta}$ ($n = 2$), and $Uqcrq^{LysM\Delta/\Delta} \Rightarrow Fth^{R26\Delta/\Delta}$ ($n = 5$) chimeric mice on day 0. (C) Schematic representation of chimeric mice, TAM administration (day 0), and flow cytometry analysis of livers from positive ($PhAM^{R26}$; $n = 3$) and negative control ($Fth^{fl/fl}$; $n = 1$) mice, and $PhAM^{LysM} \Rightarrow tdT^{R26}$ ($n = 3$), $PhAM^{LysM}Fth^{LysM\Delta/\Delta} \Rightarrow tdT^{R26}Fth^{R26\Delta/\Delta}$ ($n = 3$), $PhAM^{LysM} \Rightarrow tdT^{R26}Fth^{R26\Delta/\Delta}$ ($n = 3$) chimeras with (D) representative PhAM fluorescence histograms and (E) PhAM expression index of CD45$^-$ parenchymal cells. PhAM index is calculated as the percentage of CD45$^-$ cells that are PhAM$^+$, multiplied by CD45$^-$PhAM$^+$ cells' MFI. (F) Schematic representation of chimeric mice, TAM administration (day 0), and flow cytometry analysis of livers from positive ($PhAM^{LysM} \Rightarrow Fth^{fl/fl}$; $n = 3$) and negative control ($Fth^{fl/fl}$; $n = 3$) mice, and $Fth^{fl/fl} \Rightarrow PhAM^{R26}$ ($n = 3$), $Fth^{LysM\Delta/\Delta} \Rightarrow PhAM^{R26}Fth^{R26\Delta/\Delta}$ ($n = 3$), $Fth^{fl/fl} \Rightarrow PhAM^{R26}Fth^{R26\Delta/\Delta}$ ($n = 3$) chimeras with (G) representative PhAM fluorescence histograms and (H) PhAM expression index of LY6C$^{High}$ monocyte-derived macrophages (CD11b$^+$, F4/80$^{Int}$, LY6C$^{Hi}$, Lin$^-$, LY6G$^-$, CD11C$^-$, PhAM$^+$). PhAM index is calculated as the percentage of LY6C$^{High}$ monocyte-derived macrophages that are PhAM$^+$, multiplied by their PhAM MFI. Data in H is pooled from 2 independent experiments. (I) Schematic representation of control ($Fth^{fl/fl}$) or FTH-deficient ($Fth^{R26\Delta/\Delta}$) BMDM generation, Tamoxifen (TAM) administration and cell viability assessment. (J) Representative brightfield and fluorescent microscopy images of viability staining (PI) of BMDM treated with vehicle, or mitochondria, 15 h post-treatment. (K) Time course of BMDM cell death upon vehicle or mitochondria administration. For (J, K), wells receiving mitochondria were administered mitochondria purified from 60 mg of liver. (L) Time course of BMDM cell death upon vehicle or Fe (FAC, ferric ammonium citrate; 125 μM) administration. Data in (E, H) is presented as individual values and mean. Data in (K, L) is presented as mean $+/-$ SEM ($n = 3$ replicates per condition). Two-way ANOVA was used for comparison between two groups within time courses. Survival analysis was performed using Log-rank (Mantel–Cox) test. Survival analysis in (A) is corrected for multiple comparisons using Bonferroni test. NS: non-significant, $*P < 0.05$, $**P < 0.01$, $***P < 0.001$, $****P < 0.0001$. Source data are available online for this figure.

macrophages with the capacity to rescue the lethal outcome of *Fth* deletion in parenchyma cells. This transcriptional program (Fig. 7A–D) was activated exclusively in *Fth*-competent macrophages (Fig. EV6C–F), suggesting that FTH is essential to support this response.

There are several possible mechanisms via which FTH can regulate the transcriptional program supporting mitochondria biogenesis in macrophages. One possibility is that this occurs via the regulation of DNA methyl transferase 3 A (DNMT3A), consistent with FTH supporting DNMT3A expression (Ye et al, 2019) and with DNMT3A inducing the expression of TFAM in macrophages (Cobo et al, 2022). In support of this hypothesis, DNMT3A was highly induced in *Fth*-competent macrophages that rescue the lethal outcome of global *Fth* deletion (Fig. 7D). A non-mutually exclusive possibility is that FTH regulates TFAM via a mechanism involving the ten-eleven translocation (TET) dioxygenases, which use Fe as an essential co-factor and the TCA-derived α-ketoglutarate as a substrate to catalyze cytosine demethylation (Lopez-Moyado et al, 2024). This is consistent with FTH regulating cytosine demethylation by TET dioxygenases (Wu et al, 2024) and with TET inducing the expression of TFAM in macrophages (Pan et al, 2017).

The capacity of *Fth*-competent macrophages to restore the mitochondria of *Fth*-deleted chimeric mice (Figs. 6D–J and EV5E) was associated with intercellular mitochondrial transfer from donor *Fth*-deleted parenchyma cells to acceptor monocyte-derived macrophages (Figs. 8F–H and EV7H–J). In contrast there is no intercellular mitochondrial transfer from donor *Fth*-competent macrophages to acceptor *Fth*-deleted parenchyma cells (Figs. 8C–E and EV7E–G).

These observations are consistent with *Fth*-competent macrophages supporting the mitochondria of parenchyma cells via a mechanism involving intercellular transfer of dysfunctional mitochondria from donor *Fth*-deleted parenchyma cells. Presumably, this allows *Fth*-deleted parenchyma cells to outsource energetically demanding mitophagy via intercellular transfer of their dysfunctional mitochondria. This process termed transmitophagy (Nicolás-Ávila et al, 2022) was reported in other experimental systems (Brestoff et al, 2021; Nicolás-Ávila et al, 2020; Rosina et al, 2022).

The cytoprotective effect of FTH (Berberat et al, 2003; Gozzelino et al, 2012; Pham et al, 2004) is required for macrophages to handle the Fe contained in the mitochondria transferred from *Fth*-deleted parenchyma cells. Presumably, this explains the loss of *Fth*-deleted monocyte-derived macrophages when FTH is also deleted in parenchyma cells (Figs. 1F,J and EV6D).

In conclusion, monocyte-derived macrophages act as ferrostats to take control of organismal Fe, redox and energy metabolism in response to global *Fth* deletion. This extraordinary capacity relies on a mechanism whereby FTH supports a transcriptional program that acts in a cell autonomous manner to promote mitochondrial biogenesis and in a non-cell autonomous manner to regulate the mitochondria of parenchyma cells in different tissues. These findings support the notion that macrophages play a central role in supporting the function of parenchyma cells in different tissues to sustain homeostatic control of multicellular organisms.

## Methods

**Reagents and tools table**

| Reagent/resource | Reference or source | Identifier or catalog number |
| --- | --- | --- |
| **Experimental models** | | |
| $Fth^{fl/fl}$ (C57BL/6) | Prof. Lukas Kuhn (ETH, Switzerland) | N/A |
| $R26^{CreERT2}$ | The Jackson Laboratory | Strain: 008463 |
| $Cx3cr1^{Cre}$ | The Jackson Laboratory | Strain: 025524 |
| $LysM^{Cre}$ | The Jackson Laboratory | Strain: 004781 |
| $CD2^{Cre}$ | The Jackson Laboratory | Strain: 027406 |
| $OKD48^{Luc}$ | Oikawa et al, 2012 | Prof. Iwaki T. (RIKEN 2-1, Wako, Japan) |
| OKD48$^{Luc}Fth^{fl/fl}$ | Blankenhaus et al, 2019 | N/A |

| Reagent/resource | Reference or source | Identifier or catalog number |
|---|---|---|
| OKD48$^{Luc}$R26$^{CreERT2}$ | Blankenhaus et al, 2019 | N/A |
| OKD48$^{Luc}$R26$^{CreERT2}$Fth$^{\Delta/\Delta}$ | Blankenhaus et al, 2019 | N/A |
| R26$^{CreERT2}$Fth$^{\Delta/\Delta}$ | This manuscript | N/A |
| LysM$^{Cre}$Fth$^{\Delta/\Delta}$ | This manuscript | N/A |
| CD2$^{Cre}$Fth$^{\Delta/\Delta}$ | This manuscript | N/A |
| Cx3CR1$^{Cre}$Fth$^{\Delta/\Delta}$ | This manuscript | N/A |
| C57BL/6 Ly5.1 | The Jackson Laboratory | Strain: 002014 |
| Ccr2$^{-/-}$ | The Jackson Laboratory | Strain: 004999 |
| B6.129S Tfrc$^{fl/fl}$ | The Jackson Laboratory | Strain: 028363 |
| C57BL/6 Slc40a1$^{fl/fl}$ | Wu et al, 2023 | N/A |
| C57BL/6 Tfam$^{fl/fl}$ | Larsson et al, 1998 | MGI: 1860962 |
| C57BL/6 Uqcrq$^{fl/fl}$ | Weinberg et al, 2019 | Referred to as CIII$^{fl/fl}$; MGI: 6358198 |
| LysM$^{Cre}$ Tfrc$^{\Delta/\Delta}$ | This manuscript | N/A |
| LysM$^{Cre}$ Slc40a1$^{fl/fl}$ | This manuscript | N/A |
| LysM$^{Cre}$ Tfam$^{fl/fl}$ | Prof. David Sancho, CNIC, Spain | N/A |
| LysM$^{Cre}$ Uqcrq$^{fl/fl}$ | Prof. David Sancho, CNIC, Spain | N/A |
| R26$^{tdTomato}$ | The Jackson Laboratory | Strain: 007909 |
| R26$^{tdTomato/CreERT2}$ | This manuscript | N/A |
| R26$^{tdTomato/CreERT2}$Fth$^{\Delta/\Delta}$ | This manuscript | N/A |
| LysM$^{Cre}$ R26$^{tdTomato}$ | This manuscript | N/A |
| LysM$^{Cre}$ R26$^{tdTomato}$Fth$^{\Delta/\Delta}$ | This manuscript | N/A |
| Fth$^{V5/V5}$ | This manuscript | N/A |
| R26$^{PhAM}$ | The Jackson Laboratory | Referred to as PhAM$^{fl/fl}$; Strain: 018385 |
| R26$^{PhAM/CreERT2}$ | This manuscript | N/A |
| R26$^{PhAM/CreERT2}$Fth$^{\Delta/\Delta}$ | This manuscript | N/A |
| LysM$^{Cre}$R26$^{PhAM}$ | This manuscript | N/A |

| Reagent/resource | Reference or source | Identifier or catalog number |
|---|---|---|
| LysM$^{Cre}$R26$^{PhAM}$Fth$^{\Delta/\Delta}$ | This manuscript | N/A |
| **Antibodies** | | |
| Anti-mouse-FTH1 | Cell Signaling | #4393 |
| Anti-V5-HRP | Invitrogen | #46-0708 |
| Anti-GAPDH | SICGEN | # AB0049-200 |
| Anti- β-actin | Sigma | #A5441 |
| HRT-conjugated anti-rabbit IgG– | SantaCruz Biotechnology | sc-2030 |
| HRT-conjugated anti-mouse IgG– | SantaCruz Biotechnology | sc-2005 |
| HRT-conjugated anti-goat IgG– | ThermoFisher | PA1-28664 |
| OxPhos Rodent WB Antibody Cocktail | ThermoFisher | #45-8099 |
| HRT-conjugated anti-mouse IgG antibody | Cell signalling | #7076P2 |
| **Flow cytometry antibodies** | | |
| Cx3CR1-APC | Biolegend | #149007 |
| CD11b-FITC | BD | #553310 |
| Ly6C-PerCPCy5.5 | Biolegend | #128012 |
| Ly6G-PE | BD | #551461 |
| F4/80-PE-Cy7 | Biolegend | #123114 |
| CD45-APC-e780 | Invitrogen | #47045182 |
| CD3-biotin | BioLegend | #100243 |
| CD11b-BV785 | BioLegend | #101243 |
| CD11c-BV605 | BioLegend | #11733 |
| CD19-biotin | BD | #553784 |
| CD31-BV605 | BioLegend | #102427 |
| CD49b-biotin | BioLegend | #103521 |
| CD90.1-Thy1.1-PE | BioLegend | #202524 |
| F4/80-PE-Cy5 | BioLegend | #123112 |
| Ly6C-PE-Cy7 | eBioscience | #25-5932-8 |
| Ly6G-AF700 | Invitrogen | #56-9668-8 |
| SAV-PE-Fire-700 | BioLegend | #405174 |
| LIVE/DEAD™ Fixable viability dye | ThermoFisher | #L34957 |
| LIVE/DEAD™ Fixable viability dye | ThermoFisher | #L34959 |
| Fc-block (anti-CD16/32) | In-house | N/A |
| CD45.1-FITC | BioLegend | #110705 |
| CD45.2-APC | BioLegend | #109813 |
| TCRb-FITC | In-house | N/A |
| CD19-FITC | In-house | N/A |
| CD11b-BV421 | BioLegend | #101235 |
| Ly6G-AF700 | Invitrogen | #56-9667-82 |
| Zombie aqua viability dye | BioLegend | #423101 |

| Reagent/resource | Reference or source | Identifier or catalog number |
|---|---|---|
| **Recombinant DNA** | | |
| pgRNAbasic | Casaca et al, 2016 | N/A |
| pgRNA-V5-*Fth* | This study | N/A |
| **Oligonucleotides and other sequence-based reagents** | | |
| RT-qPCR primer *Arbp0* | IDT | Fwd: 5′-CTTTGGGCATCACCACGAA-3′<br>Rev: 5′-GCTGGCTCCCACCTTGTCT-3′ |
| RT-qPCR primer *MT-CO1* | IDT | Fwd 5′-TTCGGAGCCTGAGCGGGAAT-3′<br>Rev 5′-ATGCCTGCGGCTAGCACTGG-3′ |
| RT-qPCR primer *MT-CyB* | IDT | Fwd 5′-CTTAGCCATACACTACACATCAG-3′<br>Rev 5′-ATCCATAATATAAGCCTCGTCC-3′ |
| RT-qPCR primer *PolG* | IDT | Fwd 5′-GCCCCACTGTAGAATCCGCTG-3′<br>Rev 5′-AGCAGCAGGCAGAACTAGAGG-3′ |
| RT-qPCR primer *Cs* | IDT | Fwd 5′-TGGGGTGCTGCTCCAGTACTAT-3′<br>Rev 5′-AGTCTTAAAGGCCCCTGAAACAA-3′ |
| RT-qPCR primer *Nrf1* | IDT | Fwd 5′-CCAGVAAGTCCAGCAGGTCC-3′<br>Rev 5′-TTCCCTGTTGCCACAGCAGC-3′ |
| RT-qPCR primer *Nd1* | IDT | Fwd 5′-CTAGCAGAAACAAACCGGGC-3′<br>Rev 5′-CCGGCTGCGTATTCTACGTT-3′ |
| RT-qPCR primer *Hk2* | IDT | Fwd 5′-GCCAGCCTCTCCTGATTTTAGTGT-3′<br>Rev 5′-GGGAACACAAAAGACCTCTTCTGG-3′ |
| PCR primer *Fth*-Chk | IDT | Fwd 5′-GGCCGCTTCGAGCCTGAGCCC-3′<br>Rev 5′-GGTTGATCTGGCGGTTGATGG-3′ |
| V5-Fth RNA-up | IDT | 5′-AGGGCGAGGGAGACGCGGTGGTCA-3′ |
| V5-Fth RNA-down | IDT | 5′-AAACTGACCACCGCGTCTCCCTCG-3′ |
| V5-Fth replace | IDT | 5′-TGCAACTTCGTCGTTCCGCCGCTCCAGCGTCGCCACCGCGCCTCGCCCCGCCGCCACCATGACCGGCAAGCCCATCCCCAACCCCCTGCTGGGCCTGGACAGCACCACCACCGCGTCTCCCTCGCAAGTGCGCCAGAACTACCACCAGGACGCGGAGGCTGCCATCA-3′ |
| **Chemicals, enzymes and other reagents** | | |
| Collagenase D | Sigma-Aldrich | #11088866001 |

| Reagent/resource | Reference or source | Identifier or catalog number |
|---|---|---|
| DNase I | Sigma-Aldrich | #10104159001 |
| ECL western blotting substrate | ThermoFisher | #32106 |
| 4-hydroxytamoxifen | Sigma-Aldrich | #H6278 |
| Tamoxifen | Sigma-Aldrich | #T5648 |
| Corn oil | Sigma-Aldrich | #C8267 |
| Mitochondria Isolation Kit for Tissue | Thermo Fisher | #89801 |
| MEGAshortscript™ T7 Transcription kit | Thermo Fisher | #AM1354 |
| MEGAclear™ Transcription Clean-up kit | Thermo Fisher | #AM1908 |
| Meloxicam | Meloxidyl, 5 mg/ml, Ceva | GTIN 03411111913498 |
| Ketamine | Ketabel, Bela-pharm GmbH | 1346/01/20DFVPT |
| Xylazine | Rompun, Elanco, Bayer Animal Health GmbH | 440/01/12NFVPT |
| RPMI 1640 | Thermo Fisher | #61870044 |
| Pen/Strep | Gibco | #15140122 |
| FCS | Gibco | #A5256701 |
| L929 conditioned medium | This study | N/A |
| Ferric ammonium citrate | Sigma-Aldrich | 1185-57-5 |
| Propidium iodide | Life Technologies | #P3566 |
| Hoechst 33342 | Invitrogen | #H1399 |
| RNeasy Mini Kit | QIAgen | #74104 |
| RNeasy MinElute Cleanup Kit | QIAgen | #74204 |
| Transcriptor first strand cDNA synthesis kit | Roche | #04896866001 |
| Syber Green Master Mix | Applied Biosystems | #4309155 |
| Bathophenanthroline disulfonic acid | Sigma-Aldrich | #52746-49-3 |
| Buffer RLT plus | QIAgen | #1053393 |
| RNA 6000 pico kit | Agilent Technologies | #50671513 |
| Ilumina Tagment DNA Enzyme and Buffer | Illumina | #20034211 |
| KAPA HiFi HotStart ReadyMix | Roche | #07958935001 |
| Nextera XT Index Kit v2 Set A | Illumina | #15052166 |
| PFA | Alfa Aesar | #043368-9 M |
| Low-melting point agarose | Invitrogen | #15517-014 |
| Mowiol mounting medium | Merck | #81381-50 G |
| DAPI | | |

| Reagent/resource | Reference or source | Identifier or catalog number |
|---|---|---|
| Glutaraldehyde | Polysciences | #NC9072609 |
| Formaldehyde | EMS | #15700 |
| Osmium tetroxide | EMS | #19104 |
| Potassium ferrocyanide | Sigma-Aldrich | #702587 |
| Tannic acid | EMS | #21700 |
| EPON resin | EMS | #14910 |
| Phenol:Chloroform:Isoamyl Alcohol (25:24:1, v/v) | Sigma-Aldrich | #77617 |
| **Software** | | |
| Fiji software (ImageJ). | https://imagej.net/ij/index.html | v.1.54 f |
| Primer Blast | Ye et al, 2012 | N/A |
| FlowJo software | FlowJo, LLC | v.10.3 |
| bcl2fastq | Illumina | v.2.19.1.403 |
| STAR alignment | Dobin et al, 2013 | v.2.5.2a |
| *FeatureCounts* | Liao et al, 2014 | v.1.5.0-p1 |
| R | https://www.r-project.org/ | v.4.1.0 |
| R Studio Desktop | https://posit.co/download/rstudio-desktop/ | v.2024.12.0 + 467 |
| DESeq2 R package | Love et al, 2014 | v.1.32 |
| *ashr* algorithm | Stephens, 2016 | v.2.2–47 |
| biomaRt R package | Durinck et al, 2005, Durinck et al, 2009 | v.2.48.2 |
| ggplot2 R package | Wickham, 2016 | v.3.3.5 |
| gprofiler2 R package | Kolberg et al, 2020 | v.0.2.1 |
| Cytoscape | Shannon et al, 2003 | v.3.9.0 |
| FLIR Tools software | FLIR Systems | v.6.4 |
| Macro Interpreter | Sable Systems | v.2.41 |
| Labscribe2 | iWorx Systems | v2 |
| **Other** | | |
| Pressure-volume conductance catheter | Scisense | FTS-1912B-8018 |
| pressure-volume control unit | Scisense | FV896B |
| Cytation 5 | Agilent/BioTek | N/A |
| Hamamatsu Aequoria | Hamamatsu | N/A |

| Reagent/resource | Reference or source | Identifier or catalog number |
|---|---|---|
| Rodent Thermometer | Bioseb | BIO-TK8851 |
| Silk suture 3-0 | Mersilk | W212 |
| Ultra low-adherence T75 cell culture flasks | Corning | #734-4139 |
| Leica DMLB2 microscope | Leica | |
| NanoZoomer-SQ Digital slide scanner | Hamamatsu | |
| BD LSRFortessa X-20 | BD Biosciences | |
| Aurora spectral flow cytometer | Biotek | |
| CyanADP | Beckman Coulter | |
| BD FACSAria II | BD Biosciences | |
| ABI QuantStudio - 384 Real-Time PCR System | Applied Biosystems | |
| Agilent Bioanalyzer 2100 | Agilent Technologies | |
| NextSeq500 | Illumina | |
| FLIR E96 Compact-Infrared-Thermal-Imaging-Camera | FLIR Systems | |
| Promethion Core metabolic cages system | Sable Systems | |
| Leica Vibratome VT 1000 S | Leica | |
| Leica SP5 confocal | Leica | |
| Leica DM6000 inverted microscope | Leica | |
| PELCO BioWave Microwave Processor | PELCO | |
| Leica UC7ultramicrotome | Leica | |
| FEI Tecnai G2 Spirit electron microscope | BioTWIN | |
| Qiagen TissueLyser II | Qiagen | |

## Animals

Mice were bred and maintained under specific pathogen-free (SPF) conditions at the Gulbenkian Institute for Molecular Medicine (GIMM). For strains used, please see "Reagents and Tools Table". All experimental protocols were approved by the Ethics Committee of the IGC, the "Órgão Responsável pelo Bem-estar dos Animais" (ORBEA) (license A009/2011 and A006-2022) and the Portuguese National Entity (Direcção Geral de Alimentação e Veterinária). Experimental procedures were performed according to the Portuguese (Portaria no. 1005/92, Decreto-Lei no. 113/2013 and Decreto-lei no. 1/2019) and European (Directive 2010/63/EU) legislations, concerning housing, husbandry, and animal welfare. $R26^{CreERT2}Fth^{\Delta/\Delta}$; $LysM^{Cre}Fth^{\Delta/\Delta}$; $CD2^{Cre}Fth^{\Delta/\Delta}$ and $Cx3CR1^{Cre}Fth^{\Delta/\Delta}$ mice were generated by crossing C57BL/6 $Fth^{fl/fl}$ mice obtained from Prof. Lukas Kuhn (ETH, Switzerland) with C57BL/6 $R26^{CreERT2}$, $LysM^{Cre}$, $CD2^{Cre}$ and $Cx3cr1^{Cre}$ mice, respectively.

$LysM^{Cre}Tfrc^{\Delta/\Delta}$ and $LysM^{Cre}Slc40a1^{fl/fl}$ mice were generated by crossing B6.129S $Tfrc^{fl/fl}$ and C57BL/6 $Slc40a1^{fl/fl}$ mice with $LysM^{Cre}$ mice, respectively. $R26^{tdTomato/CreERT2}$ and $LysM^{Cre}$ $R26^{tdTomato}$ mice were generated by crossing $R26^{tdTomato}$ mice with $R26^{CreERT2}$ and $LysM^{Cre}$ mice, respectively. $R26^{tdTomato/CreERT2}Fth^{\Delta/\Delta}$ and $LysM^{Cre}$ $R26^{tdTomato}Fth^{\Delta/\Delta}$ mice were generated by further crossing $R26^{tdTomato/CreERT2}$ and $LysM^{Cre}$ $R26^{tdTomato}$ with C57BL/6 $Fth^{fl/fl}$ mice. $Fth^{V5}$ mice contain an allele encoding the FTH protein fused to a V5 epitope at its N-terminal end, just downstream of its natural ATG. This allele was generated through CRISPR/Cas 9-mediated homologous recombination. The sgRNA was generated by in vitro transcription from the "gRNA-V5-$Fth$" plasmid. This plasmid was built by hybridizing oligonucleotides "V5-$Fth$-gRNA-up" and "V5-$Fth$-gRNA-down" (see "Reagents and Tools Table") and introduced into the $BbsI$ sites of the pgRNA-basic plasmid (Casaca et al, 2016). The gRNA-V5-$Fth$ plasmid was linearized and transcribed with T7 RNA polymerase using the MEGAshortscript™ T7 Transcription kit (Thermo Fisher cat. #AM1354) and purified with the MEGAclear™ Transcription Clean-up kit (Thermo Fisher cat. #AM1908). The replacement single stranded DNA oligonucleotide "V5-$Fth$ replace" (see "Reagents and Tools Table") containing the V5 coding region flanked by genomic 60 nucleotide homology sequences was obtained from IDT. To generate the $Fth^{V5}$ mice, a mix containing V5-$Fth$-sgRNA (10 ng/µl), $Cas\,9$ mRNA (10 ng/µl), and the V5-$Fth$ replace oligo (10 ng/µl) was introduced into fertilized C57BL/6 mouse oocytes by pronuclear microinjection. Identification of the recombinant allele was done by PCR on genomic DNA purified from tail biopsies using the oligonucleotide pair "$Fth$-Chk-Fwd" and "$Fth$-Chk-Rev" (see "Reagents and Tools Table") that amplifies the relevant genomic area from both the wild type and the V5-tagged alleles (160 and 205 bps, respectively). The PCR fragments were separated by electrophoresis in a 15% polyacrylamide/TBE gel, the band corresponding to the V5-containing allele recovered, and its sequence confirmed. $R26^{PhAM/CreERT2}$ and $LysM^{Cre}R26^{PhAM}$ mice were generated by crossing $R26^{PhAM}$ mice with $R26^{CreERT2}$ and $LysM^{Cre}$ mice, respectively. $R26^{PhAM/CreERT2}Fth^{\Delta/\Delta}$ and $LysM^{Cre}R26^{PhAM}Fth^{\Delta/\Delta}$ mice were generated by further crossing $R26^{PhAM/CreERT2}$ and $LysM^{Cre}R26^{PhAM}$ mice with C57BL/6 $Fth^{fl/fl}$ mice.

## Bone marrow chimeras

All bone marrow chimera combinations were generated by lethally irradiating (8.5 Gy) recipient mice and reconstituted 4 h post-irradiation with freshly isolated or cryo-preserved bone marrow cells isolated from donor mice (retroorbital injection of $2–3 \times 10^6$ cells in 100 µL RPMI). Successful hematopoietic cell reconstitution was confirmed 6–8 weeks after bone marrow transfer via flow cytometry using differential CD45.1/CD45.2 haplotype markers and corresponding to cells derived from donor or recipient hematopoietic progenitors, respectively, or by concomitant lethal irradiation of control mice that were not reconstituted with bone marrow cells.

## Tamoxifen treatment

Conditional $Fth^{fl/fl}$ deletion ($\Delta$) in bone marrow chimeras generated with either $R26^{CreERT2}Fth^{\Delta/\Delta}$ recipient and/or donor mice was induced at 6–10 weeks post-transplantation by oral gavage of tamoxifen (Sigma-Aldrich, Cat. #T5648; 50 mg/kg body weight in 100 µL Corn Oil (Sigma, Cat. #C8267)/5% EtOH; 3× every other day). Conditional deletion of the $Fth^{fl/fl}$ allele in $R26^{CreERT2}Fth^{fl/fl}$ mice was achieved via oral gavage with tamoxifen, as described above with the following change in dosage: 225 mg/kg in 100 µL corn oil /5% ethanol; 3x every second day. Body weight and temperature (Rodent Thermometer BIO-TK8851, Bioseb, France) were monitored from daily to once a week and further downstream analyses were performed when the body weight loss of $Fth^{R26\Delta/\Delta}\Rightarrow Fth^{R26\Delta/\Delta}$ or $Fth^{LysM\Delta/\Delta}\Rightarrow Fth^{R26\Delta/\Delta}$ bone marrow chimeras was 10% or higher.

## Parabiosis

Parabiosis was performed essentially as described (Kamran et al, 2013). Briefly, age and weight-matched male mice were cohoused at least 2 weeks before surgery to ensure harmonious cohabitation. Mice were injected with Meloxicam (2 mg/kg, subcutaneous, s.c.) 30 min prior to surgery. Mice were anaesthetized (intraperitoneal, i.p.) using ketamine (75 mg/kg) and xylazine (15 mg/kg) (~ 140 µL/ mouse, 1:1 vol/vol in sterile 0.9% saline) and placed on a heating pad. The corresponding lateral body parts were shaved and disinfected with Betadine® solution. A longitudinal skin incision was made starting at 0.5 cm above the elbow to 0.5 cm below the knee joint, and the subcutaneous fascia was bluntly dissected to create about 0.5 cm of free skin. The corresponding elbow and knee joints were sutured together using a silk suture (3-0 Mersilk #W212) and the corresponding dorsal and ventral skin were attached using continuous sutures (5-0 Vicryl). Mice were resuscitated with 0.9% saline solution (1 mL, s.c.) and placed on a heating pad (30 min–2 h) until recovery from anesthesia. Following recovery, each parabiotic pair was placed in a clean cage and provided with free access to food and water by placing hydrogel or food pellets on the bottom of the cage. Mice were injected with analgesics buprenorphine (0.1 mg/kg s.c.) every 12 h for 48 h and Meloxicam (1 mg/kg s.c.) every 24 h for 48 h. Mice were monitored for signs of pain, distress and weight loss for 6–8 weeks until at least 80% of the combined original weight was recovered. Tamoxifen food was given for 10 days after which it was replaced by normal food. The survival and weight loss of the parabiotic mouse pairs were monitored for 100 days.

## Adoptive cell transfer

Adoptive transfer of bone marrow-derived monocytes (BMDM) was performed essentially as described (Wagner et al, 2014). Briefly, BMDM cells were generated from both $Fth$-competent ($tdT^{R26}$) and $Fth$-deleted ($tdT^{R26}Fth^{R26fl/fl}$). To this end, bone marrow was freshly isolated from the tibia and femurs of $tdT^{R26}$ and $tdT^{R26}Fth^{R2fl/fl}$ mice and placed in culture for 7 days in Ultra low-adherence T75 cell culture flasks (Corning cat. #734-4139) with BMDM differentiation culture medium (RPMI, 10%FCS, 1%Pen/Strep, supplemented with 10% L929 culture supernatant containing M-CSF1). On culture day 5, 4-hydroxytamoxifen was added to the medium (1 µM) to induce the CreERT2-mediated deletion of $Fth$. Concomitantly, $Fth^{fl/fl}$ and $Fth^{R26fl/fl}$ mice were treated with tamoxifen as described above to induce the deletion of $Fth$. Differentiated BMDM were collected from cell culture flasks, washed in RPMI (10 ml; 300 g, 5 min, 4 °C) and resuspended in RPMI (final concentration: $20–50 \times 10^6$ cells/ ml). On days 4, 8, 12 and 15 post-tamoxifen treatment, 100 µL of

BMDM cell suspension was injected retroorbitally in each mouse ($2$–$5 \times 10^6$ cells/mouse/injection).

## Real-time imaging of cell death

Cell death kinetics was monitored using a Cytation 5 (Agilent/BioTek) live-cell analysis system in primary mouse BMDMs, that were generated as described before (Martins et al, 2016). Briefly, bone marrow cells were freshly collected from tibia and femur from $Fth^{fl/fl}$ and $Fth^{R26fl/fl}$ mice and differentiated in tissue culture dishes containing RPMI-1640 with 10% FCS, pen/strep and 10% L929 conditioned medium for 6 days. On culture day 5, 4-hydroxytamoxifen was added to the medium (1 µM) to induce the CreERT2-mediated deletion of $Fth$. BMDM were seeded at a density of $1.25 \times 10^5$ cells/well in 96-well tissue culture plates and treated with either vehicle, Fe (125 µM ferric ammonium citrate; FAC) or mitochondria. Mitochondria were purified from the liver of $Fth^{R26\Delta/\Delta}$ mice (day 6 post-tamoxifen administration), using Mitochondria Isolation Kit for Tissue (ThermoFisher) according to the manufacturer's protocol. For cells treated with mitochondria, each well was given mitochondria purified from 60 mg of liver. Cells were stained with 100 nM of Hoechst for 30 min prior to stimulation. Cell death was measured by propidium iodide (PI; P3566, Life Technologies) incorporation following the manufacturer's protocol. The plate was scanned for the indicated time points where fluorescent and brightfield images were acquired in real-time every 3 h (technical replicates $n = 3$), and nine images per technical replicate and per time point were taken. Dead cells were identified as Hoechst and PI-positive and quantified using the software Gen5 (Agilent/BioTek).

## qRT-PCR

RNA was isolated from organs using RNeasy Mini Kit (QIAGEN). Briefly, mice were euthanized and transcardially perfused with ice cold PBS. Organs were collected into Eppendorf tubes and snap frozen in liquid nitrogen. RNA was isolated and processed according to the manufacturer instructions. cDNA was transcribed from total RNA with transcriptor first strand cDNA synthesis kit (Roche). Quantitative real-time PCR (qRT-PCR) was performed using 1 µg cDNA and Syber Green Master Mix (Applied Biosystems, Foster City, CA, USA) in duplicate on an ABI QuantStudio - 384 Real-Time PCR System (Applied Biosystems) under the following conditions: 95 °C/10 min, 40 cycles/95 °C/15 s, annealing at 60 °C/30 s and elongation 72 °C/30 s. Primers were designed using Primer Blast (Ye et al, 2012). Primer sequences are available in "Reagents and Tools Table".

## Cardiovascular function

Cardiovascular function was measured using pressure–volume conductance catheter technique (Pacher et al, 2008). Briefly, 7 days post-tamoxifen treatment as described above, $Fth^{fl/fl}$, $R26^{CreERT2}$ and $R26^{CreERT2}Fth^{\Delta/\Delta}$ mice were anesthetized with isoflurane, tracheotomized and artificially ventilated. Temperature was recorded continuously and kept stable. The apex of the left ventricle was punctured with a 27 G needle using the open chest approach and a pressure-volume conductance catheter (FTS-1912B-8018; Scisense, London, Canada) was inserted in the left ventricle. Mice were

stabilized for 3–10 min. Baseline values, values with varying preload caused by inferior vena cava clamps using a blunt forceps and aortic pressures, were recorded with the Scisense pressure-volume control unit FV896B and analyzed using the Labscribe2 (Labscribe, iWorx Systems, USA) software. The machine was calibrated with internal and cuvette calibration, as described (Pacher et al, 2008). Bone marrow chimeras were monitored following $Fth$ deletion and analyzed when body temperature dropped below 32 °C.

## In vivo luciferase assay

Following deletion of $Fth$ via Tamoxifen treatment as described above bone marrow chimeric mice on $OKD48^{Luc}$ genetic background were monitored daily for luciferase activity. Mice were anesthetized using intraperitoneal Ketamine/Xylasine injection. The abdomen was shaved, and mice received an intravenous injection of luciferin (2 mg/mouse in 100 µL PBS). Luciferase signal was acquired in a Hamamatsu Aequoria using an electron multiplying CCD (EMCCD) camera with highest sensitivity (255) and maximum gain (5) for 10 s, 30 s, 60 s, 120 s and 240 s. Quantification was performed using Fiji software (ImageJ).

## Iron quantification in organs and plasma

Non-heme iron concentration in heart and livers of bone marrow chimeras was assessed as previously described (Martins et al, 2016). Briefly, liver or heart were homogenized 1:5 in PBS and 100 µL (equivalent to 20 mg tissue) of homogenate or 20 µL plasma were hydrolyzed (65 °C; overnight) by adding 50 µL of 26% HCl, 1.8 M trichloroacetic acid. The hydrolyzed samples were then clarified by centrifugation ($3000 \times g$; RT). Clarified samples (60 µL) were transferred to a 96-well plate and mixed with 160 µL of 3.8 M sodium acetate, 575 µM bathophenanthroline disulfonic acid and 2.5 mM ascorbic acid. Samples were incubated (5 min; RT) and absorbance was measured at $\lambda_{540nm}$ and iron concentration was calculated using $[Fe] = (((A_s - A_b) \times V \times MW))/((e \times l \times t))$, where $A_s$ is the sample absorbance, $A_b$ is the blank absorbance, $V$ is the reaction volume (0.22), $MW$ is the molecular weight of iron (56 g·mol$^{-1}$), $e$ is the millimolar absorptivity of bathophenanthroline disulfonic acid (22.14 mM$^{-1}$·cm$^{-1}$), $l$ is the path length (0.6 cm) and $t$ is the weight of tissue used. In addition, measured concentrations were confirmed using standard iron serial dilutions.

## Western blot

Proteins were extracted, electrophoresed, and transferred essentially as described (Blankenhaus et al, 2019). Briefly, organs were collected from mice following euthanasia and perfusion with ice cold PBS (20 mL) and snap frozen in liquid nitrogen. For protein extraction tissue was homogenized in RIPA buffer using a tissue douncer kit, sonicated and centrifuged. Supernatant was collected and total protein was quantified using Bradford assay. Anti-mouse-FTH1 (clone D1D4; 1:1000; Cell Signaling cat. #4393), anti-V5-HRP (1:5000; Invitrogen cat. #46-0708), Anti-GAPDH (1:5000; SICGEN cat. # AB0049-200) and anti- β-actin (1:5000; Sigma cat. #A5441) were detected using peroxidase conjugated secondary antibodies (HRP-conjugated anti-rabbit IgG—Santa Cruz Biotechnology sc-2030; HRP-conjugated anti-mouse IgG— SantaCruz Biotechnology sc-2005; HRP-conjugated anti-goat IgG—Thermo

Fisher PA1-28664; 1:5000; 1 h; RT) and developed with ECL western blotting substrate (ThermoFisher Scientific). Detection of proteins of the ETC by Western blot was performed using OxPhos Rodent WB Antibody Cocktail (ThermoFisher cat. #45-8099; 1:1000; 4 °C overnight), followed by washing and incubation with secondary HRP-conjugated anti-mouse IgG antibody (1:2000, RT; Cell Signalling cat. #7076P2). WB quantification was performed using FiJi (ImageJ) or ImageLab Software (BioRad).

## Serology

Bone marrow chimeras were euthanized using $CO_2$ inhalation and blood was collected by cardiac puncture and placed in heparin tubes. Blood samples were sent to DNAtech (Clinical and veterinary analysis laboratory, Lisbon) for analysis of serologic parameters including ALT, AST, Urea, CPK, Troponin I and LDH as well as Transferrin and Transferrin saturation.

## Histology

Organs were harvested, fixed in 10% formalin, embedded in paraffin, sectioned into 3 µm-thick sections and stained with Hematoxylin and Eosin (H&E). Whole sections were analyzed and images acquired with a Leica DMLB2 microscope (Leica) and NanoZoomer-SQ Digital slide scanner (Hamamatsu). For WAT adipocytes area and BAT lipid droplets measurements, H&E-stained paraffin-embedded sections (3 µm sections) were scanned into digital images (NanoZoomer-SQ Digital slide scanner -Hamamatsu). The average WAT adipocyte size in adipose tissue sections (expressed as the mean cross-sectional area per cell ($\mu m^2$)) was determined using Fiji software, as described elsewhere (Blankenhaus et al, 2019). Briefly, a slide scanned picture was captured at 2.5x magnification. An average of 1500 adipocytes were measured per sample. The following macro was applied: run ("Set Scale…", "distance=560 known=250 pixel=1 unit=um global"); run ("Duplicate…", " "); run ("Subtract Background…", "rolling=50 light separate sliding"); run ("Despeckle"); run ("8-bit"); setAuto-Threshold("Mean dark"); //run ("Threshold…"); //setThreshold(250, 255); setOption("BlackBackground", false); run ("Convert to Mask"); run ("Make Binary"); run ("Dilate"); run ("Close-"); run ("Invert"); run ("Analyze Particles…", "size=330–15,000 circularity=0.50–1.00 display exclude clear summarize add"). The average BAT lipid droplet size (expressed as the mean cross-sectional area per lipid droplet (µm2)) were quantified in three non-overlapping ×40 magnification fields for each mouse. The following macro was applied: run ("Set Scale…", "distance=452 known=100 pixel=1 unit=um global"); run ("Duplicate…", " "); run ("Subtract Background…", "rolling=50 light separate sliding"); run ("Despeckle"); run ("8-bit"); setAutoThreshold("Mean dark"); //run ("Threshold…"); //setThreshold(245, 255); setOption("Black-Background", false); run ("Convert to Mask"); run ("Make Binary"); run ("Dilate"); run ("Close-"); run ("Invert"); run 25 ("Watershed"); run ("Analyze Particles…", "size=1-5000 circularity=0.4-1.00 display exclude clear summarize add").

## Flow cytometry

Reconstitution of bone marrow chimeras was tested by staining peripheral blood leukocytes with CD45.1-FITC (BioLegend cat. #110705), CD45.2-APC (BioLegend cat. #109813), to determine the relative contribution of donor bone marrow towards engraftment. To quantify the numbers of tissue leukocytes, animals were transcardially perfused with 10 mL cold PBS. Briefly, liver, kidney, lung and heart were cut into small pieces, digested in 10 mL digestion medium (HBSS supplemented with 1 mg/mL Collagenase D (Sigma-Aldrich cat. #11088866001) and 10 µg/mL DNase I (Sigma-Aldrich cat. #10104159001)), shaking at 220 rpm, 37 °C for 45 min. The digested solution was passed through a cell strainer (100 µm) and washed with 5 mL of RPMI. Cells were pelleted by centrifugation (300× *g*; 4 °C; 10 min) and resuspended in 5 mL ACK red blood cell (RBC) lysis buffer. After 5 min at RT RBC lysis was stopped by adding 5 mL of FACS buffer (1× PBS 3% FCS) and cells were passed through a 40 µm cell strainer. Cells were again centrifuged (300× *g*; 4 °C; 10 min) and finally resuspended in 3 mL (liver and kidney) or 1 mL (heart and lung) FACS buffer. In total, 300 µL of cell suspension were used for the staining of leukocytes with the following antibodies: Cx3CR1-APC (Biolegend, cat. #149007), CD11b-FITC (BD, cat. #553310), Ly6C-PerCPCy5.5 (Biolegend, cat. #128012), Ly6G-PE (BD, cat. #551461), F4/80-PE-Cy7 (Biolegend, cat. #123114), CD45-APC-e780 (Invitrogen, cat. #47045182), CD3-biotin (BioLegend, cat. #100243), CD11b-BV785 (BioLegend, cat. #101243), CD11c-BV605 (BioLegend, cat. #11733), CD19-biotin (BD, cat. #553784), CD31-BV605 (BioLegend, cat. #102427), CD49b-biotin (BioLegend, cat. #103521), CD90.1-Thy1.1-PE (BioLegend, cat. #202524) F4/80-PE-Cy5 (BioLegend, cat. #123112) Ly6C-PE-Cy7 (eBioscience, cat. #25-5932-8), Ly6G-AF700 (Invitrogen, cat. #56-9668-8), and SAV-PE-Fire-700 (Bio-Legend, cat. #405174). In addition, Fc-block (in-house anti-CD16/32) was used to minimize unspecific Ig binding and LIVE/DEAD™ Fixable viability dye (ThermoFisher cat. #L34957), and LIVE/DEAD™ Fixable Yellow viability dye (ThermoFisher cat. #L34959) were used to assess cell viability. Data acquisition was performed using either CyanADP (Beckman Coulter), BD LSRFortessa X-20 (BD Biosciences) or Aurora (Cytek) flow cytometers. Samples were analyzed using FlowJo software.

## Cell sorting

Monocyte/macrophages were sorted from the liver, WAT and heart of bone marrow chimeras, as indicated in Fig. EV6C. Briefly, *Fth* deletion was induced via tamoxifen treatment as described above, and 19 days later, bone marrow chimeras were euthanized, and organs were collected. Liver, and heart were cut into small pieces and incubated with digestion buffer (liver: 4 ml; heart: 1 ml; 0.2 mg/mL Liberase (Roche cat. #5401127001) and 100 µg/mL DNAse I (Sigma-Aldrich cat. #10104159001). After digestion, cell suspensions passed through a 40 µm cell strainer. Liver and heart cells were then pelleted (350 g, 5 min, 4 °C) and resuspended in 40% Percoll (GE Healthcare cat. #10607095) in RPMI (liver: 15 ml, heart: 6 ml). A density gradient was made by overlaying the cell suspension onto 80% Percoll in RPMI (liver: 5 ml, heart: 2 ml), and centrifuged (700 g no brake/lowest acceleration, 20 min RT). Liver and heart cells were collected from the density interface and washed once with FACS buffer (10–15 ml; 1× PBS 3%FCS 2 mM EDTA) and centrifuged (350× *g*, 5 min 4 °C). Plate for antibody staining. WAT was cut into small pieces and placed in 3 mL WAT digestion buffer (4 mg/mL Collagenase IV, 10 mM $CaCl_2$, 0.5% BSA, in PBS without $Ca^{2+}/Mg^{2+}$). After digestion, WAT cell suspensions were

passed through a 100 µm cell strainer and centrifuged ($500\times g$, 10 min, 4 °C). Liver and heart cell pellets were resuspended in 5 mL FACS buffer, and WAT cell pellets were resuspended in 0.25-1 mL FACS buffer. Cells were counted using trypan blue and $1–2 \times 10^6$ cells were placed on a 96-well. Cells were centrifuged ($700\times g$, 2 min 4 °C), resuspended in 150 µL of ACK buffer for red blood cell lysis and incubated (5 min, RT). Lysis was stopped by adding 50 µL FACS buffer and cells were washed ($700\times g$, 2 min 4 °C). Cells were resuspended in 50 µL FACS buffer and antibody staining was performed using: CD11b-BV421 (BioLegend, cat. #101235), Ly6G-AF700 (Invitrogen, cat. #56-9668-82), CD45.2-APC (BioLegend cat. #109813), TCRb-FITC (In-house) and CD19-FITC (In-house). In addition, Fc-block (in-house anti-CD16/32) was used to minimize unspecific Ig binding and Zombie aqua viability dye (BioLegend cat. #423101) was used to assess cell viability. Cell sorting and data acquisition was performed using BD FACSAria II (BD Biosciences) cell sorter. 500 cells/sample were sorted directly into 0.2 mL Eppendorf tubes containing 2.5 µL Buffer RLT plus (Qiagen, cat. #1053393). Flow cytometry data were analyzed using FlowJo software.

## Bulk RNA sequencing

RNA was extracted, cleaned (RNeasy MinElute Cleanup Kit, Qiagen) and its quality assessed using an Agilent Bioanalyzer 2100 (Agilent Technologies) together with an RNA 6000 pico kit (Agilent Technologies). Full-length cDNAs and sequencing libraries were generated according to the SMART-Seq2 protocol, as previously described (Ramos et al, 2022). Library preparation including cDNA 'tagmentation', PCR-mediated adaptor addition and amplification of the adapted libraries was done following the Nextera library preparation protocol (Illumina Tagment DNA Enzyme and Buffer, Illumina #20034211; KAPA HiFi HotStart ReadyMix, Roche #07958935001; Nextera XT Index Kit v2 Set A, Illumina #15052163; Nextera XT Index Kit v2 Set D, Illumina #15052166), as previously described (Ramos et al, 2022). Libraries were sequenced (NextSeq500 sequencing; Illumina) using 75 SE high throughput kit. Sequence information was extracted in FastQ format, using Illumina's bcl2fastq v.2.19.1.403, producing on average $\sim38 \times 10^6$ (liver), $32 \times 10^6$ (WAT) and $43 \times 10^6$ (heart) reads per sample. Library preparation and sequencing were optimized and performed at the Gulbenkian Institute for Molecular Medicine Genomics Unit.

FastQ reads were aligned against the mouse reference genome GRCm39 using the GENCODE vM27 annotation to extract splice junction information (STAR; v.2.5.2a) (Dobin et al, 2013). Read summarization was performed by assigning uniquely mapped reads to genomic features using *FeatureCounts* (v.1.5.0-p1) (Liao et al, 2014). Gene expression tables were imported into the R programming language and environment (v.4.1.0) to perform differential gene expression and functional enrichment analyses, as well as data visualization.

Differential gene expression was performed using the DESeq2 R package (v.1.32) (Love et al, 2014). Gene expression was modeled by genotype for each organ, which included the following factors: $tdT^{LysM}\Rightarrow Fth^{fl/fl}$, $tdT^{LysM}Fth^{LysM\Delta/\Delta}\Rightarrow Fth^{fl/fl}$ or $tdT^{LysM}Fth^{LysM\Delta/\Delta}\Rightarrow Fth^{R26\Delta/\Delta}$ bone marrow chimeras. Genes not expressed or with fewer than 10 counts across the samples were removed, leaving 22,529 (liver), 23,327 (WAT) and 29,821 (heart) genes for downstream differential gene expression analysis. We subsequently ran the function *DESeq* to estimate the size factors (by *estimateSizeFactors*), dispersion (by

*estimateDispersions*) and fit a binomial GLM fitting for βi coefficient and Wald statistics (by *nbinomWaldTest*). Pairwise comparisons tested with the function *results* (alpha = 0.05), were: 1) $tdT^{LysM}\Rightarrow Fth^{fl/fl}$, vs. $tdT^{LysM}Fth^{LysM\Delta/\Delta}\Rightarrow Fth^{fl/fl}$; 2) $tdT^{LysM}\Rightarrow Fth^{fl/fl}$ vs. $tdT^{LysM}Fth^{LysM\Delta/\Delta}\Rightarrow Fth^{R26\Delta/\Delta}$ and 3) $tdT^{LysM}Fth^{LysM\Delta/\Delta}\Rightarrow Fth^{fl/fl}$ vs. $tdT^{LysM}Fth^{LysM\Delta/\Delta}\Rightarrow Fth^{R26\Delta/\Delta}$. In addition, the $\log_2$ fold change for each pairwise comparison was shrunken with the function *lfcShrink* using the algorithm *ashr* (v.2.2–47) (Stephens, 2016). Differentially expressed genes were considered for genes with an adjusted P value < 0.05 and an absolute $\log_2$ fold change >0. Normalized gene expression counts were obtained with the function *counts* using the option normalized = TRUE. Regularized log transformed gene expression counts were obtained with *rlog*, using the option blind = TRUE. Ensembl gene ids were converted into gene symbols from Ensembl (v.107) by using the mouse reference (GRCm39) database with biomaRt R package (v.2.48.2)(Durinck et al, 2005; Durinck et al, 2009). All scatterplots, including volcano plots, were done with the ggplot2 R package (v.3.3.5) (Wickham, 2016). Functional enrichment analysis was performed with the gprofiler2 R package (v.0.2.1)(Kolberg et al, 2020). Enrichment was performed using the function *gost* based on the list of up- or downregulated genes (genes with an adjusted P value < 0.05 and a $\log_2$ fold change >0 or <0), between each pairwise comparison (independently), against annotated genes (domain_scope = "annotated") of the organism *Mus musculus* (organism = "mmusculus"). Gene lists were sorted according to adjusted p value (ordered_query = TRUE) to generate GSEA (Gene Set Enrichment Analysis) style p values. Only statistically significant (user_threshold = 0.05) enriched functions are returned (significant = TRUE) after multiple testing corrections with the default method g:SCS (correction_method = "analytical"). The gprofiler2 queries were run against all the default functional databases for mouse which include: Gene Ontology (GO:MF, GO:BP, GO:CC), KEGG (KEGG), Reactome (REAC), TRANSFAC (TF), miRTarBase (MIRNA), Human phenotype ontology (HP), WikiPathways (WP), and CORUM (CORUM). For future reference, gprofiler2 was performed using database versions Ensembl 107, Ensembl genomes 54 (database updated on 12/07/2022). For STRING database network analysis, genes contained within enriched gene sets associated related to mitochondrial proteins, mitochondrial ribosome, electron transport chain, respirasome were merged and uploaded to the STRING database (v11.5) (Shannon et al, 2003) and queried for known protein-protein interactions (organism: Mus musculus; interaction score >0.4). The resulting network was imported into Cytoscape (v.3.9.0) (Shannon et al, 2003) for network layout design.

## Thermal imaging

BAT and tail temperatures were measured in mice that were allowed to move freely in a cage, using an infrared camera, (FLIR E96 Compact-Infrared-Thermal-Imaging-Camera; FLIR Systems). At least 2 days prior, mice were anesthetized (1–2% Isoflurane) and the interscapular area was shaved. Acquired images were analyzed using FLIR Tools software (v6.4) and individual BAT and tail temperatures were taken as the maximum temperature measured at the interscapular area or tail base, respectively.

## Metabolic phenotyping

Promethion Core (Sable Systems, USA) was used to measure indirect calorimetry. *Fth* deletion was induced via tamoxifen

administration as described above and 7 (early onset) or 20 (late onset) days post-tamoxifen treatment, bone marrow chimeras were placed in metabolic cages for metabolic phenotyping. Mice were kept on a 14/10 h light/dark cycle with controlled temperature and humidity. Recording continued for the following 5–6 days. The system consists of a standard GM-500 cage with a food hopper and a water bottle connected to load cells (2 mg precision) with 1 Hz rate data collection. Additionally, the cage contains a red house enrichment. Ambulatory activity was monitored at 1 Hz rate using an XY beam break array (1 cm spacing). Oxygen, carbon dioxide and water vapor were measured using a CGF unit (Sable Systems). This multiplexed system operated in pull-mode. Air flow was measured and controlled by the CGF (Sable Systems) with a set flow rate of 2 L/min. Oxygen consumption and carbon dioxide production were reported in milliliters per minute (mL/min). Energy expenditure was calculated using the Weir equation and Respiratory Exchange Ratio (RER) was calculated as the ratio of $VCO_2/VO_2$. Raw data was processed using Macro Interpreter v2.41 (Sable Systems), as described (Ramos et al, 2022).

## Fluorescence microscopy

Bone marrow chimeras were treated with tamoxifen to induce *Fth* deletion, as described above. 21 days post tamoxifen treatment, bone marrow chimeras were euthanized and transcardially perfused with 20 mL cold PBS, followed by prefusion with 10 mL 4%PFA (Alfa Aesar cat. #043368-9 M) in PBS. Livers were collected and fixed in 4%PFA in PBS overnight in the dark. Organs were then washed in PBS overnight and embedded in 4% low-melting point agarose (Invitrogen cat. #15517-014). Tissue sections (100 μm) were obtained using a Leica Vibratome VT 1000 S (Leica Biosystems) and placed in Eppendorf tubes with PBDO permeabilization solution (1% Bovine Serum Albumin; 1% DMSO; 0.6% Triton X-100 in PBS; overnight; 4 °C). Tissue slices were then incubated with primary Abs (overnight; 4 °C): anti-FTH (1:100, Cell Signalling cat. #4393), Anti-V5 (1:200, Abcam cat. # AB9137) and Anti-CD68 (1:200, BioLegend cat. #123102). Tissue slices were further washed with PBDO (overnight; 4 °C) and were then incubated (1:500; overnight; 4 °C) with secondary antibodies: Cy5-conjugated donkey anti-goat IgG antibody (Jackson IR; cat. #705-175-147), Alexa 568 conjugated anti-rabbit (Thermo-Fisher, cat. #A-11011) and Alexa 568 conjugated anti-rat (Thermo-Fisher, cat. #A-11077). Tissue slices were washed once more with PBDO (overnight, 4 °C) and mounted with Mowiol mounting medium with 10 μg/mL of DAPI (Merck, cat. #81381-50 G). Images were then acquired on a Leica SP5 confocal-based and on a Leica DM6000 inverted microscope (458, 476, 488, 514, 561 and 633 nm lasers).

## Transmission electron microscopy

Animals were euthanized and perfused with fixating media (2% formaldehyde (EMS), 2.5% glutaraldehyde (Polysciences) in 0.1 M Phosphate Buffer (PB); pH 7.4). Organs were dissected and immersed in the same primary fixative (1 h; RT). Further processing was achieved using a PELCO BioWave Microwave Processor at 23 °C, restricted by a PELCO SteadyTemp Pro. Samples were additionally fixed in the primary fixative using a time-sequence of 7 × 2 min with ON and OFF sequential cycles of 0 and 100 W irradiating power in vacuum and rinsed with PB before post-fixation in 1% (v/v) osmium tetroxide (®EMS) with 1% (w/v)

potassium ferrocyanide (®Sigma-Aldrich) in PB for 8 × 2 min, also with ON and OFF sequential cycles of 100 W in vacuum. Subsequently, samples were washed with PB and $dH_2O$ twice and immersed in 1% (w/v) tannic acid (®EMS) followed by en-bloc staining with 0.5% (w/v) uranyl acetate. Both steps were made using a time-sequence of 7 × 2 min with ON and OFF sequential cycles of 0 and 150 W irradiating power in vacuum. Between the steps, samples were rinsed with $dH_2O$. Dehydration was done in a graded ethanol series of 30%, 50%, 75%, 90% and 100%, for 40 s at 150 W each. EPON resin (®EMS), 25%, 50%, 75% and 100% was infiltrated, for 3 min at 250 W in vacuum each step, and cured overnight, at 60 °C. Sections of 70 nm were obtained on a Leica UC7 and mounted on palladium-copper grids coated with 1% (w/v) formvar (®Agar Scientific) in chloroform (®VWR). Sections were stained with 1% (w/v) uranyl acetate and Reynolds lead citrate for 5 min each and imaged on an FEI Tecnai G2 Spirit BioTWIN Transmission Electron Microscope operating at 120 keV.

## Mitochondria quantification

Livers, WAT and heart were harvested and snap frozen as described above. A piece of the tissue was cut and placed into lysis buffer (100 mM NaCl, 10 mM EDTA, 0.5% SDS and 20 mM Tris-HCl, pH 7.4) and homogenized in a Qiagen TissueLyser II using tungsten carbide beads. Upon homogenization, an equal volume of Phenol:Chloroform:Isoamyl Alcohol (25:24:1, v/v) was added. DNA, present in the aqueous phase, was precipitated using 1vol. isopropanol and 0.3 M sodium acetate for 3 h at −20 °C. The isolated DNA was used to perform the quantification of mitochondrial DNA (mtDNA) as compared to nuclear DNA (nDNA) using a qRT-PCR-based method, similar to what was previously described (Blankenhaus et al, 2019; Quiros et al, 2017). Briefly, qRT-PCR was performed using 20 ng of DNA and SYBR Green Master Mix (Applied Biosystems, Foster City, CA, USA), in duplicate on a ABI QuantStudio - 384 Real-Time PCR System (Applied Biosystems), under the following conditions: 50 °C/2 min and 95 °C/5 min (Hold stage), 45 cycles/95 °C/10 s, annealing at 60 °C/30 s, and elongation 72 °C/20 s, followed by melting curve: 95 °C for 15 s, 60 °C for 1 min, and gradual increase in temperature up to 95 °C. Primers for NADH-ubiquinone oxidoreductase chain 1 encoded by the mitochondrial gene MT-Nd1 (Nd1) and for the nuclear encoded hexokinase 2 gene (Hk2) (Blankenhaus et al, 2019; Quiros et al, 2017) are available in "Reagents and Tools Table". Mitochondria number per cell was calculated by the ratio of mRNA expression of the single copy mitochondrial gene *Nd1* and the single copy nuclear gene *Hk2*.

## Statistical analysis

Statistical analysis was conducted using GraphPad Prism v8.4.2 software. All data are displayed as means ± standard deviation of the mean (SD) unless otherwise noted. Sample sizes were estimated based on previous experience and publications, using the Power/sample-size calculator (http://www.stat.ubc.ca/~rollin/stats/ssize/n2.html). Statistical comparison between two groups was performed using either a Student's *T* test or Mann–Whitney *U* test. Groups of three or more were analyzed by one-way analysis of variance (ANOVA), Welch ANOVA or the Kruskal–Wallis test, using Tukey's range test, Dunnett's T3 test, or

Dunn's test for multiple comparison correction, respectively. Survival was assessed using a log-rank (Mantel–Cox) test. No experiment blinding was performed. Statistical outliers (ROUT Q = 1%) were excluded from analysis. Statistical parameters for each experiment can be found within the corresponding figure legends. *P* values for each statistical comparison can be found in Appendix Table S1.

## Data availability

Data from RNA sequencing studies are available at GEO database GSE292630 (https://www.ncbi.nlm.nih.gov/geo/query/acc.cgi?acc=GSE292630; Figs. 7A–D, EV6E,F, Appendix Fig. S4A–C, EV7A–D).

The source data of this paper are collected in the following database record: biostudies:S-SCDT-10_1038-S44318-025-00622-x.

## Peer review information

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

## Acknowledgements

The authors are indebted to all members of the Inflammation laboratory (GIMM) for insightful technical and intellectual contributions, to Erin M. Tranfield and Ana L. Vinagre at the Electron Microscopy Unit (GIMM), to the flow cytometry, genomics and animal facility staff at GIMM. MPS was supported by Gulbenkian, 'la Caixa' Foundation (HR18-00502) and Fundação para a Ciência e a Tecolognia (FCT, PTDC/IMI-IMU/5723/2014; FEDER/29411/2017; PTDC/MED-FSL/4681/2020; and 2022.02426.PTDC), Oeiras-

ERC Frontier Research Incentive Awards, SymbNET Research Grants (H2020-WIDESPREAD-2020-5-952537), Deutsche Forschungsgemeinschaft Cluster of Excellence "Balance of the Microverse" (DFG, EXC 2051; 390713860) and Congento (LISBOA-01-0145-FEDER-022170). RM was supported by an EMBO long-term fellowship (ALTF290-2017ARC), Marie Skłodowska-Curie Research Fellowship (MSCA-IF-EF-ST-753236) and Fundação para a Ciência e a Tecnologia (FCT, 2021.03494.CEECIND/CP1674/CT0004). BB and SC were supported in part by European Community 7th Framework 294709-DAMAGECONTROL ERC-2011-AdG to MPS, FB by Marie Skłodowska-Curie Research Fellowship (REGDAM 707998). SW was supported by the German Ministry of Education and Research (BMBF; grant 01 EO 1502) via the Jena Center of Sepsis Control and Care. MM was supported by Fundação para a Ciência e a Tecnologia (FCT, UI/BD/152257/2021). QW was supported by a Marie Skłodowska-Curie Research Fellowship (MSCA-IF2019-892773), the International Postdoctoral Exchange Fellowship Program from the People´s Republic of China (20190090), and the National Natural Science Foundation of China (32171166, 82030003).

## Author contributions

**Rui Martins**: Conceptualization; Formal analysis; Supervision; Investigation; Visualization; Methodology; Writing—original draft; Writing—review and editing. **Birte Blankenhaus**: Conceptualization; Validation; Investigation; Methodology. **Faouzi Braza**: Investigation; Methodology. **Miguel Mesquita**: Investigation; Methodology. **Pedro Ventura**: Investigation. **Sumnima Singh**: Investigation; Methodology. **Sebastian Weis**: Investigation; Methodology. **Maria Pires**: Investigation. **Sara Pagnotta**: Investigation. **Qian Wu**: Investigation. **Sílvia Cardoso**: Investigation. **Elisa Jentho**: Investigation. **Ana Figueiredo**: Investigation. **Pedro Faísca**: Investigation. **Ana Nóvoa**: Resources. **Vanessa Alexandra Morais**: Resources. **Stefanie K Wculek**: Resources. **David Sancho**: Resources. **Moises Mallo**: Resources. **Miguel P Soares**: Conceptualization; Resources; Supervision; Funding acquisition; Visualization; Writing—original draft; Project administration; Writing—review and editing.

Source data underlying figure panels in this paper may have individual authorship assigned. Where available, figure panel/source data authorship is listed in the following database record: biostudies:S-SCDT-10_1038-S44318-025-00622-x.

## Disclosure and competing interests statement

The authors declare no competing interests.

# Expanded View Figures

**Figure EV1.  Monocyte-derived macrophages are depleted in chimeric *Fth*-deleted mice.**

Schematic representation of TAM-induced *Fth* deletion in chimeric mice (day 0) and representative flow cytometry dot plots for (**A**) monocyte-derived macrophage (backgated as Ly6G$^-$, CD11b$^+$, F4/80$^{low}$) and (**B**) tissue-resident macrophage (Ly6G$^-$, CD11b$^{-/low}$, F4/80$^{high}$) populations present in the liver, heart, lungs and kidneys of *Fth*$^{fl/fl}$⇨*Fth*$^{fl/fl}$ ($n = 7$), *Fth*$^{R26\Delta/\Delta}$⇨*Fth*$^{R26\Delta/\Delta}$ ($n = 5$), *Fth*$^{fl/fl}$⇨*Fth*$^{R26\Delta/\Delta}$ ($n = 7$) and *Fth*$^{LysM\Delta/\Delta}$⇨*Fth*$^{R26\Delta/\Delta}$ ($n = 6$) chimeric mice, 7 to 19 days post-TAM administration. Data in (**A**, **B**) is pooled from 3 independent experiments with similar trends. (**C**) Absolute number of tissue-resident macrophages (Ly6G$^-$CD11b$^{-/low}$F4/80$^{high}$) in the liver, heart, lungs and kidneys of *Fth*$^{fl/fl}$⇨*Fth*$^{fl/fl}$ ($n = 7$), *Fth*$^{R26\Delta/\Delta}$⇨*Fth*$^{R26\Delta/\Delta}$ ($n = 5$), *Fth*$^{fl/fl}$⇨*Fth*$^{R26\Delta/\Delta}$ ($n = 7$) and *Fth*$^{LysM\Delta/\Delta}$⇨*Fth*$^{R26\Delta/\Delta}$ ($n = 6$) chimeric mice, 7 to 19 days post-TAM administration. Data in (**C**) presented as individual values (circles) and mean (red bars), pooled from 3 independent experiments with similar trends. (**D**) Schematic representation of TAM-induced *Fth* deletion (day 0) in chimeric mice, and survival of *Fth*$^{fl/fl}$⇨*Fth*$^{fl/fl}$ ($n = 10$), *Fth*$^{Cx3cr1\Delta/\Delta}$⇨*Fth*$^{R26\Delta/\Delta}$ ($n = 12$) and *Fth*$^{fl/fl}$⇨*Fth*$^{R26\Delta/\Delta}$ ($n = 10$) chimeric mice. Data in (**D**) pooled from 3 independent experiments with similar trends. (**E**) Schematic representation of TAM-induced *Fth* deletion in chimeric mice, and survival of *Ccr2*$^{+/+}$⇨*Fth*$^{fl/fl}$ ($n = 7$), *Ccr2*$^{-/-}$⇨*Fth*$^{R26\Delta/\Delta}$ ($n = 6$) and *Ccr2*$^{+/+}$⇨*Fth*$^{R26\Delta/\Delta}$ ($n = 15$) chimeric mice following TAM administration on day 0. Data in (**E**) is pooled from 3 independent experiments with similar trends. (**F**) Time course of relative body weight changes of *Fth*$^{fl/fl}$⇔*Fth*$^{fl/fl}$ ($n = 7$), *Fth*$^{R26\Delta/\Delta}$⇔*Fth*$^{R26\Delta/\Delta}$ ($n = 10$), *Fth*$^{fl/fl}$⇔*Fth*$^{R26\Delta/\Delta}$ ($n = 14$) and *Fth*$^{LysM\Delta/\Delta}$⇔*Fth*$^{R26\Delta/\Delta}$ ($n = 11$) parabiotic mouse pairs, following TAM administration on day 0. Data in (**F**) are presented as mean ± SD, normalized to the initial body weight (t$_0$) and pooled from 4 independent experiments with similar trends. Survival analysis was performed using Log-rank (Mantel–Cox) test. One-way ANOVA with Tukey's range test for multiple comparison correction was used for comparison between multiple groups. NS: non-significant, *$P < 0.05$, **$P < 0.01$, ***$P < 0.001$.

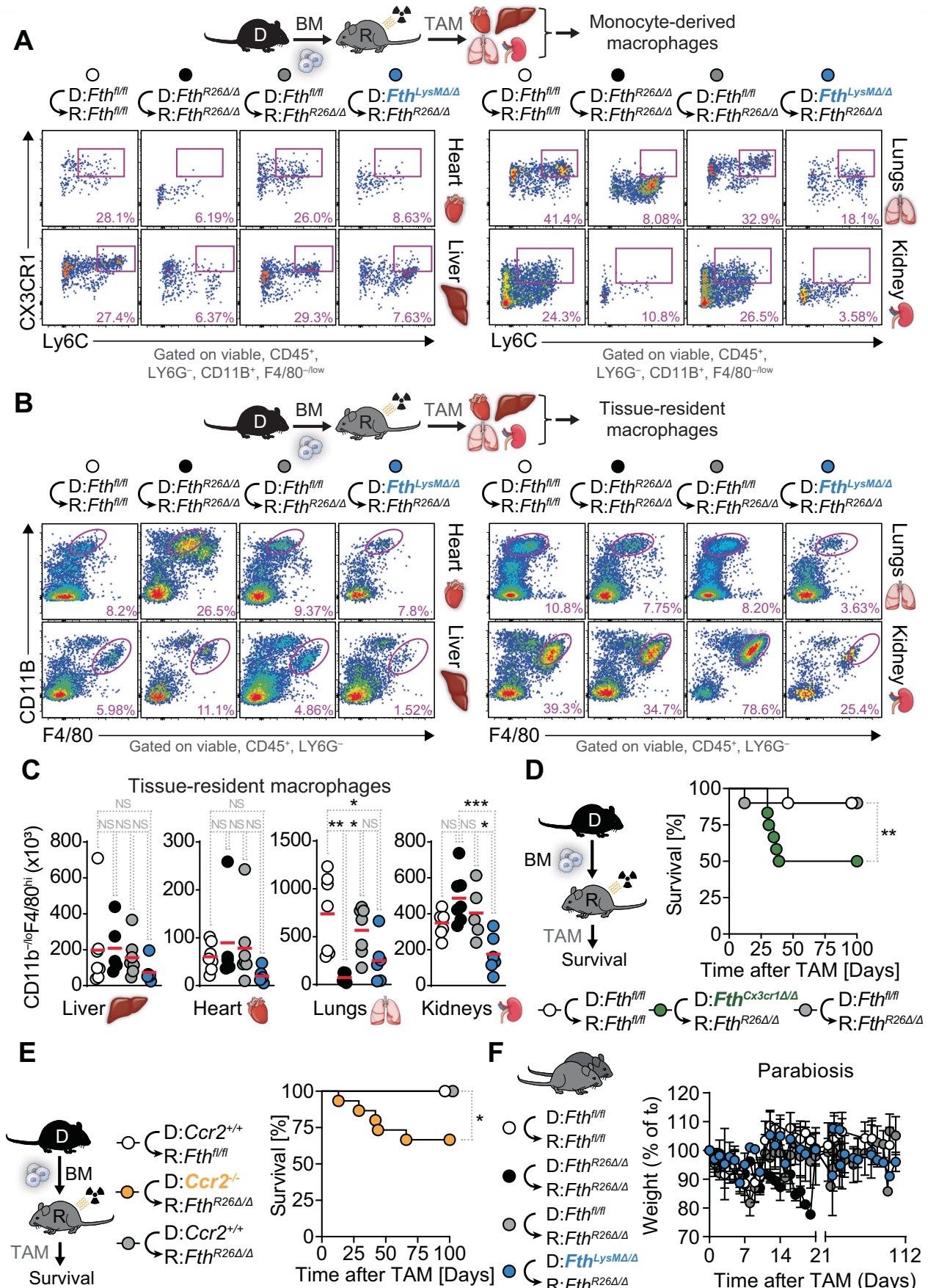

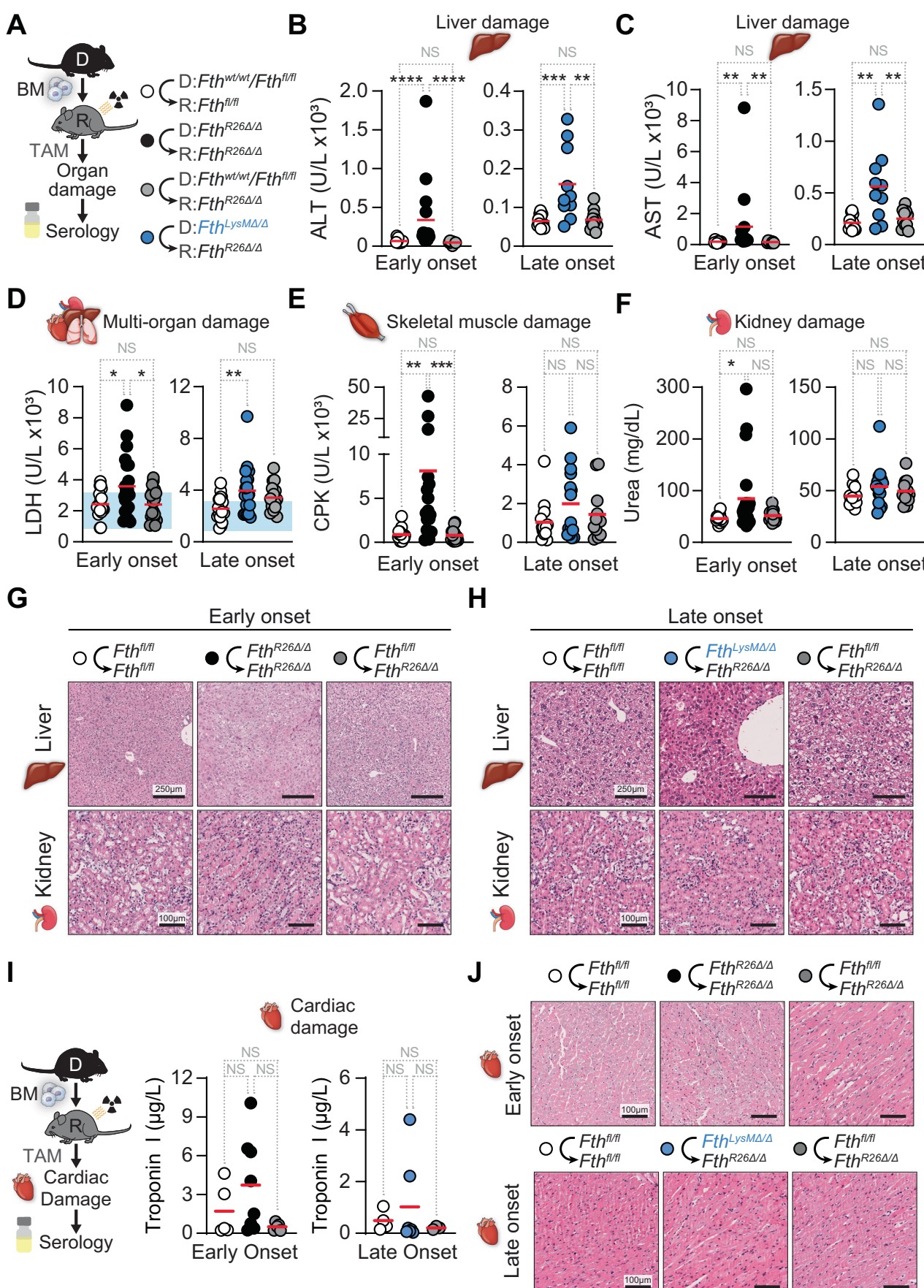

◀ **Figure EV2.** *Fth*-competent myeloid cells support tissue function in chimeric *Fth*-deleted mice.

(A) Schematic representation of chimeric mice and TAM-induced *Fth* deletion. Plasma levels of (B) alanine transaminase (ALT), (C) aspartate transaminase (AST), (D) lactate dehydrogenase (LDH), (E) creatine phosphokinase (CPK) and (F) urea, measured in $Fth^{fl/fl} \Rightarrow Fth^{fl/fl}$ ($n = 11$–21), $Fth^{R26\Delta/\Delta} \Rightarrow Fth^{R26\Delta/\Delta}$ ($n = 15$–25), $Fth^{fl/fl} \Rightarrow Fth^{R26\Delta/\Delta}$ ($n = 9$–18) and $Fth^{LysM\Delta/\Delta} \Rightarrow Fth^{R26\Delta/\Delta}$ ($n = 11$–20) chimeric mice on days 7–15 (early onset), or 19–35 (late onset) following TAM administration. Data in (B–F) represented as individual values (circles) and mean (red bars) pooled from 4 to 6 independent experiments with similar trends. (G, H) Representative hematoxylin and eosin (H&E) stained histology images of liver, kidney and heart from $Fth^{fl/fl} \Rightarrow Fth^{fl/fl}$, $Fth^{R26\Delta/\Delta} \Rightarrow Fth^{R26\Delta/\Delta}$, $Fth^{fl/fl} \Rightarrow Fth^{R26\Delta/\Delta}$ and $Fth^{LysM\Delta/\Delta} \Rightarrow Fth^{R26\Delta/\Delta}$ chimeric mice on day 7 (early onset) (G), or 22 (late onset) (H) following TAM administration. One-way ANOVA with Tukey's range test for multiple comparison correction was used for comparison between multiple groups. NS: non-significant, $^*P < 0.05$, $^{**}P < 0.01$, $^{***}P < 0.001$, $^{****}P < 0.0001$.

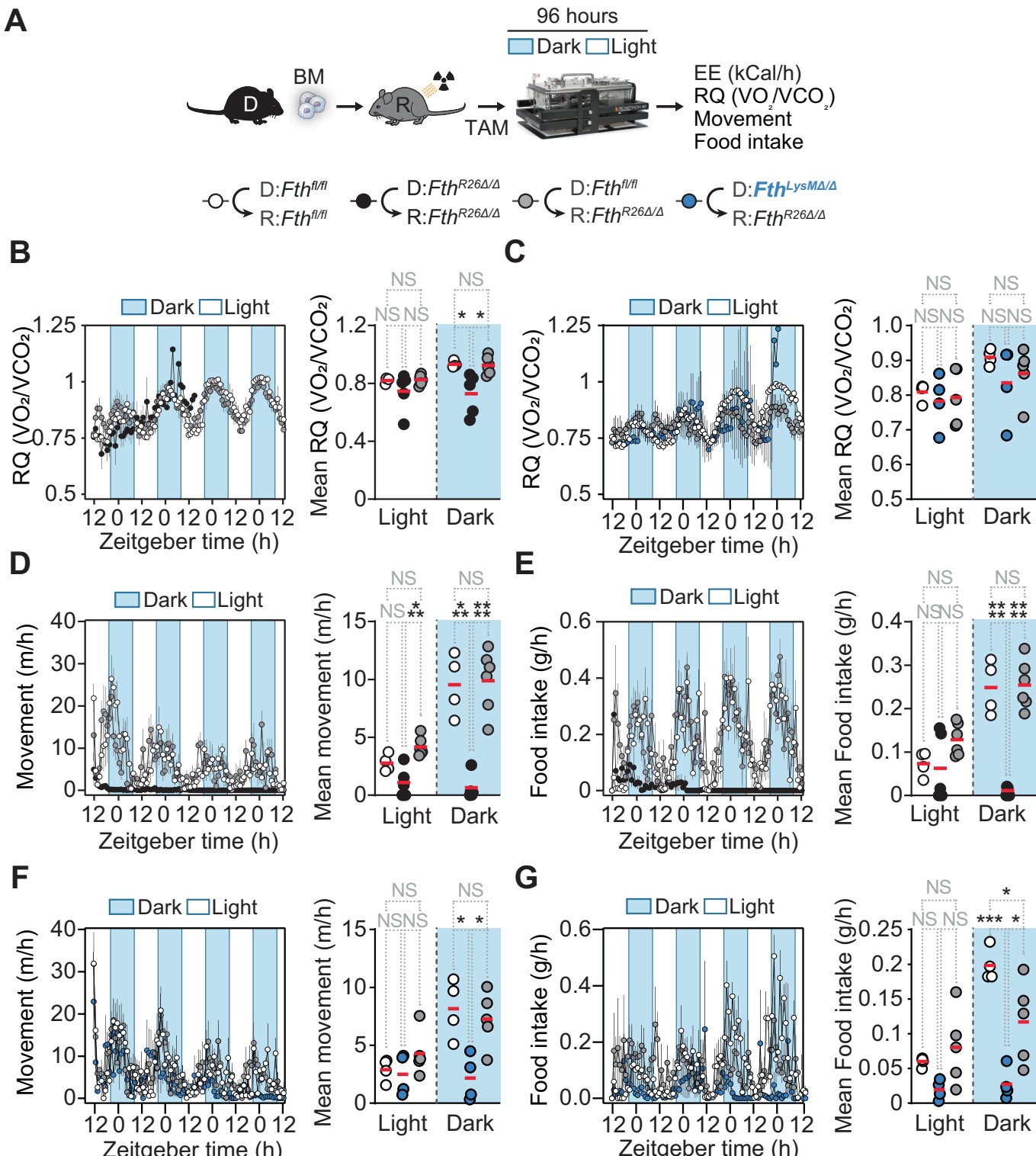

**Figure EV3. *Fth*-competent myeloid cells restore movement and food intake in chimeric *Fth*-deleted mice.**

(A) Schematic representation of TAM-induced *Fth* deletion in chimeric mice (day 0) and metabolic cage assessment of metabolic parameters. (B, C) Time course of respiratory quotient (RQ, calculated as $VO_2/VCO_2$), and mean (red bars) RQ during daytime/nighttime (dot plots) of $Fth^{fl/fl} \Rightarrow Fth^{fl/fl}$ ($n = 4$), $Fth^{R26\Delta/\Delta} \Rightarrow Fth^{R26\Delta/\Delta}$ ($n = 5$), $Fth^{fl/fl} \Rightarrow Fth^{R26\Delta/\Delta}$ ($n = 5$–6) and $Fth^{LysM\Delta/\Delta} \Rightarrow Fth^{R26\Delta/\Delta}$ ($n = 4$) chimeric mice, assessed from day 7 (B; early onset), or day 20 (C; late onset) post TAM administration. Time course and mean (red bars) of daytime/nighttime values (dot plots) for mouse movement (m/h; D), and rate of food intake (g/h; E) of $Fth^{fl/fl} \Rightarrow Fth^{fl/fl}$ ($n = 4$), $Fth^{R26\Delta/\Delta} \Rightarrow Fth^{R26\Delta/\Delta}$ ($n = 5$) and $Fth^{fl/fl} \Rightarrow Fth^{R26\Delta/\Delta}$ ($n = 6$) chimeric mice, assessed from day 7 (early onset). Time course and mean (red bars) of daytime/nighttime values (dot plots) for mouse movement (m/h; F), and rate of food intake (g/h; G) of $Fth^{fl/fl} \Rightarrow Fth^{fl/fl}$ ($n = 4$), $Fth^{LysM\Delta/\Delta} \Rightarrow Fth^{R26\Delta/\Delta}$ ($n = 4$) and $Fth^{fl/fl} \Rightarrow Fth^{R26\Delta/\Delta}$ ($n = 5$) chimeric mice, assessed from day 20 (late onset). Data in (B–G) is displayed as mean ± SD (time course), or as individual values (circles) and mean (red bars) (dot plots). Data in (B–G) is pooled from 2 independent experiments with similar trend. One-way ANOVA with Tukey's range test for multiple comparison correction was used for comparison between multiple groups. NS: non-significant, $*P < 0.05$, $***P < 0.001$, $****P < 0.0001$.

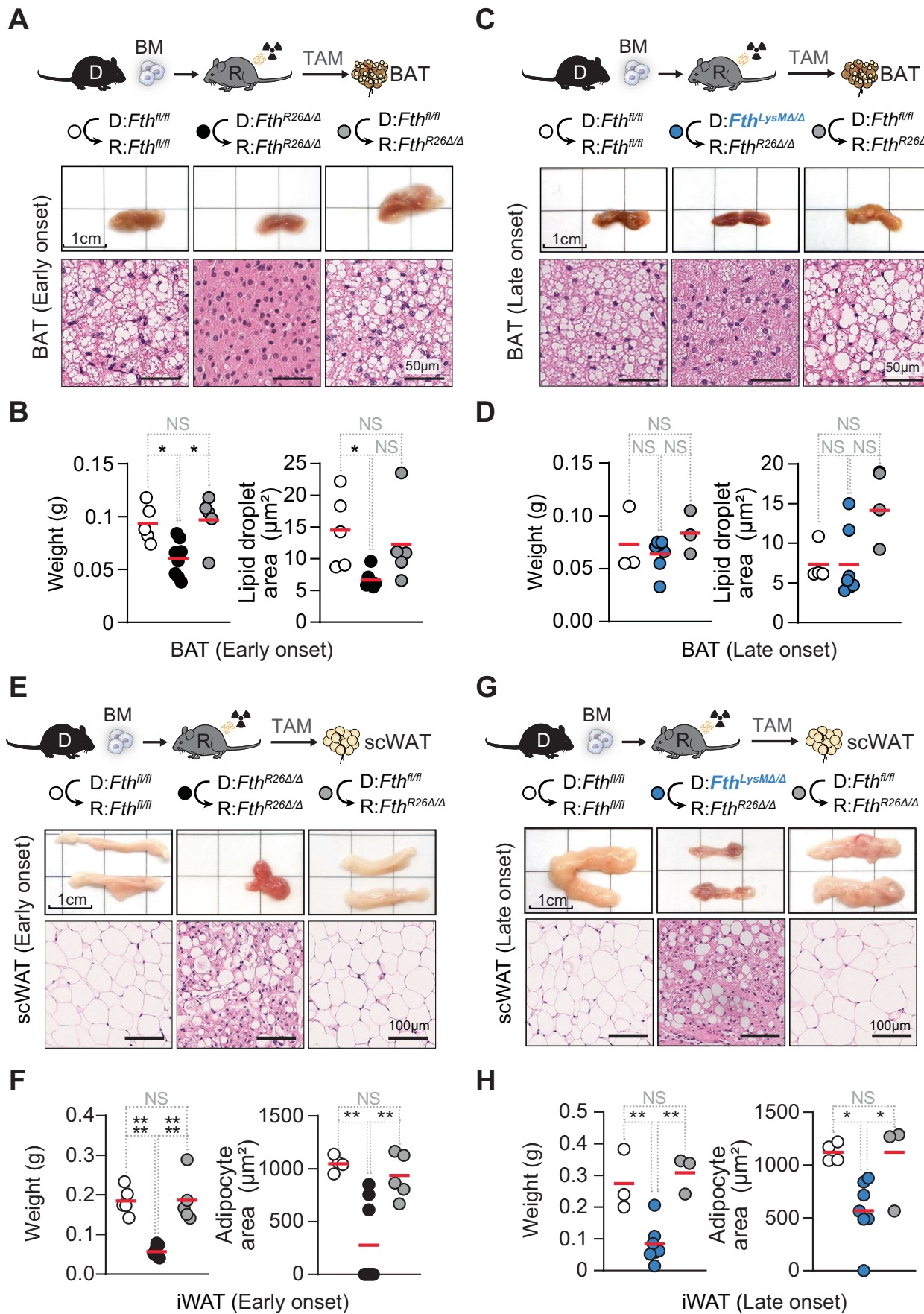

◀ **Figure EV4.** *Fth*-competent myeloid cells support BAT and WAT function in chimeric *Fth*-deleted mice.

(A) Schematic representation of chimeric mice and TAM-induced *Fth* deletion, and representative macroscopic and histological images of brown adipose tissue pads (BAT) of $Fth^{fl/fl} \Rightarrow Fth^{fl/fl}$ ($n = 5$), $Fth^{R26\Delta/\Delta} \Rightarrow Fth^{R26\Delta/\Delta}$ ($n = 8$) and $Fth^{fl/fl} \Rightarrow Fth^{R26\Delta/\Delta}$ ($n = 5$) chimeric mice, collected on day 7 (early onset) post TAM administration. (B) BAT pad weight (left) and mean (red bars) BAT adipocyte lipid droplet area (right) of $Fth^{fl/fl} \Rightarrow Fth^{fl/fl}$ ($n = 5$), $Fth^{R26\Delta/\Delta} \Rightarrow Fth^{R26\Delta/\Delta}$ ($n = 8$) and $Fth^{fl/fl} \Rightarrow Fth^{R26\Delta/\Delta}$ ($n = 5$) chimeric mice, collected on day 7 (early onset) post TAM administration. (C) Schematic representation of chimeric mice and TAM-induced *Fth* deletion, and representative macroscopic and histological images of BAT of $Fth^{fl/fl} \Rightarrow Fth^{fl/fl}$ ($n = 3–4$), $Fth^{LysM\Delta/\Delta} \Rightarrow Fth^{R26\Delta/\Delta}$ ($n = 6$) and $Fth^{fl/fl} \Rightarrow Fth^{R26\Delta/\Delta}$ ($n = 3–4$) chimeric mice, collected between days 16 and 39 (late onset) post TAM administration. (D) BAT pad weight (left) and mean (red bars) BAT adipocyte lipid droplet area (right) of $Fth^{fl/fl} \Rightarrow Fth^{fl/fl}$ ($n = 3–4$), $Fth^{LysM\Delta/\Delta} \Rightarrow Fth^{R26\Delta/\Delta}$ ($n = 6$) and $Fth^{fl/fl} \Rightarrow Fth^{R26\Delta/\Delta}$ ($n = 3–4$) chimeric mice, collected between days 16 and 39 (late onset) post TAM administration. Data in (B, D) represented as individual values (circles) and mean (red bars), pooled from three independent experiments with similar trends. (E) Schematic representation of chimeric mice and TAM-induced *Fth* deletion, and representative macroscopic and histological images of inguinal white adipose tissue pads (iWAT) of $Fth^{fl/fl} \Rightarrow Fth^{fl/fl}$ ($n = 4–5$), $Fth^{R26\Delta/\Delta} \Rightarrow Fth^{R26\Delta/\Delta}$ ($n = 8$) and $Fth^{fl/fl} \Rightarrow Fth^{R26\Delta/\Delta}$ ($n = 5$) chimeric mice, collected on day 7 (early onset) post TAM administration. (F) iWAT pad weight (left) and mean (red bars) iWAT adipocyte area (right) of $Fth^{fl/fl} \Rightarrow Fth^{fl/fl}$ ($n = 4–5$), $Fth^{R26\Delta/\Delta} \Rightarrow Fth^{R26\Delta/\Delta}$ ($n = 8$) and $Fth^{fl/fl} \Rightarrow Fth^{R26\Delta/\Delta}$ ($n = 5$) chimeric mice, collected on day 7 (early onset) post TAM administration. (G) Schematic representation of chimeric mice and TAM-induced *Fth* deletion, and representative macroscopic and histological images of iWAT of $Fth^{fl/fl} \Rightarrow Fth^{fl/fl}$ ($n = 3–4$), $Fth^{LysM\Delta/\Delta} \Rightarrow Fth^{R26\Delta/\Delta}$ ($n = 7$) and $Fth^{fl/fl} \Rightarrow Fth^{R26\Delta/\Delta}$ ($n = 3$) chimeric mice, collected between days 16 and 39 (late onset) post TAM administration. (H) iWAT pad weight (left) and mean (red bars) iWAT adipocyte area (right) of $Fth^{fl/fl} \Rightarrow Fth^{fl/fl}$ ($n = 3–4$), $Fth^{LysM\Delta/\Delta} \Rightarrow Fth^{R26\Delta/\Delta}$ ($n = 7$) and $Fth^{fl/fl} \Rightarrow Fth^{R26\Delta/\Delta}$ ($n = 3$) chimeric mice, collected between days 16 and 39 (late onset) post TAM administration. Data in (F, H) represented as individual values (circles) and mean (red bars), pooled from 3 independent experiments with similar trends. One-way ANOVA with Tukey's range test for multiple comparison correction was used for comparison between multiple groups. NS: non-significant, *$P < 0.05$, **$P < 0.01$, ****$P < 0.0001$.

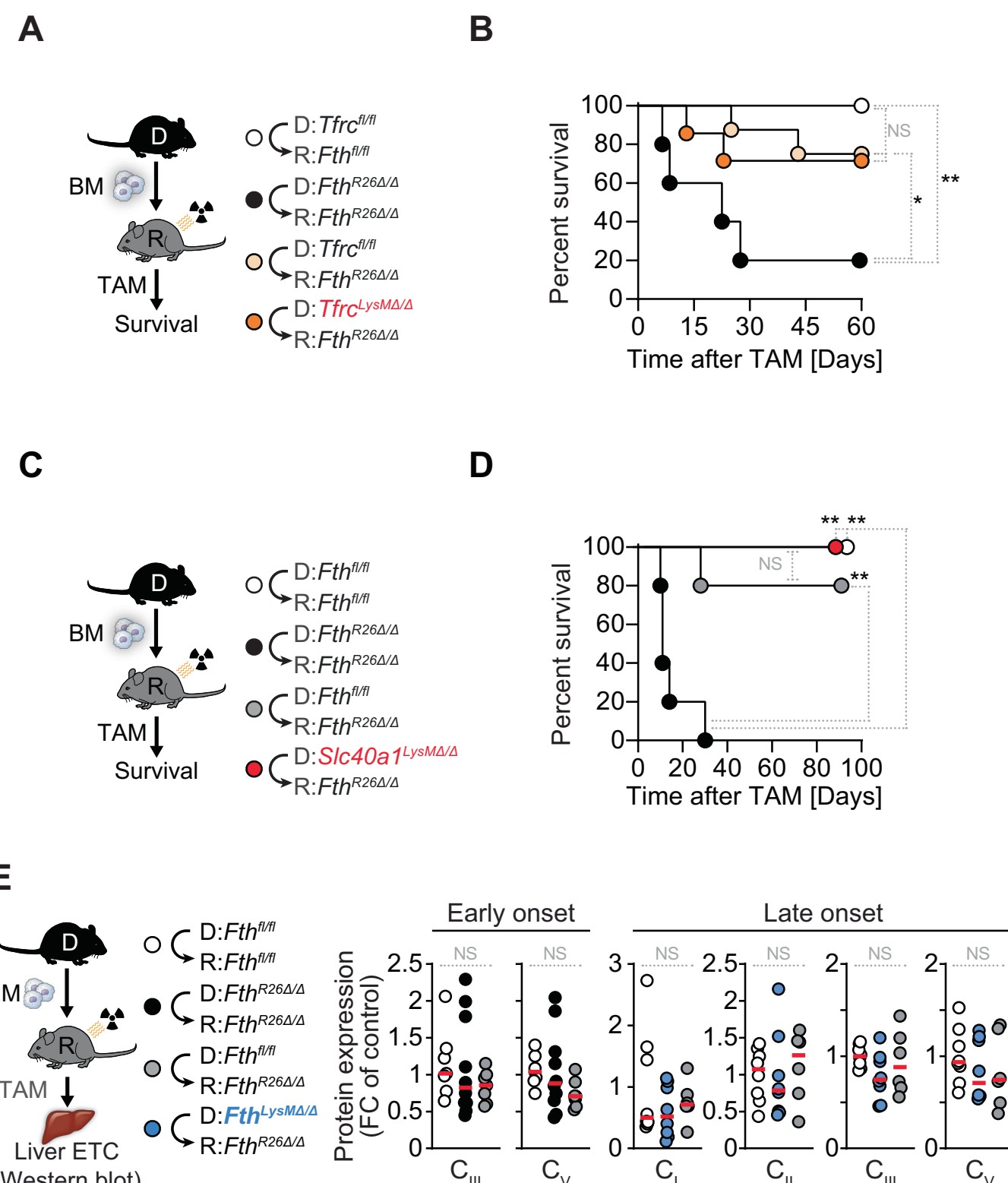

◀

**Figure EV5.   Myeloid cells rescue chimeric *Fth*-deleted mice irrespectively of cellular Fe import/export.**

(A) Schematic representation of TAM-induced *Fth* deletion in chimeric mice, and (B) survival of $Tfrc^{fl/fl}{\Rightarrow}Fth^{fl/fl}$ ($n = 7$), $Fth^{R26\Delta/\Delta}{\Rightarrow}Fth^{R26\Delta/\Delta}$ ($n = 5$), $Tfrc^{fl/fl}{\Rightarrow}Fth^{R26\Delta/\Delta}$ ($n = 8$) and $Tfrc^{LysM\Delta/\Delta}{\Rightarrow}Fth^{R26\Delta/\Delta}$ ($n = 7$) chimeric mice following TAM administration. Data in (B) is pooled from 2 independent experiments with similar trends. (C) Schematic representation of TAM-induced *Fth* deletion in chimeric mice, and (D) survival of $Fth^{fl/fl}{\Rightarrow}Fth^{fl/fl}$ ($n = 5$), $Fth^{R26\Delta/\Delta}{\Rightarrow}Fth^{R26\Delta/\Delta}$ ($n = 5$), $Fth^{fl/fl}{\Rightarrow}Fth^{R26\Delta/\Delta}$ ($n = 5$), and $Slc40a1^{LysM\Delta/\Delta}{\Rightarrow}Fth^{R26\Delta/\Delta}$ ($n = 5$) chimeric mice on day 0. (E) quantification of ETC subunits for complexes I-V in the livers of $Fth^{fl/fl}{\Rightarrow}Fth^{fl/fl}$ ($n = 7$), $Fth^{R26\Delta/\Delta}{\Rightarrow}Fth^{R26\Delta/\Delta}$ ($n = 10$) and $Fth^{fl/fl}{\Rightarrow}Fth^{R26\Delta/\Delta}$ ($n = 7$) chimeric mice, between days 7–8 (early onset), or from $Fth^{fl/fl}{\Rightarrow}Fth^{fl/fl}$ ($n = 8$), $Fth^{LysM\Delta/\Delta}{\Rightarrow}Fth^{R26\Delta/\Delta}$ ($n = 8$) and $Fth^{fl/fl}{\Rightarrow}Fth^{R26\Delta/\Delta}$ ($n = 6$) chimeric mice, between days 19 and 22 (late onset). Data in (E) is pooled from 4 independent experiments with similar trends. One-way ANOVA with Tukey's range test for multiple comparison correction was used for comparison between multiple groups. Survival analysis was performed using Log-rank (Mantel–Cox) test. NS: non-significant, *$P < 0.05$, **$P < 0.01$.

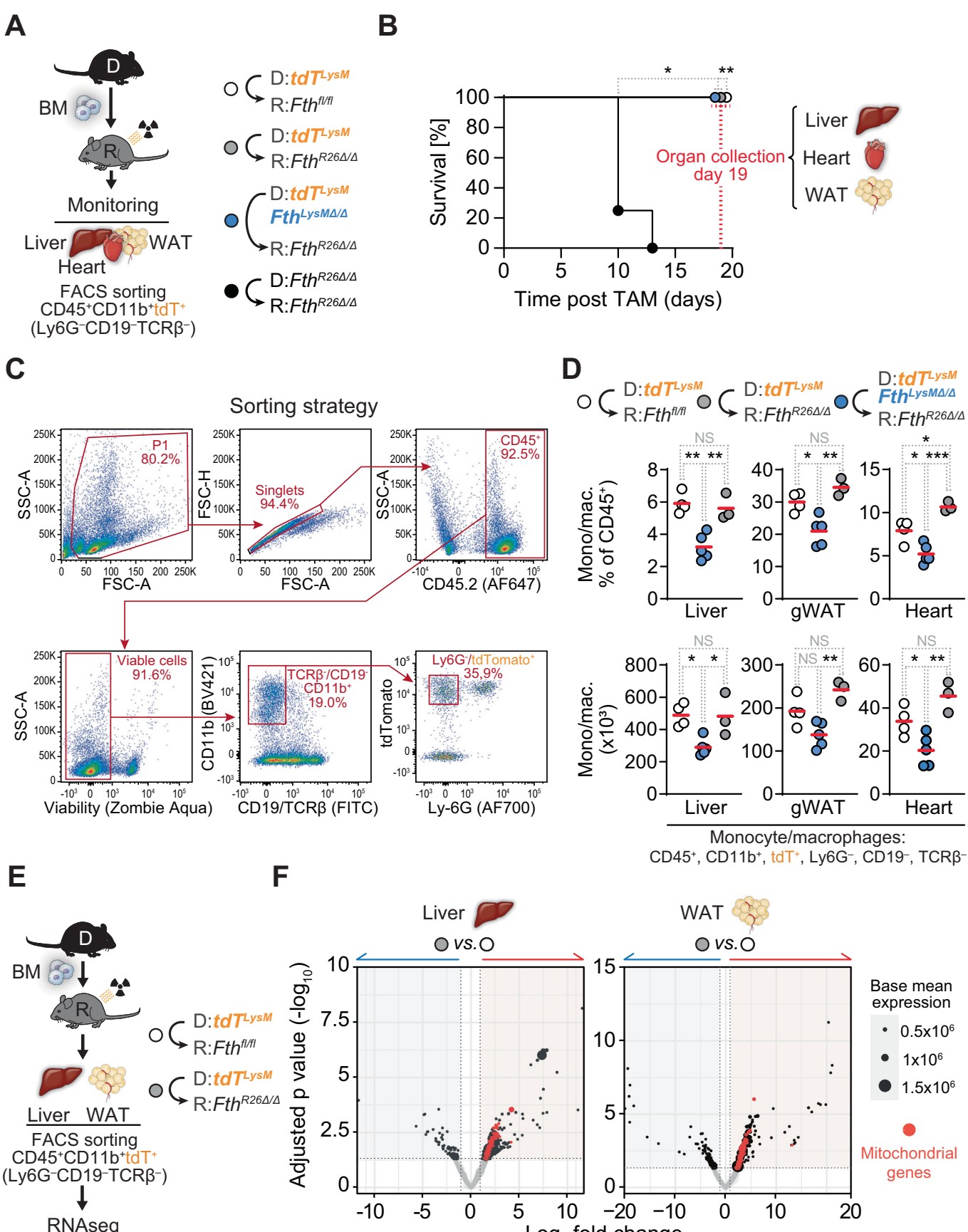

◀ **Figure EV6. RNA sequencing analysis of monocyte/macrophages in chimeric *Fth*-deleted mice.**

(A) Schematic representation of chimeric mice, TAM administration and fluorescence-activated cell sorting (FACS) of *LysM*$^+$ monocyte/macrophages (CD45$^+$,CD11b$^+$,Ly6G$^-$,CD19$^-$,TCRβ$^-$) in liver, heart and WAT. (B) Survival of *tdT$^{LysM}$*⇒*Fth$^{fl/fl}$* ($n = 4$), *tdT$^{LysM}$*⇒*Fth$^{R26Δ/Δ}$* ($n = 3$), *tdT$^{LysM}$Fth$^{LysMΔ/Δ}$*⇒*Fth$^{R26Δ/Δ}$* ($n = 5$) and *Fth$^{R26Δ/Δ}$*⇒*Fth$^{R26Δ/Δ}$* ($n = 4$) chimeric mice until organ collection on day 19 post-TAM administration. (C) Gating strategy for sorting *LysM*$^+$ monocyte/macrophages (CD45$^+$, CD11b$^+$, Ly6G$^-$, CD19$^-$, TCRβ$^-$) in liver, WAT and Heart. (D) Proportion and number and of viable *LysM*$^+$ monocyte/macrophages (CD45$^+$, CD11b$^+$, Ly6G$^-$, CD19$^-$, TCRβ$^-$) in the liver, gWAT and heart of *tdT$^{LysM}$*⇒*Fth$^{fl/fl}$* ($n = 4$), *tdT$^{LysM}$*⇒*Fth$^{R26Δ/Δ}$* ($n = 3$) and *tdT$^{LysM}$Fth$^{LysMΔ/Δ}$*⇒*Fth$^{R26Δ/Δ}$* ($n = 5$) chimeric mice on day 19 post-TAM administration. (E) Schematic representation of chimeric mice, TAM administration and fluorescence-activated cell sorting (FACS) of *LysM*$^+$ monocyte/macrophages (CD45$^+$, CD11b$^+$, Ly6G$^-$, CD19$^-$, TCRβ$^-$) in liver and WAT. (F) Volcano plots of differentially regulated genes between *LysM*$^+$ monocyte/macrophages sorted from liver (left) or WAT (right) of *tdT$^{LysM}$*⇒*Fth$^{R26Δ/Δ}$* ($n = 3$), *tdT$^{LysM}$*⇒*Fth$^{fl/fl}$* ($n = 4$) chimeric mice, on day 19 post TAM administration. Red dots depict mitochondrial genes that are significantly differentially regulated. One-way ANOVA with Tukey's range test for multiple comparison correction was used for comparison between multiple groups. Survival analysis was performed using Log-rank (Mantel–Cox) test. NS: non-significant, *$P < 0.05$, **$P < 0.01$, ***$P < 0.001$.

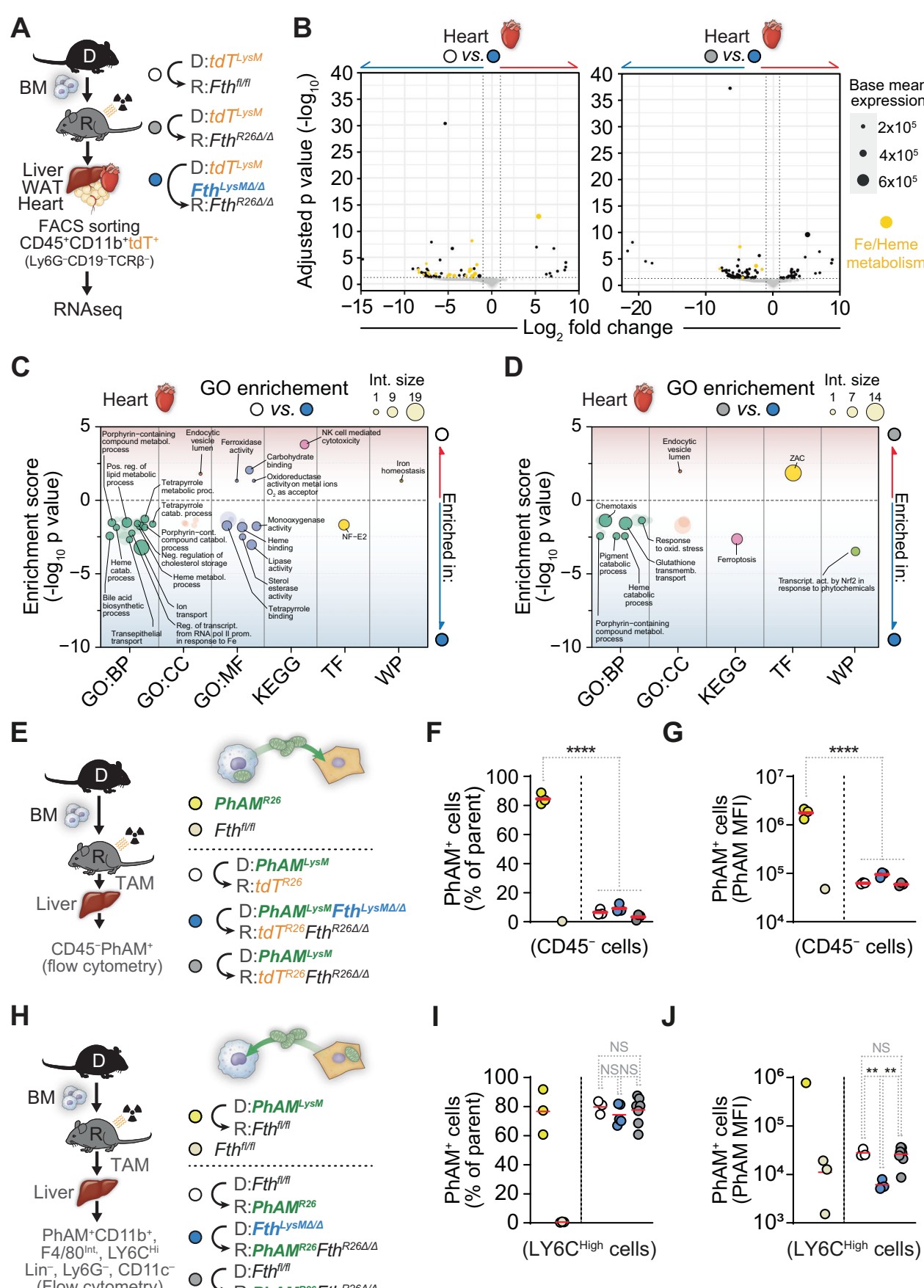

◀

**Figure EV7.  *Fth*-competent monocyte-derived macrophages in the heart of *Fth*-deleted chimeras do not employ a mitochondrial gene transcriptional program.**

(A) Schematic representation of chimeric mice, TAM administration and fluorescence-activated cell sorting (FACS) of *LysM*+ monocyte/macrophages (CD45+, CD11b+, Ly6G−, CD19−, TCRβ−) in liver, WAT and heart. (B) Volcano plots of differentially regulated genes between *LysM*+ monocyte/macrophages sorted from the heart of *tdT^LysM^Fth^LysMΔ/Δ^⇒Fth^R26Δ/Δ^* (n = 5) vs. *tdT^LysM^⇒Fth^fl/fl^* (n = 4; left) or *tdT^LysM^⇒Fth^R26Δ/Δ^* (n = 3; right) chimeric mice, on day 19 post TAM administration. Yellow dots depict genes involved in iron/heme metabolism that are significantly differentially regulated. Gene ontology analysis depicting ontologies that are significantly enriched comparing *LysM*+ monocyte/macrophages sorted from heart of (C) *tdT^LysM^Fth^LysMΔ/Δ^⇒Fth^R26Δ/Δ^* (n = 5) chimeras, vs. *tdT^LysM^⇒Fth^fl/fl^* (n = 4; left) or (D) *tdT^LysM^⇒Fth^R26Δ/Δ^* (n = 3; right) chimeric mice on day 19 post-TAM administration. Ontologies significantly enriched in *LysM*+ monocyte/macrophages from *tdT^LysM^⇒Fth^fl/fl^* are depicted as: enrichment score −log₁₀ P value > 1.301. Ontologies significantly enriched in *LysM*+ monocyte/macrophages from *tdT^LysM^Fth^LysMΔ/Δ^⇒Fth^R26Δ/Δ^* or *tdT^LysM^⇒Fth^R26Δ/Δ^* chimeras are depicted as: enrichment score -log₁₀ P value < −1.301. Ontology classes: TF = transcription factors; GO:CC = gene ontology: cellular component; GO:MF = GO: molecular function; GO:BP = GO: biological process; KEGG = Kyoto Encyclopedia of Genes and Genomes pathway; WP = Wiki Pathways. (E) Schematic representation of chimeric mice, TAM administration (day 0), and flow cytometry analysis of livers from positive (*PhAM^R26^*; n = 3) and negative control (*Fth^fl/fl^*; n = 1) mice, and *PhAM^LysM^⇒tdT^R26^* (n = 3), *PhAM^LysM^Fth^LysMΔ/Δ^⇒tdT^R26^Fth^R26Δ/Δ^* (n = 3), *PhAM^LysM^⇒tdT^R26^Fth^R26Δ/Δ^* (n = 3) chimeras. (F) Percentage and (G) median fluorescence intensity (MFI) of CD45− cells that are PhAM+. (H) Schematic representation of chimeric mice, TAM administration (day 0), and flow cytometry analysis of livers from positive (*PhAM^R26^⇒Fth^fl/fl^*; n = 3) and negative control (*Fth^fl/fl^*; n = 3) mice, and *Fth^fl/fl^⇒PhAM^R26^* (n = 3), *Fth^LysMΔ/Δ^⇒PhAM^R26^Fth^R26Δ/Δ^* (n = 3), *Fth^fl/fl^⇒PhAM^R26^Fth^R26Δ/Δ^* (n = 3) chimeras. (I) Percentage and (J) median fluorescence intensity (MFI) of LY6C^High^ monocyte-derived macrophages (CD11b+, F4/80^Int^, LY6C^Hi^, Lin−, LY6G−, CD11C−, PhAM+) that are PhAM+. Data in (F) is pooled from 2 independent experiments. Data in (F, G, I, J) is presented as individual values and mean. One-way ANOVA with Tukey's range test for multiple comparison correction was used for comparison between multiple groups. NS: non-significant, **P < 0.01, ****P < 0.0001.

