## [Peer Review File · The EMBO Journal]

Homeostatic control of energy metabolism by monocyte-derived macrophages

Rui Martins, Birte Blankenhaus, Faouzi Braza, Miguel Mesquita, Pedro Ventura, Sumnima Singh, Sebastian Weis, Maria Pires, Sara Pagnotta, Qian Wu, Sílvia Cardoso, Elisa Jenthó, Ana Figueiredo, Pedro Faísca, Ana Nóvoa, Vanessa Morais, Stefanie Wculek, David Sancho, Moises Mallo, and Miguel Soares

Corresponding author(s): Miguel Soares (miguel.soares@gimm.pt)

Review Timeline:

Submission Date:	27th Feb 25
Editorial Decision:	14th May 25
Revision Received:	28th Jul 25
Editorial Decision:	11th Aug 25
Revision Received:	13th Sep 25
Accepted:	28th Sep 25

Editor: Ioannis Papaioannou

Transaction Report:

Dear Miguel,

Thank you again for submitting your manuscript EMBOJ-2025-120530 for consideration by The EMBO Journal, and for your patience during peer review. As I have already informed you, your manuscript has been seen by two experts in the field, and we have received their detailed, constructive, and well-informed reports, which I have already shared with you (they are included again below).

I am pleased to say that both referees indicate interest in the study and the findings, recognize the novelty and importance of the work, find the amount and quality of the performed experimental work commendable, and conclude that it will make a significant contribution to the field. They also list a number of reasonable suggestions for strengthening the work further.

Given the referees' supportive comments and positive recommendations, I would like to invite you to submit a revised version of your manuscript taking the referees' suggestions on board, along with a detailed point-by-point response addressing all referees' comments. I should add that it is The EMBO Journal policy to allow only a single round of major revision, and acceptance of your manuscript will therefore depend on the completeness of your responses in this revised version. Please let me know if you have any questions or comments that you would like to discuss with me. If there are any major points you do not agree with or cannot address during your revision, I would encourage you to share them with me as early as possible to discuss how to proceed further in the most efficient way.

We generally allow three months as standard revision time (August 13, 2025). As a matter of policy, competing manuscripts published during this period will not negatively impact our assessment of the conceptual advance presented by your study. However, we request that you contact us as soon as possible upon publication of any related work, to discuss how to proceed. Should you foresee a problem in meeting this three-month deadline, please let us know in advance and we will be able to grant an extension.

Thank you for the opportunity to consider your work for publication in The EMBO Journal. I look forward to your revision.

Best wishes,

Ioannis

Instructions for preparing your revised manuscript

1. When you are ready to submit the revision, please upload:

- A Word file of the manuscript text (including legends of main Figures, EV Figures and Tables). Please make sure that changes are highlighted (or "tracked") to be clearly visible.

- Individual production-quality figure files (one file per figure). When assembling your figures, please refer to our figure preparation guidelines in order to ensure proper formatting and readability in print as well as on screen:

If the data shown in a figure are obtained from n {less than or equal to} 2, please use scatter plots showing the individual data points.

- i. the name of the statistical test used to generate error bars and P values
- ii. the number (n) of independent experiments (please specify technical or biological replicates) underlying each data point (discussion of statistical methodology can be reported in the Materials and Methods section, but figure legends should contain a basic description of n , P , and the test applied)
- iii. the nature of the bars and error bars (s.d., s.e.m.).

- A point-by-point response to the referees' comments, with a detailed description of the changes made (as a word file). All

referees' concerns must be fully addressed and their suggestions taken on board. When preparing your letter of response to the referees' comments, please bear in mind that this will form part of the Review Process File and will therefore be available online to the community. Please note that you have the possibility to opt out of the transparent process at any stage prior to publication by letting the editorial office know (contact@embojournal.org); if you do opt out, the Review Process File link will point to the following statement: "No Review Process File is available with this article, as the authors have chosen not to make the review process public in this case.". For more details on our Transparent Editorial Process, please visit our website: <https://www.embopress.org/page/journal/14602075/authorguide#transparentprocess>

- Expanded View (EV) files (replacing Supplementary Information) that are collapsible/expandable online. A maximum of 5 EV Figures can be typeset. EV Figures should be cited as "Figure EV1, Figure EV2" etc. in the text, and their respective legends should be included in the manuscript file after the legends of regular figures. See detailed instructions regarding Expanded View files here:

- For the figures that you do NOT wish to display as Expanded View figures, they should be bundled together with their legends in a single PDF file called "Appendix", which should start with a short Table of Contents (including page numbers). Appendix figures should be referred to in the main text as: "Appendix Figure S1, Appendix Figure S2" etc. Please see detailed instructions here: <https://www.embopress.org/page/journal/14602075/authorguide#expandedview>

- A complete author checklist, which you can download from our author guidelines (<https://www.embopress.org/page/journal/14602075/authorguide>). Please note that the checklist will also be part of the Review Process File.

2. Please note that no statistics should be calculated and shown in Figures if $n=2$. Please also note that each p value should be reported as an exact value.

3. Before submitting your revision, primary datasets (and computer code, where appropriate) produced in this study need to be deposited in appropriate public databases (see <https://www.embopress.org/page/journal/14602075/authorguide#dataavailability>). In particular, you are kindly requested to deposit all RNA sequencing data produced in this study in an appropriate database. The accession numbers, database, and the specific URLs (links) should be listed in a formal "Data availability" section (placed after Methods), following the example below:

"The RNA-seq datasets produced in this study are available in the following database:
Gene Expression Omnibus GSE46843 (<https://www.ncbi.nlm.nih.gov/geo/query/acc.cgi?acc=GSE46843>)"

*** All links should resolve to a page where the data can be accessed. ***

*** Please remember to provide in the Data availability section of your revised manuscript reviewer passwords if the datasets are not yet public. ***

*** The Data Availability Section is restricted to new primary data that are part of this study. In case you have no data that require deposition in a public database, please state so instead of referring to the database: "Our study includes no data deposited in public repositories." under the heading "Data availability". ***

4. The materials and methods need to be described in the manuscript using our structured methods format, which is now required for all research articles. According to this format, the Methods section includes a single "Reagents and Tools Table" - listing key reagents, experimental models, software and relevant equipment including their sources and relevant identifiers- followed by a "Methods and Protocols" section describing the methods. Please download and fill our Reagents and Tools Table template (.docx), which you can find in our author guide:

<https://www.embopress.org/page/journal/14602075/authorguide#structuredmethods>. When submitting your revised manuscript, please do not include the Reagents and Tools Table in the Methods section of the manuscript but instead upload it as a separate file choosing the file type "Reagent Table".

5. Please check that the title and the abstract of the manuscript are brief, yet explicit, even to non-specialists. The length of the title should not exceed 100 characters, and the abstract should be a single paragraph not exceeding 175 words.

6. Please also note our reference format: <https://www.embopress.org/page/journal/14602075/authorguide#referencesformat>.

8. Please remember: digital image enhancement is acceptable practice, as long as it accurately represents the original data and conforms to community standards. If a figure has been subjected to significant electronic manipulation, this must be noted in the figure legend or in the "Materials and Methods" section. The editors reserve the right to request original versions of figures and

the original images that were used to assemble the figure.

9. Our journal encourages inclusion of data citations in the reference list to directly cite datasets that were obtained from public databases. Data citations in the article text are distinct from normal bibliographical citations and should directly link to the database records from which the data can be accessed. In the main text, data citations are formatted as follows: "Data ref: Smith et al, 2001" or "Data ref: NCBI Sequence Read Archive PRJNA342805, 2017". In the Reference list, data citations must be labeled with "[DATASET]". A data reference must provide the database name, accession number/identifiers, and a resolvable link to the landing page from which the data can be accessed at the end of the reference. Further instructions are available at: <https://www.embopress.org/page/journal/14602075/authorguide#referencesformat>.

10. We request authors to consider both actual and perceived competing interests. Please review our policy (<https://www.embopress.org/page/journal/14602075/authorguide#conflictsofinterest>) and update your competing interests statement if necessary. Please name this section 'Disclosure and competing interests statement' and place it after the Acknowledgements section.

11. Please note that all corresponding authors are required to provide an ORCID ID upon submission of a revised manuscript (<https://orcid.org/>). Please find instructions on how to link your ORCID ID to your account in our manuscript tracking system in our Author guidelines (<https://www.embopress.org/page/journal/14602075/authorguide#authorshipguidelines>).

12. We use CRediT to specify the contributions of each author in the journal submission system. CRediT replaces the author contribution section, which should be removed from the manuscript. Please use the free text box to provide more detailed descriptions. See also guide to authors: <https://www.embopress.org/page/journal/14602075/authorguide#authorshipguidelines>.

14. We would also welcome the submission of cover suggestions or motifs to be used by our Graphics Illustrator in designing a cover.

15. Please use the link below to submit your revision:
<https://emboj.msubmit.net/cgi-bin/main.plex>

Referee #1:

Martins et al. investigated the mechanisms of the lethal effects of Fth depletion in parenchymal cells. They find, remarkably and as shown in extraordinary detail, that Fth-competent monocyte-derived macrophages are necessary to protect the entire organism from the effect of genetic Fth depletion. The authors use a painstaking and complete approach to their in vivo experiments, carefully proceeding step-by-step to ultimately arrive at the conclusion that BM-derived monocytes rescue the energy balance and viability of the entire organism. The authors further show that Mo-derived macrophages must maintain their mitochondria function (as shown by the Tfam and CIII KO experiments) to exert their system-wide effects. Overall, the findings presented by Martins et al. are a landmark in understanding global Fe metabolism and macrophage-"other cell" cross-talk. The authors also set the stage for new mechanistic approaches to understanding this crosstalk - although beyond the scope of this massive compendium of experiments, simple ex vivo/in vitro approaches will be needed to dissect exactly how Fth-competent myeloid cells do their job. This is especially important given that mitochondrial transfer was excluded as a predominant mechanism.

There are some minor issues:

1. In my copy downloaded from the journal web site, Figs. 5C (Gapdh), 5F (top left panel), 6D (one of the lower panels) and 6I are all missing the information intended.
2. The authors mention the kinetic differences in the effects of the LysM-cre compared to the other mice (and the Cx3cr1 experiments). However, as LysM-cre is an incomplete, mosaic delete, such difference e.g., in 1D could be the result of (i) incomplete deletion, (ii) in vivo selection for cells that express some Fth, or (iii) both. The authors should comment on these possibilities and the (many) limitations of the LysM-cre mice.
3. Following from point 2, the similar issues exist with the Ccr2 KO. In fact, the authors data (S2C) shows what one would expect from mice that retain ~10-15% of circulating monocytes. Thus, one can speculate that even a small number of Fth-competent monocytes can rescue some mice, to some level.

4. One element to discuss is the viability of cells lacking ferritin - there could be a very complex scenario where relative viability of different cells plays a major role in the outcomes. For example, in 1B, C, what the viability/half life of Fth-deficient monocytes? The relative viability of the monocytes is probably best tested in a parabiosis experiment where two pools of monocytes can be measured relative to one another. Another way to ask this question concerns whether Fth-deficient monocytes die of ferroptosis and whether an Fth KO monocyte can do its job if ferroptosis is suppressed (e.g., by overexpression of Fsp1, etc.). This is a discussion point.

Referee #2:

Martins et al sought to understand how hematopoietic-derived cells contribute to organismal homeostasis, a fundamental biological process. The manuscript builds up on a previous publication by the team showing that FTH is a regulator of organismal Fe homeostasis. Previously, they showed that ferritin is essential to support energy expenditure, thermogenesis, mitochondrial function and integrity of parenchymal tissues. In the present manuscript, they showed that monocyte-derived macrophages regulate thermogenesis and energy balance. By employing a variety of mouse strains and bone marrow chimeras, they concluded that monocyte-derived macrophages perform this function intrinsically and extrinsically via FTH. The manuscript has an impressive amount of well performed work, employing a variety of mouse strains, clever ways of crossing them to achieve their goals, and an astounding number of bone marrow chimeras allowing them to discern the role of radioresistant (parenchyma) vs radiosensitive cells (hematopoietic compartment) on homeostasis. They shed light on the role of macrophages in supporting the function of parenchyma cells, that in turn, contributes to the maintenance of tissue homeostasis. The authors were very careful with their conclusions since Fth deletion in LysM+ cells also leads to significantly fewer monocyte-derived macrophages. Thus, the main concern is that it was difficult to disentangle the effects mediated by FTH produced by LysM+ cells from other roles these macrophages could play when organismal homeostasis is disrupted by the absence of Fth in parenchyma cells. Please see general and specific comments below.

General comments

- o How the authors explain the early vs late onset using global deletion of Fth vs specific deletion in LysM+ cells, respectively?
- o It is puzzling that the severity of disease is higher in Fth-deficient mice reconstituted with Fth-deficient BM compared to Fth-deficient LysM+ cells. Although eventually, all mice succumb to disease, what are the author's thoughts on the delayed onset in a subset of mice analyzed? Is it possible to compare the effects of BMC donors with global deletion of Fth vs LysM specific deletion at an earlier time point?
- o Is it possible that the fewer mitochondria in Fth-depleted macrophages can cause their death or affect BM output?

Specific comments

1. In Fig 1F-H, how can we tell apart the role of Fth produced by macrophages if the population itself is depleted when Fth is ablated from LysM+ cells? Can the authors explain the 50% effect when they ablated Fth specifically on CXCR1+ cells?
2. In Fig 1J-K, where do the transferred BMDM go? Do they undergo tissue differentiation?
3. Can the authors explain the discrepancy between the survival when mice were reconstituted with Fth-sufficient BM vs Fth-sufficient BMDM (the grey symbol in both panels D and K of Fig 1)? Presumably, in both cases survival extension is maintained by monocyte-derived macrophages. In D, only 25% of mice succumb to disease but in K all mice succumb to disease. Is the recovery of monocyte-derived macrophages the same in Fth-deficient and Fth-sufficient BMDM?
4. In Fig S6, can the authors clarify the time points post tamoxifen treatment chosen for the analysis? If the animals were about to die, isn't it expected they would move and eat less? Based on this data, can the authors conclude about energy metabolism in this scenario?
5. In Fig 4, similar to comments above, how to disentangle the effect of FTH from other roles macrophages could be playing on homeostasis maintenance since they are also largely absent when using a conditional deletion of FTH in LysM+ cells? Can the authors perform these analyses at an earlier time point, long before animals were supposed to die?
6. In Fig 5A-B, was the WB performed on whole tissue lysates? Since specific deletion of FTH in LysM+ cells also results in decreased numbers of monocyte-derived macrophages, how can the authors exclude the possibility that the FTH signal they see is not coming from the monocyte-derived macrophages that are also found in the heart, liver, kidneys and lungs?
7. In Fig 7, is possible that the deletion of Fth is under reported using the TdTomato strategy, since the efficiency of Rosa26 is likely higher than Fth flox (Rosa26 locus is more permissive). However, they do see an effect in the macrophages from Fth flox mice compared to WT suggesting there is an enrichment of Fth-deficient cells among the tomato+ cells. Is it possible to sort the tomato+ cells and check for Fth expression?
8. Is S9D reporting total number of macrophages in the indicated organs or number of tomato+ macrophages? Can the authors show the proportion of tomato+ cells in Fth-deficient vs sufficient mice?

Minor comments

- o Fig 6I came out blank (no bands are visible, at least in the PDF I received).
- o Fig S9C has a typo on the axis: dTomato instead of tdTomato. Also S9D, LysM mice are referred as LyzM mice.
- o Fig S4A and Fig 2J. Fthwt/wt is referred in the text as Fthf/f.

Reviewer 1:

Martins et al. investigated the mechanisms of the lethal effects of Fth depletion in parenchymal cells. They find, remarkably and as shown in extraordinary detail, that Fth-competent monocyte-derived macrophages are necessary to protect the entire organism from the effect of genetic Fth depletion. The authors use a painstaking and complete approach to their in vivo experiments, carefully proceeding step-by-step to ultimately arrive at the conclusion that BM-derived monocytes rescue the energy balance and viability of the entire organism. The authors further show that Mo-derived macrophages must maintain their mitochondria function (as shown by the Tfam and CIII KO experiments) to exert their system-wide effects. Overall, the findings presented by Martins et al. are a landmark in understanding global Fe metabolism and macrophage-"other cell" cross-talk. The authors also set the stage for new mechanistic approaches to understanding this crosstalk - although beyond the scope of this massive compendium of experiments, simple ex vivo/in vitro approaches will be needed to dissect exactly how Fth-competent myeloid cells do their job. This is especially important given that mitochondrial transfer was excluded as a predominant mechanism.

There are some minor issues:

#1. *In my copy downloaded from the journal web site, Figs. 5C (Gapdh), 5F (top left panel), 6D (one of the lower panels) and 6I are all missing the information intended.*

Reply to #1: We believe that this may occur during the PDF conversion at the EMBO J. platform. We confirmed that Figures 5C (GAPDH), 5F (top left panel), 6D (one of the lower panels) and 6I contain the information intended.

#2. *The authors mention the kinetic differences in the effects of the LysM-cre compared to the other mice (and the Cx3cr1 experiments). However, as LysM-cre is an incomplete, mosaic delete, such difference e.g., in 1D could be the result of (i) incomplete deletion, (ii) in vivo selection for cells that express some Fth, or (iii) both. The authors should comment on these possibilities and the (many) limitations of the LysM-cre mice.*

Reply to #2: We agree with this reviewer's comment and as suggested, added a paragraph on page 4 of the revised manuscript providing further context to the limitations of the $LysM^{Cre}$ mice approach used in our study.

“Although $Fth^{LysM\Delta/\Delta} \Rightarrow Fth^{R26\Delta/\Delta}$ chimeras succumbed to Fth deletion, the temporal dynamics were delayed, as compared to $Fth^{R26\Delta/\Delta} \square Fth^{R26\Delta/\Delta}$ chimeras. This likely reflects (i) mosaic/incomplete Fth deletion in $LysM^{Cre}$ cells, (ii) in vivo selection for cells that express some FTH or (iii) both. As such, subsequent analyses were conducted as the body weight of $Fth^{R26\Delta/\Delta} \Rightarrow Fth^{R26\Delta/\Delta}$ or $Fth^{LysM\Delta/\Delta} \Rightarrow Fth^{R26\Delta/\Delta}$ was reduced by more than 10% defining “early onset” and “late onset”, respectively.”

Moreover, we verified the extent of $LysM^{Cre}$ recombination in monocyte/macrophage populations under the experimental conditions used in our study. Namely, we reconstituted $Fth^{R26fl/fl}$ or control $Fth^{fl/fl}$ mice with BM cells expressing the tdTomato (*tdT*) fluorescent reporter under the control of the $LysM^{Cre}$ promoter ($LysM^{Cre}tdTomato^{fl/fl}$ mice, here termed tdT^{LysM+}). We confirmed that Cre recombination occurs in approximately ~70-75% Ly6C^{Hi} monocytes/macrophages, as assessed in the liver of $tdT^{LysM+} \Rightarrow Fth^{fl/fl}$ and $tdT^{LysM+} \Rightarrow Fth^{R26\Delta/\Delta}$ chimeras, expressing FTH or not

in parenchyma cells, respectively (see Fig. 1 below). This observation is in line with the comment from the reviewer and provides support for the added paragraph.

Figure 1: (A) Schematic representation of chimeric mice, Tamoxifen (TAM) administration and flow cytometry analysis of LY6C^{Hi} monocyte/macrophages (CD45⁺, CD11b⁺, Ly6G⁻, CD19⁻, TCRβ⁻, LY6C^{Hi}) in the liver of lethally-irradiated control ($Fth^{fl/fl}$) or FTH-deficient ($Fth^{R26\Delta/\Delta}$) mice, reconstituted with LysM reporter BM (tdT^{LysM+}). (B) Representative histogram and (C) percentage of LY6C^{Hi} monocyte/macrophages expressing tdTomato. Circles represent individual mice (n=2 per group) from one experiment.

#3. Following from point 2, the similar issues exist with the *Ccr2* KO. In fact, the authors data (S2C) shows what one would expect from mice that retain ~10-15% of circulating monocytes. Thus, one can speculate that even a small number of *Fth*-competent monocytes can rescue some mice, to some level.

Reply to #3: We agree with the reviewer’s comment. Most likely, there is a relatively low number of circulating monocytes (~10-15% of control), originating from CCR2 knock-out BM (Tsou, Peters et al., 2007), that suffice to reduce the deleterious effects of *Fth* deletion in parenchyma cells. This is supported by the observation that adoptive transfer of a relatively low number of BM-derived monocytes (~100 cells per mg/body weight) is sufficient to provide a survival advantage to *Fth*-deleted mice (see Figure 1D). We added a paragraph on page 6 of the revised manuscript to reflect this point.

*“This is likely explained by *Ccr2* deletion not fully preventing monocyte BM egression (Tsou et al., 2007), with low numbers of monocyte-derived macrophages being sufficient to support organismal homeostasis, when *Fth* is deleted in parenchyma cells. As a non-mutually exclusive hypothesis, monocyte-derived macrophages might support organismal homeostasis via a mechanism that is only partially CCL2/CCR2-dependent.”*

#4. One element to discuss is the viability of cells lacking ferritin - there could be a very complex scenario where relative viability of different cells plays a major role in the outcomes. For example, in 1B, C, what the viability/half-life of *Fth*-deficient monocytes? The relative viability of the monocytes is probably best tested in a parabiosis experiment where two pools of monocytes can be measured relative to one another. Another way to ask this question concerns whether

Fth-deficient monocytes die of ferroptosis and whether an *Fth* KO monocyte can do its job if ferroptosis is suppressed (e.g., by overexpression of *Fsp1*, etc.). This is a discussion point.

Reply to #4 We agree with the reviewer's hypothesis. We included in the revised version of the manuscript experimental evidence showing that FTH is essential to support the viability of monocyte-derived macrophages, as these engage into the different phenotypes described. Briefly, we found that FTH is essential to support macrophage viability as these cells take up mitochondria from parenchyma cells (*revised Manuscript Fig. 8C-E, I-K illustrated below as Fig. 2A-C*). Moreover, we confirmed that FTH-deficient BMDM are more susceptible to iron-mediated cytotoxicity, as compared to FTH-competent cells (*revised Manuscript Fig. 8I,L, illustrated below as Fig. 2A,D*). These data are now included in the *revised Manuscript Fig. 8I-L* and are described in the results section (see page 18 of the revised manuscript).

Figure 2: (A) Schematic representation of control (*Fth^{fl/fl}*) or FTH-deficient (*Fth^{R26Δ/Δ}*) BMDM generation, Tamoxifen (TAM) administration and cell viability assessment. (B) Representative brightfield and fluorescent microscopy images of viability staining (PI) of BMDM treated with vehicle, or mitochondria, 15h hours post-treatment. (C) Time course of BMDM cell death upon vehicle or mitochondria administration. For (B, C), wells receiving mitochondria were administered mitochondria purified from 60mg of liver. (D) Time course of BMDM cell death upon vehicle or Fe (FAC, ferric ammonium citrate; 125μM) administration. Data in (B, C) is presented as mean +/- SEM (n=3 replicates). Two-Way ANOVA was used for comparison between two groups within time courses. NS: non-significant, ** P<0.01, **** P<0.0001.

Reviewer 2:

Martins et al sought to understand how hematopoietic-derived cells contribute to organismal homeostasis, a fundamental biological process. The manuscript builds up on a previous publication by the team showing that FTH is a regulator of organismal Fe homeostasis. Previously, they showed that ferritin is essential to support energy expenditure, thermogenesis, mitochondrial function and integrity of parenchymal tissues. In the present manuscript, they showed that monocyte-derived macrophages regulate thermogenesis and energy balance. By employing a variety of mouse strains and BM chimeras, they concluded that monocyte-derived macrophages perform this function intrinsically and extrinsically via FTH. The manuscript has an impressive amount of well performed work, employing a variety of mouse strains, clever ways of crossing them to achieve their goals, and an astounding number of BM chimeras allowing them to discern the role of radioresistant (parenchyma) vs radiosensitive cells (hematopoietic compartment) on homeostasis. They shed light on the role of macrophages in supporting the function of parenchyma cells, that in turn, contributes to the maintenance of tissue homeostasis. The authors were very careful with their conclusions since *Fth* deletion in *LysM*⁺ cells also leads to significantly fewer monocyte-derived macrophages. Thus, the main concern is that it was difficult to disentangle the effects mediated by FTH produced by *LysM*⁺ cells from other roles these macrophages could play when organismal homeostasis is disrupted by the absence of *Fth* in parenchyma cells. Please see general and specific comments below.

General comments

#1. How the authors explain the early vs late onset using global deletion of *Fth* vs specific deletion in *LysM*⁺ cells, respectively?

Reply to #1: *LysM*^{Cre} provides an incomplete (mosaic) Cre expression and therefore Cre-mediated deletion through the myeloid lineage. We verified the penetrance of *LysM*^{Cre} recombination in monocyte/macrophage populations under the experimental conditions used in our study. Namely, we reconstituted *Fth*^{R26fl/fl} or control *Fth*^{fl/fl} mice with BM cells expressing the tdTomato (*tdT*) fluorescent reporter under the control of the *LysM*^{Cre} promoter (*LysM*^{Cre}*tdTomato*^{fl/fl}; *tdT*^{LysM+} mice). We confirmed that Cre recombination occurs in approximately ~70-75% Ly6C^{Hi} monocytes/macrophages, as assessed in the liver of *tdT*^{LysM+} ⇨ *Fth*^{fl/fl} and *tdT*^{LysM+} ⇨ *Fth*^{R26Δ/Δ} chimeras, expressing FTH or not in parenchyma cells, respectively (see Fig. 3A-C below). As such, the early vs. late onset of lethality using global vs. *LysM*^{Cre} deletion, respectively, could result from (i) incomplete/mosaic deletion, (ii) *in vivo* selection for cells that express some FTH, or (iii) both (see also reply to reviewer #1, point #2 above).

Figure 3: (A) Schematic representation of chimeric mice, Tamoxifen (TAM) administration and flow cytometry analysis of LY6C^{Hi} monocyte/macrophages (CD45⁺, CD11b⁺, Ly6G⁻, CD19⁻, TCRβ⁻, LY6C^{Hi}) in

the liver of lethally-irradiated control ($Fth^{fl/fl}$) or FTH-deficient ($Fth^{R26\Delta/\Delta}$) mice, reconstituted with LysM reporter BM (tdT^{LysM+}). (B) Representative histogram and (C) percentage of LY6C^{Hi} monocyte/macrophages expressing tdTomato. Circles represent individual mice (n=2 per group) from one experiment.

We added the following paragraph to the relevant section (see page 4 of the revised manuscript; see also reply to reviewer #1, point #2 above).

*“Although $Fth^{LysM\Delta/\Delta} \Rightarrow Fth^{R26\Delta/\Delta}$ chimeras succumbed to Fth deletion, the temporal dynamics were delayed, as compared to $Fth^{R26\Delta/\Delta} \Rightarrow Fth^{R26\Delta/\Delta}$ chimeras. This likely reflects (i) mosaic/incomplete Fth deletion in $LysM^{Cre}$ cells, (ii) *in vivo* selection for cells that express some FTH or (iii) both. As such, subsequent analyses were conducted as the body weight of $Fth^{R26\Delta/\Delta} \Rightarrow Fth^{R26\Delta/\Delta}$ or $Fth^{LysM\Delta/\Delta} \Rightarrow Fth^{R26\Delta/\Delta}$ was reduced by more than 10% defining “early onset” and “late onset”, respectively.”*

It is possible that leukocytes not targeted by $LysM^{Cre}$ might contribute to the survival advantage of Fth -competent BM-derived cells in Fth -deleted mice. The following paragraph is included in the *Discussion* section (see pages 19-20 of the revised version of the manuscript) to reflect this point:

While illustrating the extraordinary capacity of circulating monocytes to partake in the inter-organ crosstalk that regulates systemic Fe and energy metabolism, this does not exclude other circulating myeloid-derived, such as myeloid-derived suppressor cells (Veglia, Sanseviero et al., 2021), from contributing to this process.”

#2. *It is puzzling that the severity of disease is higher in Fth -deficient mice reconstituted with Fth -deficient BM compared to Fth -deficient $LysM+$ cells. Although eventually, all mice succumb to disease, what are the author's thoughts on the delayed onset in a subset of mice analyzed? Is it possible to compare the effects of BMC donors with global deletion of Fth vs $LysM$ specific deletion at an earlier time point?*

Reply to #2: The explanation here is similar to the one provided in Comment 1, in that there is an incomplete (mosaic) deletion of FTH in the myeloid compartment of Fth -deficient mice reconstituted with BM from Fth -deficient $LysM+$ cells vs. Fth -deficient BM. As such, the delayed onset of lethality in the first vs. the second set of chimeras, respectively, could be the result of (i) incomplete/mosaic deletion, (ii) *in vivo* selection for cells that express some FTH, or (iii) both. In addition, it is possible that BM-derived leukocytes, other than myeloid cells, and therefore not targeted by $LysM$, can contribute to the protective effects of the BM-derived cells.

For logistical reasons it has not been possible to “to compare the effects of BMC donors with global deletion of Fth vs $LysM$ specific deletion at an earlier time point”. The lethality of the chimeric mice is heterogeneous in time and therefore it is difficult to pre-define an earlier timepoint at which this analysis could be performed. Instead, these analyses were performed when the body weight loss was 10% or higher, as this is the first reliable physiologic indicator of the irreversible onset of symptoms following FTH deletion in response to tamoxifen. We have now added this information explicitly both in the Methods (see page 43 of the revised manuscript), as well as in the Results section when presenting the model (see page 4 of the revised manuscript).

#3. *Is it possible that the fewer mitochondria in Fth-depleted macrophages can cause their death or affect BM output?*

Reply to #3. We agree with the reviewer that FTH deletion could impair BM output and performed flow cytometry analysis of BM progenitors to assess that (see reply Figure 4A, B for strategy). Indeed, myeloid BM output is reduced in chimeric mice fully deleted for FTH (black group), including myeloid progenitors (MP), common myeloid progenitors (CMP) as well as granulocyte-monocyte progenitors (GMP) (see reply Figure 4C-E below). Nevertheless, the numbers of myeloid progenitors in the BM of FTH-deficient recipient mice reconstituted with *Fth*^{LysMΔ/Δ} BM (blue group) are not significantly reduced (see reply Figure 4C-E below). This suggests that BM output is not directly linked to the deleterious phenotype observed upon FTH deletion in parenchyma. This data is not included in the body of the revised manuscript due to space constraints.

Figure 4: (A) Schematic representation of chimeric mice, Tamoxifen (TAM) administration and flow cytometry analysis of BM cells of BM chimeras. (B) Flow cytometry gating strategy to assess the numbers of myeloid progenitors in the BM of chimeric mice. Absolute number of (C) myeloid progenitors (MP), (D) common myeloid progenitors and (E) granulocyte-monocyte progenitors (GMP), in the BM of lethally-irradiated control (*Fth*^{wt/wt}) or FTH-deficient (*Fth*^{R26Δ/Δ}) mice, reconstituted with control *Fth*^{wt/wt}, *Fth*^{R26Δ/Δ} or *Fth*^{LysMΔ/Δ}. Data in (C-E) represented as individual values (circles, n=3-8 per group) and mean (red bars). Data in (C-E) is pooled from 3 independent experiments. One-Way ANOVA with Tukey's range test for multiple comparison correction was used for comparison between multiple groups. NS: non-significant, * P<0.05.

We agree with the possibility that FTH deletion could promote cell death of macrophages. This point is addressed in detail in the reply to comment #4 from reviewer #1. We included in the

revised version of the manuscript experimental evidence showing that FTH is essential to support the viability of monocyte-derived macrophages, as these cells take up mitochondria from parenchyma cells, as these engage into the different phenotypes described. Briefly, we found that FTH is essential to support macrophage viability as these cells take up mitochondria from parenchyma cells (*revised Manuscript Fig. 8C-E, I-K* illustrated below as *reply Fig. 5A-C*). Moreover, we confirmed that FTH-deficient BMDM are more susceptible to iron-mediated cytotoxicity, as compared to FTH-competent cells (*revised Manuscript Fig. 8I,L*, illustrated below as *reply Fig. 5A,B*). These data are now included in the *revised Manuscript Fig. 8I-L* and are described in the results section (see page 18 of the revised manuscript).

Figure 5: (A) Schematic representation of control ($Fth^{fl/fl}$) or FTH-deficient ($Fth^{R26\Delta/\Delta}$) BMDM generation, Tamoxifen (TAM) administration and cell viability assessment. (B) Representative brightfield and fluorescent microscopy images of viability staining (PI) of BMDM treated with vehicle, or mitochondria, 15h hours post-treatment. (C) Time course of BMDM cell death upon vehicle or mitochondria administration. For (B, C), wells receiving mitochondria was administered mitochondria purified from 60mg of liver. (D) Time course of BMDM cell death upon vehicle or Fe (FAC, ferric ammonium citrate; 125 μ M) administration. Data in (B, C) is presented as mean \pm SEM (n=3 replicates).

Specific comments

#1. In Fig 1F-H, how can we tell apart the role of Fth produced by macrophages if the population itself is depleted when Fth is ablated from LysM+ cells? Can the authors explain the 50% effect when they ablated Fth specifically on CXCR1+ cells?

Reply to #1. We agree that the depletion of macrophages upon FTH deletion in LysM+ BM and a potential role of macrophage-derived FTH are difficult to disentangle. However, the absence of these cells in the tissues strongly suggests that FTH is required for their survival or retention in peripheral organs. In keeping with this notion, we found that FTH is essential to support macrophage viability as these cells take up mitochondria from parenchyma cells (*Manuscript Fig. 8C-E, I-K*, illustrated above as *reply Fig. 5A-C*). Moreover, we confirmed that FTH-deficient BMDM are more susceptible to iron-mediated cytotoxicity, as compared to FTH-competent cells (*Manuscript Fig. 8I, L*, illustrated above as *reply Fig. 5A, D*).

While the loss of monocyte-derived macrophages precludes functional analysis of FTH within surviving cells, the loss of the population itself is informative and highlights the critical, cell-

intrinsic requirement for FTH in macrophage survival, and ultimately their function within the tissues.

The partial lethality observed in CX3CR1 BM chimeras likely reflect a level of mosaicism in cells of monocyte/macrophage lineage regarding CX3CR1 expression levels, and the required threshold for Cre-mediated recombination of the loxP sites on the *Fth* locus. As demonstrated by Yona, Kim et al. (2013), shorter-lived LY6C⁺ (BM or blood) monocytes from *Cx3cr1^{Cre}xYfp^{fl/fl}* reporter mice show roughly 30~45% YFP expression, indicating that CX3CR1-driven Cre expression may be transiently below the threshold for recombination. Although these cells may eventually differentiate into LY6C⁻ monocytes, where high CX3CR1 expression induces recombination in virtually all cells (as indicated by ~100% YFP⁺ cells) (Yona et al., 2013), it is likely that during this process, a reduced number of monocyte-derived macrophages (in particular LY6C⁺) still conserve some level of FTH expression before *Cx3cr1^{Cre}* expression is sufficient to induce its deletion. This residual FTH expression by LY6C⁺ monocyte-derived macrophages potentially provides some support, such that not all mice succumb to FTH deletion in the parenchyma. As such, we believe that this likely explains the incomplete penetrance of lethality of FTH deletion in this setting and supports the conclusion that CX3CR1⁺ monocyte-derived macrophages are critical for the rescue from the fatal phenotype induced by global FTH deletion.

We have added a brief paragraph (copied below) to the results section (see pages 5-6 of the revised manuscript), to better clarify this question:

*“This partial lethality likely reflects some level of mosaicism in CX3CR1 expression, and therefore in CX3CR1-driven Cre expression in monocyte/macrophages (Yona et al., 2013). As such, mosaic FTH deletion may justify why not all in *Fth^{Cx3cr1Δ/Δ} ⇨ *Fth^{R26Δ/Δ}** chimeras succumb to *Fth* deletion. These data suggest, nevertheless, that *Fth*-competent classical monocytes contribute to support organismal homeostasis, when *Fth* is deleted in parenchyma cells.”*

#2. *In Fig 1J-K, where do the transferred BMDM go? Do they undergo tissue differentiation?*

Reply to #2. To address this question, we performed additional adoptive transfers, similar to those illustrated in Figure 1J, K of the original Manuscript (Figure 1H in the revised version). The tdT⁺ BMDM were analyzed by flow cytometry in blood, liver and spleen, prior to, and 24h after, adoptive transfer. We detected relatively high numbers of tdT⁺ cells in the liver and spleen, but only residual numbers in the blood (see reply Fig.6A-D below). This suggests that upon intravenous administration, BMDM enter the circulation, migrate and settle in the spleen as well as in other visceral organs such as the liver. While these cells are likely also present in other organs, for logistic reasons this could not be tested. The data for liver and spleen are illustrated in Figure 1J in the revised Manuscript. The data is shown here for the reviewer (see reply Figure 6).

We added a paragraph (see below) on page 7 of the Results Section of the revised manuscript (blood analysis not included due to space constraints).

Reply Figure 6: (A) Schematic representation of Tamoxifen (TAM) administration, BMDM adoptive transfer and flow cytometry analysis of tdTomato⁺ cells recovered from the blood liver and spleen of control (*Fth^{fl/fl}*) or FTH-deficient (*Fth^{R26Δ/Δ}*) mice, adoptively transferred with control, or FTH-deficient tdTomato reporter BMDM (*tdT^{R26}*, *tdT^{R26}Fth^{R26Δ/Δ}*, respectively). (B) Number of adoptively-transferred tdT⁺ cells per μL of blood from recipient mice, 24h post-adoptive transfer, on day 7 post-tamoxifen treatment. (C) Total number of adoptively-transferred tdT⁺ cells per liver from recipient mice, 24h post-adoptive transfer, on day 7 post-tamoxifen treatment. (D) Total number of adoptively-transferred tdT⁺ cells per spleen from recipient mice, 24h post-adoptive transfer, on day 7 post-tamoxifen treatment. Data in (B-D) represented as individual values (circles, n=4-6 per group) and mean (red bars) and is pooled from 2 independent experiments. One-Way ANOVA with Tukey's range test for multiple comparison correction was used for comparison between multiple groups. NS: non-significant, * P<0.05, ** P<0.01, *** P<0.001, **** P<0.0001.

*The number of Fth-deleted tdT⁺ cells recovered from the liver and spleen of recipient Fth-deleted (*Fth^{R26Δ/Δ}*) mice, 24 hours after adoptive transfer of tdT⁺ BMDM, was markedly lower, as compared to Fth-competent tdT⁺ cells (Fig. 1H,J). This indicates that Fth-deleted BMDM have an impaired capacity to migrate and/or differentiate into tissue-resident macrophages. Moreover, recovery of FTH-competent tdT⁺ BMDM from the liver, but not the spleen, of recipient FTH-deficient (*Fth^{R26Δ/Δ}*) was higher than in control (*Fth^{fl/fl}*) recipient mice (Fig. 1H,J). This suggests that FTH-deficient organs (e.g. liver) have an increased capacity to recruit, and/or retain, monocyte-derived macrophages upon the stress imposed by FTH deletion.*

To gain further insight into the outcome of the transferred BMDM, we used the same flow cytometry antibody panel to define the BMDM marker phenotype prior to the adoptive transfer. We then performed tSNE analysis both on BMDM and all viable CD45⁺ tdTomato⁺ cells recovered from the liver. Briefly, we observed identical tSNE population profiles after transferring control or *Fth*-deleted tdT⁺ cells indicating that FTH deletion in either parenchyma or donor cells did not influence the differentiation of BMDM into monocyte-derived tissue macrophages following adoptive transfer (see reply Fig.7A below). Both BMDM and tdTomato⁺ cells recovered from the liver presented some heterogeneity (e.g. CX3CR1, LY6C, CD11B expression), likely corresponding to different subpopulations (see reply Fig.7B below). However, regardless of donor or recipient genotype, the transferred tdT⁺ cells acquired a more monocyte-like marker expression, as compared to cells prior to transfer, with higher expression of LY6C and lower expression of F4/80 (see reply Fig.7B below). These data are illustrated in Figure 1K, L of the revised version of the manuscript:

Reply Figure 7: (A) tSNE analysis of tdTomato⁺ cells recovered from the liver of control ($Fth^{fl/fl}$) or FTH-deficient ($Fth^{R26\Delta/\Delta}$) mice following adoptive transfer with control, or FTH-deficient tdTomato reporter BMDM (tdT^{R26} , $tdT^{R26} Fth^{R26\Delta/\Delta}$, respectively). Rightmost panel shows merged tSNE maps combining the three groups. (B) tSNE plots showing marker expression profiles in control or FTH-deficient (tdT^{R26} , $tdT^{R26} Fth^{R26\Delta/\Delta}$, respectively) BMDMs, prior to adoptive transfer (top 2 rows), as well as marker expression profiles in tdTomato⁺ cells recovered from the liver of control ($Fth^{fl/fl}$) or FTH-deficient ($Fth^{R26\Delta/\Delta}$) mice following adoptive transfer with control, or FTH-deficient tdTomato reporter BMDM (tdT^{R26} , $tdT^{R26} Fth^{R26\Delta/\Delta}$, respectively). Each plot displays the expression of a specific marker across the tSNE projection for each genotype, highlighting phenotypic changes dependent on FTH expression. Data in (A; B) represented as a dot plot where dots indicate individual cells. tSNE was used to generate projections. tSNE projections generated from 2500 cells (downsampled) for cells prior to adoptive transfer, and from 701 to 1932 cells for BMDM recovered following adoptive transfer (pooled from 3 mice per group).

The following paragraphs were added to the Results Section of the revised version of the manuscript on pages 7-8:

“To gain further insight into the differentiation of tdT⁺ BMDM after adoptive transfer, we performed a t-SNE (t-distributed Stochastic Neighbor Embedding) analysis based on CX3CR1, LY6C, F4/80 and CD11B expression, prior to, and 24 hours after BMDM transfer (Fig. 1H,K). The t-SNE profile of tdT⁺ BMDM before and after adoptive transfer was distinct, as assessed in liver of recipient mice (Fig.

1H,K,L). There was heterogeneity in CX3CR1, LY6C, F4/80 and CD11B expression with a sub-population of cells increasing Ly6C and decreasing CX3CR1 and F4/80 expression, while maintaining CD11b and tdT expression (Fig. 1H,K,L). This suggests that upon adoptive transfer, tdT⁺ BMDM differentiate into distinct tissue-specific monocyte-derived macrophage subpopulations.

Fth-deleted and Fth-competent BMDM tdT⁺ cells from the liver of recipient Fth-deleted ($Fth^{R26\Delta/\Delta}$) mice presented similar t-SNE profiles (Fig. 1H,K,L). There was also no discernable difference in the t-SNE profiles of Fth-competent tdT⁺ cells from the liver of Fth-deleted ($Fth^{R26\Delta/\Delta}$) vs. control ($Fth^{fl/fl}$) mice (Fig. 1H,K,L). This suggests that FTH does not regulate the initial capacity of BMDM to differentiate into monocyte-derived tissue macrophages. “

#3. Can the authors explain the discrepancy between the survival when mice were reconstituted with Fth-sufficient BM vs Fth-sufficient BMDM (the grey symbol in both panels D and K of Fig 1)? Presumably, in both cases survival extension is maintained by monocyte-derived macrophages. In D, only 25% of mice succumb to disease but in K all mice succumb to disease. Is the recovery of monocyte-derived macrophages the same in Fth-deficient and Fth-sufficient BMDM?

Reply to #3. The discrepancy in survival noted by the reviewer is likely attributed to the supply or not of Fth-competent monocyte-derived macrophages derived from BM progenitors. This occurs in BM chimeric mice (*revised version of the manuscript Fig .1D*) or parabiotic mice (*revised version of the manuscript Fig .1G*) but not upon the adoptive transfer of BMDM, where the numbers of Fth-competent monocyte-derived macrophages is limited by the frequency and number of BMDM transferred (*revised version of the manuscript Fig .1I*).

We performed additional adoptive transfers of Fth-deficient or Fth-competent BMDM to compare the number monocyte-derived macrophages recovered from the liver and spleen 24 hours post-transfer. The number of tdTomato⁺ recovered from the liver or spleen of recipient Fth-deleted ($tdT^{R26}Fth^{R26\Delta/\Delta}$) mice was lower when receiving Fth-deleted ($tdT^{R26}Fth^{R26\Delta/\Delta}$) vs. Fth-competent (tdT^{R26}) BMDM (see reply to point #2 and *reply Fig.6C-D*). This suggests that FTH deletion impairs the ability of BMDM to migrate to, and/or settle in, tissues, and consequently, compromises their capacity to support tissue function. Of note, we did not recover tdT⁺ donor cells from these organs 72h post-BMDM transfer. This suggests that these cells are short-lived and therefore require continuous replenishment from the BM progenitors.

#4. In Fig S6, can the authors clarify the time points post tamoxifen treatment chosen for the analysis? If the animals were about to die, isn't it expected they would move and eat less? Based on this data, can the authors conclude about energy metabolism in this scenario?

Reply to #4. Following FTH deletion, mice keep a normal body temperature/activity/food intake, up to the point where body weight starts to decrease. From this moment, most mice succumb to FTH deletion within 4 to 6 days, with progressively worsening symptoms (e.g. drop in body temperature, decreased locomotor activity, etc.). Because of this, our window of opportunity to observe the metabolic dysfunction imposed by global FTH deletion is relatively narrow. As such, we chose to start the metabolic analysis in mice that have lost 10% or more, in body weight, as a telltale sign that the metabolic alterations will soon follow. In the experiments depicted in original manuscript *Fig. S6*, for chimeric mice where FTH is deleted in both parenchyma and hematopoietic tissue ($Fth^{R26\Delta/\Delta} \Rightarrow Fth^{R26\Delta/\Delta}$; i.e. full FTH deletion), analysis was performed on day

7 post-tamoxifen administration. For $Fth^{LysM\Delta/\Delta} \Rightarrow Fth^{R26\Delta/\Delta}$ (i.e. LysM-FTH KO and global FTH deletion in recipient), analysis was performed on day 20 post-tamoxifen administration.

We focused our conclusions on the metabolic data based on what occurs once FTH deletion induces metabolic alterations, which point towards a direct or indirect the involvement of FTH on those metabolic functions. Furthermore, these experiments were designed to also address whether BM transplantation with FTH-competent BM, would lead to improved metabolic parameters on FTH-deleted recipients (i.e. rescue), as opposed to transplantation with global FTH-deleted, or LysM-FTH KO BM.

#5. *In Fig 4, similar to comments above, how to disentangle the effect of FTH from other roles macrophages could be playing on homeostasis maintenance since they are also largely absent when using a conditional deletion of FTH in LysM+ cells? Can the authors perform these analyses at an earlier time point, long before animals were supposed to die?*

Reply to #5. The explanation here is similar to the one provided in Comment 1, in that there is an incomplete (mosaic) deletion of FTH in the myeloid compartment of FTH-deficient mice reconstituted with BM from FTH-deficient LysM+ cells vs. FTH-deficient BM. As such, the delayed onset of lethality in the first vs. the second set of chimeras, respectively, could be the result of (i) incomplete/mosaic deletion, (ii) *in vivo* selection for cells that express some FTH, or (iii) both. In addition, it is possible that BM-derived leukocytes, other than myeloid cells (and therefore not targeted by LysM) can contribute to the protective effects of the BM-derived cells.

The lethality of the BMC is heterogeneous in time and therefore it is not possible to address a single timepoint at which this analysis could be performed. Instead, we have chosen to perform the analysis once body weight loss reaches 10% or higher, as this is the first indicator of the irreversible onset of symptoms following FTH deletion. We have now added this explicitly in the methods (see page 43 of the revised manuscript) and results section when presenting the model (see page 4 of the revised manuscript).

#6. *In Fig 5A-B, was the WB performed on whole tissue lysates? Since specific deletion of FTH in LysM+ cells also results in decreased numbers of monocyte-derived macrophages, how can the authors exclude the possibility that the FTH signal they see is not coming from the monocyte-derived macrophages that are also found in the heart, liver, kidneys and lungs?*

Reply to #6. The WB in the manuscript Fig. 5A,B were performed in whole tissue lysates. We believe that given the data, we can assess the relative contribution of parenchyma to the whole organ FTH levels by comparing control chimeras (white group; $Fth^{wt/wt} \Rightarrow Fth^{fl/fl}$) to FTH-deleted recipients reconstituted with FTH-competent BM (grey group; $Fth^{fl/fl} \Rightarrow Fth^{R26\Delta/\Delta}$), where the decrease in FTH signal represents the amount of FTH that would be expressed by parenchymal cells. Conversely, by comparing FTH-deleted recipients reconstituted with either FTH-competent BM (grey group; $Fth^{fl/fl} \Rightarrow Fth^{R26\Delta/\Delta}$) or FTH-deficient BM (blue group; $Fth^{LysM\Delta/\Delta} \Rightarrow Fth^{R26\Delta/\Delta}$), we can estimate the amount of FTH expressed by LysM+ cells in the organs. Given that both the control (white group; $Fth^{wt/wt} \Rightarrow Fth^{fl/fl}$) and $Fth^{fl/fl} \Rightarrow Fth^{R26\Delta/\Delta}$ (grey group) have comparable numbers of monocyte-derived macrophages in the organs (Manuscript Fig. 1F), we expect that the FTH signal remaining in $Fth^{fl/fl} \Rightarrow Fth^{R26\Delta/\Delta}$ chimeras (grey group), corresponds to the amount of FTH expressed by cells of hematopoietic origin in the organs. Nevertheless, because $Fth^{LysM\Delta/\Delta} \Rightarrow Fth^{R26\Delta/\Delta}$ chimeras (blue group) have lower numbers of monocyte-derived macrophages, it is possible that the reduction in FTH signal observed in $Fth^{LysM\Delta/\Delta} \Rightarrow Fth^{R26\Delta/\Delta}$ (blue group) vs. $Fth^{fl/fl} \Rightarrow Fth^{R26\Delta/\Delta}$ (grey group) chimeras is the result of both FTH deletion in LysM+ cells, as well as reduced numbers of LysM+ monocyte-derived macrophages in the organs, as suggested by the reviewer. These hypotheses are not mutually exclusive, but we believe they do not alter the interpretation of the data.

#7. In Fig 7, is possible that the deletion of *Fth* is under reported using the *TdTomato* strategy, since the efficiency of *Rosa26* is likely higher than *Fth* flox (*Rosa26* locus is more permissive). However, they do see an effect in the macrophages from *Fth* flox mice compared to *WT* suggesting there is an enrichment of *Fth*-deficient cells among the tomato+ cells. Is it possible to sort the tomato+ cells and check for *Fth* expression?

Reply to #7. As suggested by the reviewer, we used RT-qPCR to quantify the expression of *Fth* from tomato+ cells sorted from the BM chimeras as in the manuscript Figure 7A. Given that both tomato expression and FTH deletion are under the control of the *LysM* promoter, tomato+ cells from $tdT^{LysM}Fth^{LysM\Delta/\Delta}$ BM donors (blue group) are efficiently deleted for FTH in both the liver, heart and WAT of chimeric mice (see reply Figure 8 below).

Reply Figure 8: (A) Schematic representation of chimeric mice, TAM administration and fluorescence-activated cell sorting (FACS) of *LysM*⁺ monocyte/macrophages (CD45⁺, CD11b⁺, Ly6G⁻, CD19⁻, TCRβ⁻) in liver, heart and WAT. Expression of *Fth* mRNA in the (B) liver, (C) Heart and (D) WAT of $tdT^{LysM} \Rightarrow Fth^{fl/fl}$, $tdT^{LysM} \Rightarrow Fth^{R26\Delta/\Delta}$ and $tdT^{LysM}Fth^{LysM\Delta/\Delta} \Rightarrow Fth^{R26\Delta/\Delta}$ chimeric mice, on day 19 post-TAM administration. Data in (B-D) is presented as individual values (circles, n = 2-5) and mean (red bars). One-Way ANOVA with Tukey's range test for multiple comparison correction was used for comparison between multiple groups. NS: non-significant, * P<0.05, ** P<0.01.

8. Is S9D reporting total number of macrophages in the indicated organs or number of tomato+ macrophages? Can the authors show the proportion of tomato+ cells in *Fth*-deficient vs sufficient mice?

Reply to #8. We agree with the reviewer, that the figure legend was not clear regarding what's depicted. In the original Manuscript Figure S9D, we show the total number of cells per organ indicated. In this figure, we labeled these cells "macrophages", however, based on the markers used (CD45+, CD11b+, tdT+, Ly6G-, CD19-, TCRβ-), we have now revised the figure labels to refer to this population as "Monocyte/macrophage". In addition, we have also included cell proportion data of these cells in the indicated organs (please see revised Fig. S9D below).

Minor comments

#1 Fig 6I came out blank (no bands are visible, at least in the PDF I received).

Reply to #1. We believe that this may occur during the PDF conversion used by EMBO J and accessed by the reviewer. We confirmed that Fig. 6I contains the information intended.

#2. Fig S9C has a typo on the axis: *dTomato* instead of *tdTomato*. Also S9D, *LysM* mice are referred as *LyzM* mice.

Reply to #2. The two typos were corrected.

#3. Fig S4A and Fig 2J. *Fth*^{wt/wt} is referred in the text as *Fth*^{f/f}

Reply to #3. We apologize for the mix-up, and have corrected the relevant figures and corresponding text.

References:

Tsou CL, Peters W, Si Y, Slaymaker S, Aslanian AM, Weisberg SP, Mack M, Charo IF (2007) Critical roles for CCR2 and MCP-3 in monocyte mobilization from bone marrow and recruitment to inflammatory sites. *J Clin Invest* 117: 902-9

Veglia F, Sanseviero E, Gabrilovich DI (2021) Myeloid-derived suppressor cells in the era of increasing myeloid cell diversity. *Nat Rev Immunol* 21: 485-498

Yona S, Kim KW, Wolf Y, Mildner A, Varol D, Breker M, Strauss-Ayali D, Viukov S, Guilliams M, Misharin A, Hume DA, Perlman H, Malissen B, Zelzer E, Jung S (2013) Fate mapping reveals origins and dynamics of monocytes and tissue macrophages under homeostasis. *Immunity* 38: 79-91

Dear Miguel,

Thank you for the submission of your revised manuscript (EMBOJ-2025-120530R) to The EMBO Journal for our consideration. Your manuscript has been sent back to the two original referees who had previously assessed the first version of your manuscript, and we have now received their comments, which you can find below.

As you will see, both referees recognize that all initially raised concerns have been satisfactorily addressed, and explain that the manuscript has been significantly strengthened by the addition of new data and further explanations/clarification in the text. None of the referees have any other comments. In light of this input, I am very pleased to inform you that your manuscript has in principle been accepted for publication in The EMBO Journal.

Before we can move forward with formal acceptance and publication, there are a few changes and corrections we need you to make in a final version of the manuscript:

- Please note that, as per our journal's policy, "equal contribution" is permitted only for co-first or co-corresponding authors. Co-second authorship is not permitted. Would you be so kind as to review and revise accordingly the author list? I should also note that at our journal we use the CRediT contributor role taxonomy system for specifying the nature of each co-author's contribution in detail (please see below for more information), and we also allow and encourage specifying author contributions at the Figure panel level (for more information please see "Author contributions" in our guide to authors: <https://www.embopress.org/page/journal/14602075/authorguide#manuscriptpreparation>).

- "SUMMARY" should be renamed to "Abstract".

- Please provide a list of up to 5 keywords (preferably broad terms to enhance search engine discoverability of your article) after the Abstract of your revised manuscript.

- Please add the permanent and specific URL of your deposited dataset at GEO in the Data availability statement of your revised manuscript. You are also kindly requested to make sure that the data will be publicly available at the time of publication.

- Please note that a conflict-of-interest statement (with heading "Disclosure and competing interests statement") is mandatory; for more information, please see our guide to authors: <https://www.embopress.org/page/journal/14602075/authorguide#conflictsofinterest>.

- The author contributions statement should be removed from the manuscript file. Instead, we use CRediT to specify the contributions of each author in the journal submission system. Please feel free to use the free text box to provide more detailed descriptions during submission. See also our guide to authors for more information: <https://www.embopress.org/page/journal/14602075/authorguide#authorshippinguidelines>.

- We noticed that the Figure number is missing in callout "Fig. E,G" on p15. Please correct the callout in your revised manuscript.

- We noticed that your manuscript contains 11 "Supplementary Figures". We encourage you to select a limited number (typically around 5; no more than 7-8) of them for inclusion in the article as Expanded View figures in order to improve their accessibility, visibility and utility. Any extra figures that are not promoted to the Expanded View should be included in a "traditional" supplementary PDF file (along with supplementary text and tables) now called the Appendix. Please see our guide to authors for more information: <https://www.embopress.org/page/journal/14602075/authorguide#expandedview>.

- Main and Expanded View Figures need to be uploaded as individual, high-resolution Figure files; their legends should be placed below the References list in the main manuscript file. The nomenclature for Expanded View (EV) Figures should be "Figure EV1#", instead of "Supplementary Figure #".

- The Appendix file must be uploaded as a single PDF file; the heading on its title page should be "Appendix for:", followed by the manuscript's title, and a Table of Contents including page numbers for all listed items. The nomenclature for the Appendix Figures should be "Appendix Fig. S#" etc., throughout the Appendix file and the main manuscript file.

- The materials and methods need to be described in the manuscript using our structured methods format, which is now required for all research articles. According to this format, the Methods section includes a single "Reagents and Tools Table" -listing key reagents, experimental models, software and relevant equipment including their sources and relevant identifiers- followed by a "Methods and Protocols" section describing the methods. Please download and fill our Reagents and Tools Table template (.docx), which you can find in our author guide: <https://www.embopress.org/page/journal/14602075/authorguide#structuredmethods>. When submitting your revised manuscript, please do not include the Reagents and Tools Table in the Methods section of the manuscript but instead upload it as a

separate file choosing the file type "Reagent Table".

- "MATERIALS AND METHODS" should be renamed to "Methods".

- Please upload all Source Data for the main Figures of your manuscript (you are also welcome to upload the Source Data for the Expanded View and Appendix Figures, if you wish), according to the instructions you have previously received from our team, and along with a completed Source Data checklist.

- Please note that EMBO press papers are accompanied online by:

A) a short (2 sentences) summary of the findings and their significance,

B) 2-5 short bullet points highlighting the key results, and

C) a synopsis image in .jpg or .png format that is exactly 550 pixels wide and 300-600 pixels high (the height is variable). Please note that all text needs to be legible at the final size.

Please upload this information along with your revised manuscript (the text for A and B should be provided in a separate Word file).

- During our routine data checks, our data editors have raised the following query regarding figures and legends. Please make sure that the request below is completely addressed in the final version of your manuscript (please highlight all changes in the revised manuscript):

- Please provide the exact p-values in the legends of Figures 1D, F, G, I, J; 2B, C, E, F, G, H, I, J; 3B, C, D, E, F, G; 4B, D, F, H; 5B, E; 6B, C, F, G, J; 7F, G; 8A, B, E, H, K, L.

- The headings and order of the manuscript sections must be corrected as follows: Title page - Abstract and Keywords - Introduction - Results - Discussion - Methods - Data Availability - Acknowledgements - Disclosure and Competing Interests Statement - References - Figure Legends - main Tables (if there are any) - Expanded View Figure Legends.

Please also note that as part of the EMBO publications' Transparent Editorial Process, The EMBO Journal publishes online a Peer Review File along with each accepted manuscript. This File will be published in conjunction with your paper and will include the referee reports, your point-by-point response and all pertinent correspondence relating to the manuscript. You can opt out of this by letting the editorial office know (contact@embojournal.org). If you do opt out, the Peer Review File link will point to the following statement: "No Peer Review File is available with this article, as the authors have chosen not to make the review process public in this case."

We look forward to seeing a final version of your manuscript as soon as possible. Please let us know if you have any questions and use this link to submit your revision: <https://emboj.msubmit.net/cgi-bin/main.plex>.

Best regards,

Ioannis

Referee #1:

The authors have addressed all the (minor) concerns raised. Amazing manuscript!

Referee #2:

The authors have satisfactorily addressed all concerns raised by both reviewers and I recommend the manuscript to be published in the revised format. They added new data that supports the overall conclusions of the manuscript and clarified some of the concerns with new explanation throughout the text (new results and discussion). Specifically, they addressed the major

concerns raised by both reviewers which were related to the incomplete penetrance of the LysM strategy and the viability of cells lacking FTH. The new figure 8 panels adequately addressed the latter point and show that FTH support the viability of monocyte-derived macrophages. The new panels in Figure 1 addressed the outcome of transferred BMDM in the absence of FTH and the revised text in results section addressed the limitations of the LysM cre approach.

All editorial and formatting issues were resolved by the authors.

Dear Miguel,

Congratulations on an excellent manuscript! I am very pleased to inform you that it has been accepted for publication in The EMBO Journal. Thank you for comprehensively addressing the initially raised referee criticisms and the editorial requests for corrections and changes.

If you have any questions, please do not hesitate to contact the Editorial Office. Thank you for your contribution to The EMBO Journal. Working with you has been a pleasure!

Best regards,

Ioannis
